# The shared and unique neural correlates of personal semantic, general semantic, and episodic memory

**Annick FN Tanguay[1,2]\*, Daniela J Palombo[3], Brittany Love[1], Rafael Glikstein[1], Patrick SR Davidson[1], Louis Renoult[2]\***

[1]School of Psychology, University of Ottawa, Ottawa, Canada; [2]School of Psychology, University of East Anglia, Norwich, United Kingdom; [3]Department of Psychology, University of British Columbia, Vancouver, Canada

**\*For correspondence:**
atang027@uottawa.ca (AFNT);
L.Renoult@uea.ac.uk (LR)

**Competing interest:** The authors declare that no competing interests exist.

**Abstract** One of the most common distinctions in long-term memory is that between semantic (i.e., general world knowledge) and episodic (i.e., recollection of contextually specific events from one's past). However, emerging cognitive neuroscience data suggest a surprisingly large overlap between the neural correlates of semantic and episodic memory. Moreover, personal semantic memories (i.e., knowledge about the self and one's life) have been studied little and do not easily fit into the standard semantic-episodic dichotomy. Here, we used fMRI to record brain activity while 48 participants verified statements concerning general facts, autobiographical facts, repeated events, and unique events. In multivariate analysis, all four types of memory involved activity within a common network bilaterally (e.g., frontal pole, paracingulate gyrus, medial frontal cortex, middle/ superior temporal gyrus, precuneus, posterior cingulate, angular gyrus) and some areas of the medial temporal lobe. Yet the four memory types differentially engaged this network, increasing in activity from general to autobiographical facts, from autobiographical facts to repeated events, and from repeated to unique events. Our data are compatible with a component process model, in which declarative memory types rely on different weightings of the same elementary processes, such as perceptual imagery, spatial features, and self-reflection.

## Editor's evaluation

There has been much theoretical and empirical work focused on the distinctions between episodic and semantic memory. The current work is important because it provides convincing and compelling evidence that this two-part model is not sufficient to explain the range of memory retrieval modes, particularly when considering personal semantic memory. This will be of great interest to memory researchers and will become a benchmark for future studies of autobiographical memory.

## Introduction

One of the most fundamental distinctions in human long-term memory is that between semantic and episodic (*Greenberg and Verfaellie, 2010*; *Squire, 2009*; *Tulving, 1972*; *Tulving, 1983*; *Tulving, 2002*). Semantic memory refers to one's non-personal, general knowledge of the world (e.g., *I know that yoga is a relaxing form of exercise*), whereas episodic memory concerns the recollection of contextually specific events from one's personal past (*I remember arriving quite late to my yoga class this weekend, which made my teacher angry*). This distinction is classic (*Herrmann, 1982*; *Renoult and Rugg, 2020*) and remains crucial for cognition and neuroscience today. Importantly, although these two putative memory systems have long been recognized as 'partially overlapping' (*Tulving,*

*1972*) and interacting (*Greenberg and Verfaellie, 2010*), they have largely been investigated via separate research traditions.

In recent years, however, emerging cognitive neuroscience data have suggested a surprisingly large overlap between the neural correlates of semantic and episodic memory. For example, when one compares the functional neuroimaging findings from the semantic memory literature (a 'general semantic network'; *Binder, 2016*; *Binder et al., 2009*; *Binder and Desai, 2011*) to those from the episodic memory literature (an 'episodic core recollection network'; *Rugg and Vilberg, 2013*; *Svoboda et al., 2006*), it becomes evident that the two networks share the midline frontal, middle temporal, parahippocampal, ventral parietal, and midline posterior regions (*Renoult et al., 2019*). Although based on the previous literature one might expect that the hippocampus would be much more strongly associated with episodic memory, and the anterior temporal lobe with semantic memory (*Lambon Ralph et al., 2017*), recent research suggests otherwise: The hippocampus can facilitate the acquisition and retrieval of a rich semantic network (*Blumenthal et al., 2017*; *Cutler et al., 2019*; *Klooster et al., 2020*; *Sheldon and Moscovitch, 2012*), partly through relational processing (*Duff et al., 2019*) or pattern completion (*Solomon and Schapiro, 2020*). Anterior hippocampal atrophy features in neurodegenerative diseases affecting semantic and episodic memory alike (*Chapleau et al., 2016*). Similarly, the lateral anterior temporal lobe may also be involved in episodic recollection, as shown by its recruitment when successfully learning word pairs (*de Chastelaine et al., 2016*; *Renoult et al., 2019*).

The considerable overlap between the two has sparked a rethinking of the classic semantic-episodic dichotomy (*Renoult et al., 2019*). A major impediment, though, has been that a large proportion of the existing data come from indirect comparisons of semantic and episodic memory between different experiments. As mentioned above, this is partly due to semantic and episodic memory being investigated in somewhat separate fields of study: comprehension of the world through language or object identification for example (*Binder and Desai, 2011*; *Kumar, 2021*; *Lambon Ralph et al., 2017*) vs. memory retrieval to think about the past, and particularly, personal events (*Maguire, 2001*; *Svoboda et al., 2006*). Memory researchers sometimes control for semantic or basic linguistic processes, but rarely fully match the semantic and episodic aspects or compare them. This may stem from the difficulty in minimizing any differences between semantic and episodic memory on task demands, including retrieval times (*Renoult and Rugg, 2020*), and other cognitive operations (e.g., control; *Vatansever et al., 2021*).

Another critical issue is that several types of declarative memory do not fit easily into the standard semantic-episodic dichotomy. This is the case for 'personal semantics', which involves knowledge of one's personal past (*Conway, 2005*; *Grilli and Verfaellie, 2014*; *Grilli and Verfaellie, 2015*; *Martinelli et al., 2013*; *Renoult et al., 2012*). Personal semantics lives conceptually between (or, perhaps, across) the boundaries of semantic and episodic memory. It is personal (like episodic memories), but detached from its context of acquisition (like semantic memories). Personal semantics includes personal factual knowledge, such as autobiographical facts (e.g., *I am adept at yoga*), and knowledge of repeated personally experienced events, including contextual details that have been abstracted across several instances (e.g., *my yoga routine when going to the gym*). Many early descriptions assumed that personal semantics was part of semantic memory (*Cermak and O'Connor, 1983*; *Kopelman et al., 1989*), but more recent evaluations suggest that this view was too simple: whereas some forms of personal semantics—such as autobiographical facts—appear to have neural correlates similar to semantic memory, others—such as memories of repeated events—have neural correlates that are more similar to those of episodic memory. For instance, in amnesic patients, memories of unique and repeated episodes are often impaired together, whereas knowledge of general and personal facts are typically better preserved (reviewed in *Renoult et al., 2012*).

Personal semantics is an understudied form of memory relative to semantic and episodic memory, even though it has been reported to be the most common form of autobiographical memory elicited in free and cued recall (*Barsalou, 1988*), and in brain stimulation studies (*Curot et al., 2017*). Personal semantics play an important role in the retrieval of specific events (*Conway, 1987*; *Conway and Pleydell-Pearce, 2000*; *Haque and Conway, 2001*), and such facilitation can vary with the type of personal semantics (e.g., repeated events induce a greater facilitation than autobiographical facts; *Sheldon et al., 2020*). The conceptualization of personal semantics is grounded in the close proximity between semantic and episodic memory (as reviewed above; *Duff et al., 2019*; *Renoult et al., 2019*).

For instance, some general semantic concepts, such as knowledge about unique entities like famous individuals, can be 'autobiographically significant' and are tightly associated with episodic memories (*Lambert, 2020*; *Renoult et al., 2015*; *Westmacott et al., 2004*; *Westmacott and Moscovitch, 2003*).

Despite the apparent importance of personal semantics and the notable interest generated by a taxonomy of personal semantics (*Renoult et al., 2012*), few functional neuroimaging studies have directly compared personal semantics to either semantic or episodic memory, and even fewer have compared different types of personal semantics to one another or personal semantics to both semantic and episodic memory. On the one hand, these have indicated some differences across general semantic, personal semantic, and episodic memory: Autobiographical facts have been associated with greater activity in the left medial frontal cortex and retrosplenial cortex than general facts (*Maguire and Frith, 2003*), but have not differed in lateral temporal activity (*Maguire and Frith, 2003*; *Maguire and Mummery, 1999*). Repeated events have elicited less activity than unique events in the frontal pole, parahippocampal gyrus, posterior cingulate, and precuneus (*Ford et al., 2011*; *Holland et al., 2011*; *Levine et al., 2004*), but not differed in hippocampal activity (notably, *Addis et al., 2004a*; *Addis et al., 2004b*). On the other hand, many existing data are compatible with the idea that these different types of memories involve different levels of activity within a common network. For instance, several neuroimaging studies have reported a graded increase in activity across semantic memories, autobiographical facts, and episodic memories in medial prefrontal, temporal polar, and retrosplenial cortex (*Maguire, 2001*; *Maguire and Frith, 2003*; *Maguire and Mummery, 1999*). In a recent event-related potential study (*Renoult et al., 2016*) that compared semantic memory (i.e., general facts), two types of personal semantics (i.e., autobiographical facts and repeated events) and episodic memory (i.e., unique events), a similar graded pattern occurred. Both types of personal semantics produced intermediate mean amplitudes for the N400, an index of semantic processing (*Kutas and Federmeier, 2011*), and for the late positive component, an index of episodic processing (*Rugg and Curran, 2007*), compared to general facts and unique events (*Renoult et al., 2016*). The two types of personal semantics did not differ from one another in this study (*Renoult et al., 2016*).

Given the questions about the relation between memory types across the semantic-episodic spectrum, here we used fMRI to directly compare brain activity during general semantic, episodic, and personal semantic processing within the same participants. They indicated whether closely matched sentences were true in four conditions: (1) facts about people in general (i.e., general facts, reflective of semantic memory), (2) personal events that happened once (i.e., unique events, reflective of episodic memory), (3) facts about themselves (i.e., autobiographical facts, a type of personal semantics), and (4) events that happened repeatedly (i.e., repeated events, a type of personal semantics). The design enabled us to make as close of a comparison as possible between memory conditions. For example, the stimuli differed minimally in wording across conditions, being adjusted only in self-reference (i.e., referring to the self in all conditions but general facts) and temporal specificity (general for both types of facts, somewhat more specific for repeated events, and very specific for unique events). Further, even though semantic and episodic retrieval typically involve different retrieval times, our comparison of these memory types is unbiased by response time differences between conditions (*Renoult et al., 2016*).

We leveraged multivariate analysis methods (i.e., partial least squares [PLS]; *Krishnan et al., 2011*; *McIntosh et al., 2004a*; *McIntosh and Lobaugh, 2004b*) to compare brain activity across these four memory conditions within a single analysis in a sample of 48 participants. Our analytical approach tested whether data in the spatial and temporal domain explained adequately two hypothesized relations between memory conditions: (1) a continuum of contextual specificity, and (2) a dissociation between knowledge of facts and recollection of events.

Although we expected memory types to be distinguishable, we also expected that the neural correlates of autobiographical facts would appear more similar to general facts whereas the neural correlates of repeated events would appear more similar to unique events (*Renoult et al., 2012*). Further, the relation between memory types could be best described as one of a continuum: They would engage predominantly a common set of regions from the core memory network, but with increased intensity from the least (i.e., general facts) to the most (i.e., unique events) contextually specific memory type. These relations would depend on the differential engagement of component

processes, such as sensory-perceptual imagery, spatial and temporal features, and self-reflection, which should be observed at the cognitive and brain level.

In a behavioral study, we considered three key component processes that could dissociate the four memory types: self-relevance (expecting it to be lower for general facts than all personal forms of memory), visual details (expecting a linear increase), and the integration of details within a scene (expecting it to be low for general/autobiographical facts and high for repeated/unique events). Accordingly, brain regions involved in visuospatial processing and imagery, like the precuneus, and in self-reflection, like the medial prefrontal cortex, should be more tightly associated with memories of unique events, and minimally associated with general factual knowledge (*Renoult et al., 2012*). Similarly, brain regions known to have a crucial role in spatial processing and in representing scenes, such as the hippocampus, parahippocampal cortex, and retrosplenial cortex, should show greater activity for memories of unique and repeated events than for autobiographical and general facts. The visual details and scene components may depend on hippocampal processes, like relational (*Duff et al., 2019*) or constructive processes (*Hassabis and Maguire, 2007*; *Schacter and Addis, 2007*), to some extent, as the association with a diverse source of personal information becomes stronger from general facts to unique events (*Sheldon et al., 2020*; *Sui and Humphreys, 2015*).

In this study, we aimed to go beyond dichotomies commonly used in memory research (e.g., semantic vs. episodic, anterior vs. posterior brain regions, different vs. identical) to examine the multi-dimensional and complex relations across the spectrum in declarative memory, and importantly do so through direct comparisons. Our operationalization captures the prototypical definitions of several memory types, in close alignment with a taxonomy of personal semantics (*Renoult et al., 2012*), and possible characteristic function in daily life. Additionally, the analyses aim to uncover patterns that could act like heuristics to characterize declarative memory function. Critically, however, our additional focus on component processes in the behavioral study (i.e., amount of visual details, ability to evoke a scene, self-relevance) relies on the theoretical perspective that relations between memory types can be explained through the information accessed and cognitive processes engaged. Thus, it is implicit to our approach that the relations across memory types are not rigid and could be altered depending on task or personal goal (e.g., *Grilli and Verfaellie, 2016*). This study seeks to develop a framework suitable to bridge the divide in research about semantic and episodic aspects in declarative memory, and offers a complementary approach to explore the multiplicity of factors that coalesce to define the mnesic experience.

**Table 1.** *t*-test values for pairwise comparisons of memory types on self-relevance and visual details.

| Comparison | t(105) | p | g | CI 95% of g |
|---|---|---|---|---|
| Self-relevance | | | | |
| General facts vs. autobiographical facts | −13.94 | < .001* | −1.35 | [−1.61, −1.08] |
| General facts vs. repeated events | −12.23 | < .001* | −1.18 | [−1.43, −0.93] |
| General facts vs. unique events | −14.90 | < .001* | −1.44 | [−1.71, −1.17] |
| Autobiographical facts vs. repeated events | −0.61 | .541 | −0.06 | [−0.25, 0.13] |
| Autobiographical facts vs. unique events | −4.49 | < .001* | −0.44 | [−0.63, −0.24] |
| Repeated events vs. unique events | −4.32 | < .001* | −0.42 | [−0.62, −0.22] |
| Visual details | | | | |
| General facts vs. autobiographical facts | −5.14 | < .001* | −0.50 | [−0.70, −0.30] |
| General facts vs. repeated events | −6.56 | < .001* | −0.64 | [−0.84, −0.43] |
| General facts vs. unique events | −8.98 | < .001* | −0.87 | [−1.09, −0.64] |
| Autobiographical facts vs. repeated events | −2.07 | .041* | −0.20 | [−0.39, −0.01] |
| Autobiographical facts vs. unique events | −5.46 | < .001* | −0.53 | [−0.73, −0.33] |
| Repeated events vs. unique events | −3.45 | < .001* | −0.33 | [−0.53, −0.14] |

Note: *Significant after correction for multiple comparisons.

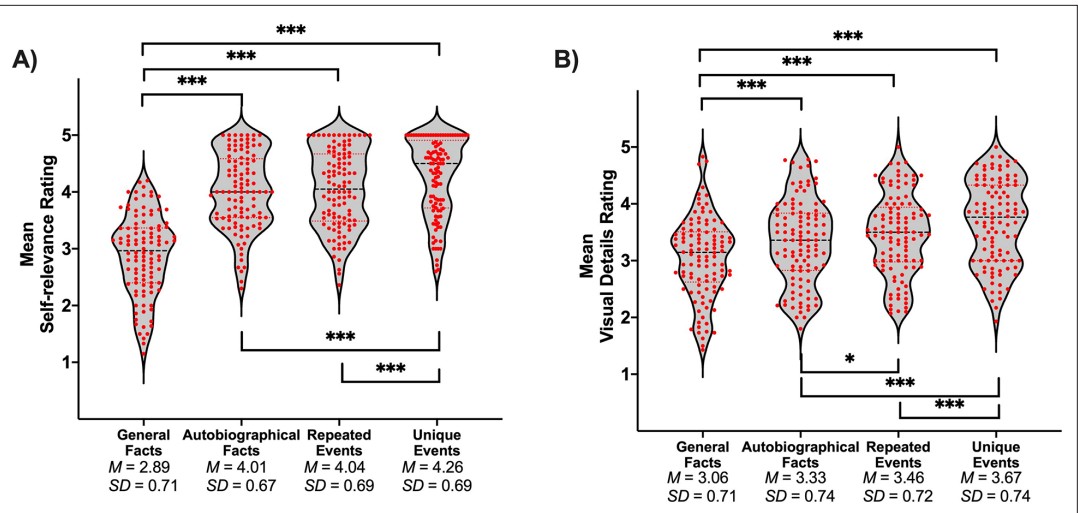

**Figure 1.** Violin plots of (**A**) self-relevance ratings and (**B**) visual details ratings per condition. Red points represent scores of individual participants (*N* = 106). A black line shows the median and red lines show the quartiles. *\*p* < .05, \*\*\**p* < .001.

## Results

### Behavioral study

A sample of 106 participants rated their subjective experience during memory retrieval, using the same paradigm as the fMRI study, but were not scanned. Repeated-measures ANOVA showed that the main effect of memory was significant for all dependent variables (*ps* < .001). Self-relevance was lowest for general facts compared to all personal forms of memory, and lower for personal semantics (i.e., autobiographical facts and repeated events) than unique events (see *Table 1* and *Figure 1A*). Additionally, the amount of visual detail increased from general facts to autobiographical facts to repeated events to unique events (see *Table 1* and *Figure 1B*).

The conjunction of details within a spatial context, or a scene, is sometimes perceived as integral to the recollection of events (*Hassabis and Maguire, 2009*). From that perspective, visual details and scene imagery can be dissociable constructs. A smaller proportion of general facts evoked a scene compared to the three personal memory types (see *Table 2* and *Figure 2C*). Similarly, a smaller proportion of autobiographical facts elicited images of a scene compared to repeated events and unique events. Repeated events and unique events did not differ in the proportion of memories perceived as scenes (see Appendix 1 for the statistical tests of vague/nothing and object responses).

**Table 2.** *t*-test values for pairwise comparisons of memory types on the proportion of scenes.

| Comparison | *t*(105) | *p* | *g* | CI 95% of *g* |
|---|---|---|---|---|
| General facts vs. autobiographical facts | −2.34 | .021* | −0.23 | [−0.42, −0.03] |
| General facts vs. repeated events | −6.51 | < .001* | −0.63 | [−0.84, −0.42] |
| General facts vs. unique events | −7.98 | < .001* | −0.77 | [−0.99, −0.56] |
| Autobiographical facts vs. repeated events | −4.75 | < .001* | −0.46 | [−0.66, −0.26] |
| Autobiographical facts vs. unique events | −6.52 | < .001* | −0.63 | [−0.84, −0.42] |
| Repeated events vs. unique events | −1.28 | .204 | −0.12 | [−0.31, 0.07] |

Note: *Significant after correction for multiple comparisons.

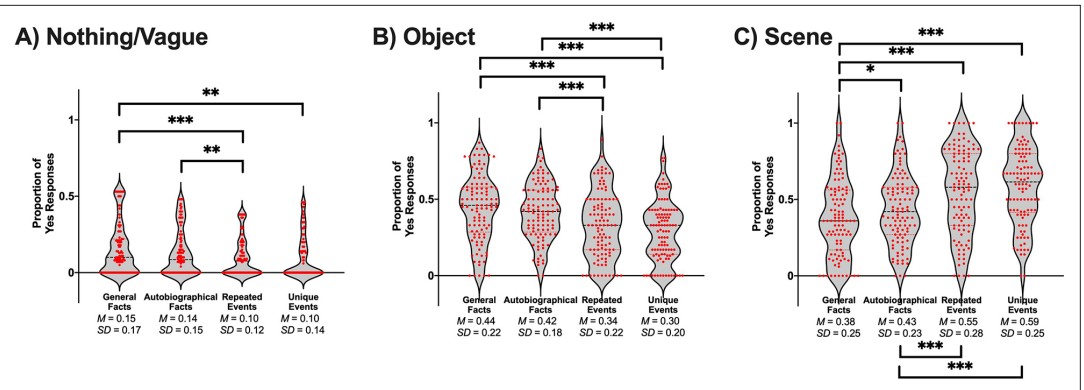

**Figure 2.** Violin plots of the proportion of yes responses that were perceived as (**A**) nothing/vague, (**B**) an object, (**C**) a scene. Red points represent scores of individual participants (*N* = 106). A black line shows the median and red lines show the quartiles. *$p < .05$, **$p < .01$, ***$p < .001$.

## fMRI study

### Non-rotated PLS with a priori contrast

We used a non-rotated PLS to test two theoretically plausible relations between the four memory conditions: a linear contrast (−3, −1, 1, 3) and one comparing general/autobiographical facts and repeated/unique events (−1, −1, 1, 1; abbreviated to facts vs. events subsequently). A linear contrast would be consistent with the continuum perspective of personal semantics (see Box 3 in ***Renoult et al., 2012***), which would predict an increase in activity from general facts to autobiographical facts, from autobiographical facts to repeated events, and from repeated events to unique events. The increase in visual details (described above) followed precisely this pattern. Similarly, personal relevance increased from general facts to personal semantics (autobiographical facts and repeated events) to unique events, suggesting similar dynamics between component processes (e.g., contextual specificity may increase along with personal relevance). The facts vs. events contrast would favor the view of personal semantics as a subtype of general semantics (see Box 2 in ***Renoult et al., 2012***). Of all personal semantics, autobiographical facts correspond best with this view due to its abstraction from events, its more objective quality than other forms of personal knowledge (e.g., trait knowledge), and the feeling of 'knowing' the 'facts' rather than recollecting events (i.e., 'noetic' consciousness; ***Tulving, 2002***). Repeated events would instead group with unique events as 'event memory' due to the common construction of a scene (***Rubin and Umanath, 2015***). Indeed, participants in the behavioral study perceived scenes as frequently for repeated and unique events. However, it is less clear how one would accommodate the greater number of scenes evoked for autobiographical than general facts in a way that aligns with that perspective. Thus, the conjunction of the visualization of scenes, amount of visual details, and personal relevance agrees most with a continuum of contextual specificity.

The linear contrast and facts vs. events contrast were significant (*ps* < .001), explaining 48.78% and 51.22% of the cross-block covariance respectively. Brain scores guide the interpretation of PLS results. A participant's brain score for each task is derived from the multiplication of a voxel's BOLD signal with how much it contributes to the latent variable (LV; i.e., its salience); the values from all voxels at all lags are then all summed together (see ***Figures 3a and 4a***; ***Krishnan et al., 2011***). Brain scores indicate how well each participant expressed the brain and task relation represented by the LV for each of the memory conditions. The temporal 'brain scores' show that the conditions were maximally dissociated at the seventh lag, that is the seventh brain volume after the first acquisition or 8.4 s post cue onset, during the response screen (see ***Figures 3c and 4c*** and ***Appendix 4—figure 1***). The maximal dissociation suggests this lag characterizes best the LV (see Appendices 2 and 3 for information on additional lags).

The linear contrast showed that activity increased from general to autobiographical facts, from autobiographical facts to repeated events, and from repeated to unique events bilaterally in the large regions of the frontal cortex (frontal pole, paracingulate gyrus, frontal medial cortex), precuneus, posterior cingulate cortex, retrosplenial cortex, angular gyrus, and with activity of the right middle

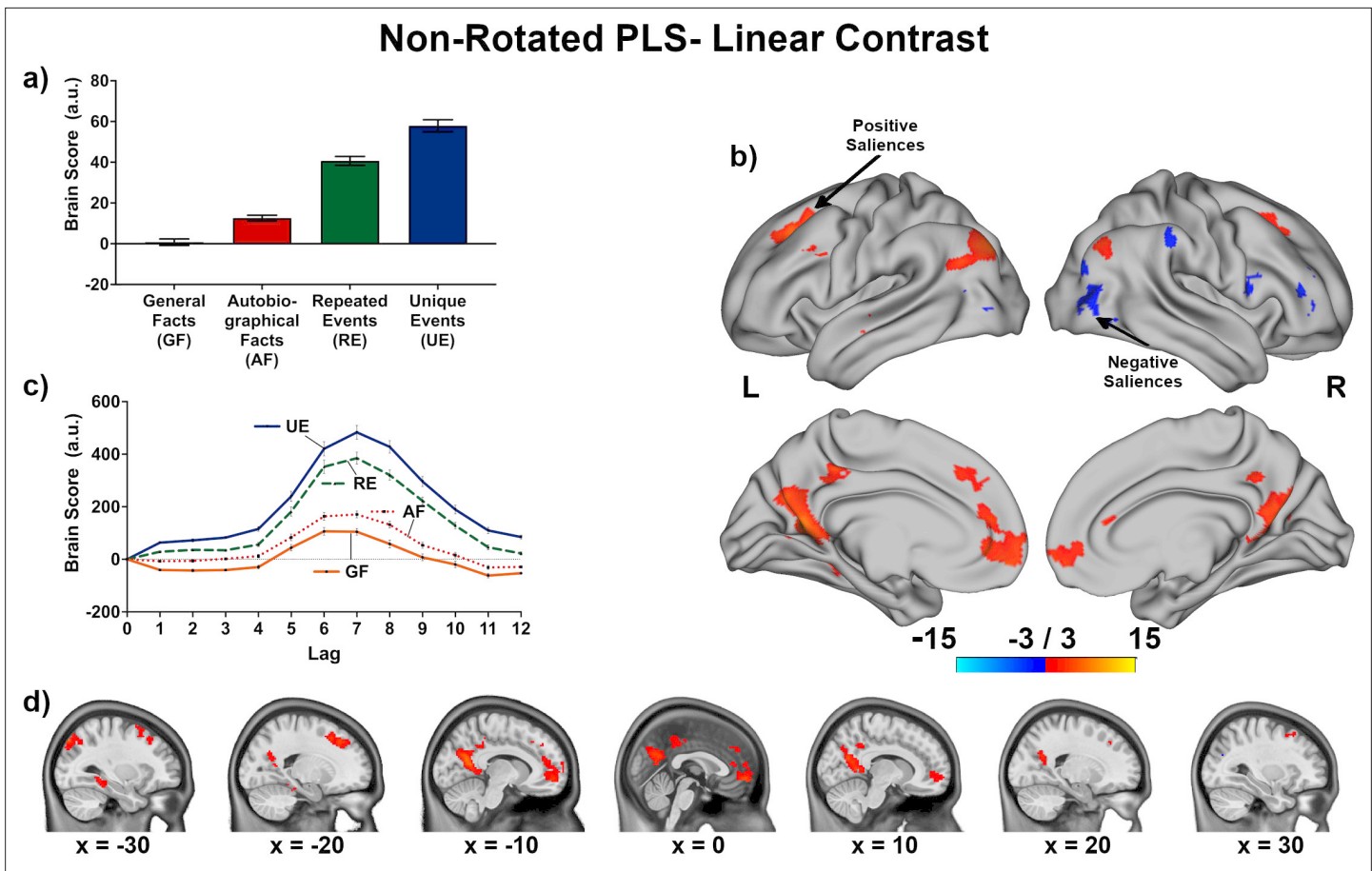

**Figure 3.** This non-rotated partial least squares (PLS) tested a linear contrast (−3, −1, 1, 3). This latent variable (LV) shows regions associated with an increase in activity from general facts to autobiographical facts to repeated events to unique events. (**a**) Average brain score. Error bars are ±1 *SE* of bootstrap estimates. (**c**) Brain scores shown at each lag (i.e., each TR/1.2 s). Error bars are ±1 *SE*. (**b** and **d**) Brain scores with positive saliences shown in warm colors (increased activity from general facts to autobiographical facts to repeated events to unique events) and negative saliences shown in cold colors (decreased activity from general facts to autobiographical facts to repeated events to unique events). Brain scores are projected onto a surface from the Human Connectome Project (S1200; *Van Essen et al., 2012*) using Connectome Workbench (*Marcus et al., 2011*) in (**b**) and the MNI152NLin2009cAsym volume using FSLeyes (*McCarthy, 2021*), in (**d**). Bootstrap ratios are thresholded at ± 3, *p* < .001, cluster size ≥ 80 voxels. See *Appendix 2—figure 1* for additional lags.

frontal gyrus, left parahippocampal gyrus, left hippocampus, and left middle and superior temporal gyrus (see *Figures 3 and 5*, *Table 3*, and Appendix 2 for additional lags). Activity progressively decreased from general to autobiographical facts, from autobiographical facts to repeated events, and from repeated to unique events in areas of the right frontal inferior gyrus, superior parietal lobule, supramarginal gyrus, and bilateral lateral occipital cortex. In supplementary analyses, the brain scores of each memory condition differed from one another (see Appendix 2).

The facts vs. events contrast was similar to the linear contrast (see *Figures 4 and 5*, *Table 4*, and Appendix 3 for additional lags) with the first set of regions (i.e., frontal pole, precuneus, etc.) being associated with greater activity for events than facts, and the second set of regions (i.e., right frontal inferior gyrus, etc.) being associated with reduced activity for events than facts. In supplementary analyses, the statistical comparison of brain scores showed that general and autobiographical facts did not differ in how well they expressed the 'facts' portion of the LV, and repeated and unique events did not differ in how well they expressed the 'events' portion of the LV (see Appendix 3).

## Discussion

Recent progress in cognitive neuroscience has reignited interest in the classic semantic-episodic distinction (*Renoult et al., 2019*) and in memory types that may fall between or cut across it (*Grilli*

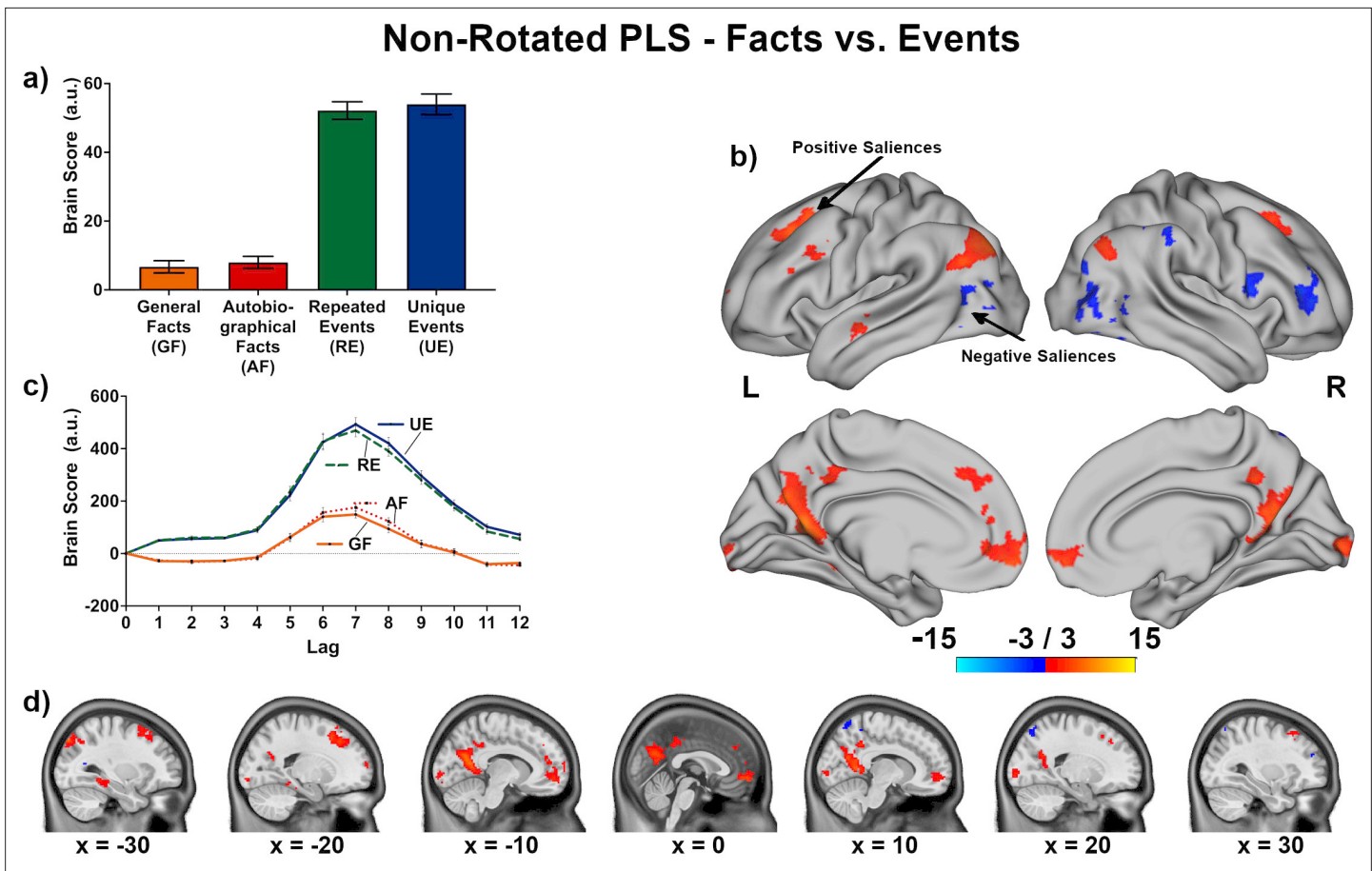

**Figure 4.** This non-rotated partial least squares (PLS) tested a contrast comparing general/autobiographical facts and repeated/unique events (−1, −1, 1, 1). (**a**) Average brain score. Error bars are ±1 *SE* of bootstrap estimates. (**c**) Brain scores shown at each lag (i.e., each TR/1.2 s). Error bars are ±1 *SE*. (**b** and **d**) Brain scores with positive saliences shown in warm colors (increased activity for repeated/unique events relative to general/autobiographical) and negative saliences shown in cold colors (reduced activity for repeated/ unique events relative to general/autobiographical). Brain scores are projected onto a surface from the Human Connectome Project (S1200; *Van Essen et al., 2012*) using Connectome Workbench (*Marcus et al., 2011*) in (**b**) and the MNI152Nlin2009cAsym volume using FSLeyes (*McCarthy, 2021*) in (**d**). Bootstrap ratios are thresholded at ± 3, *p* < .001, cluster size ≥ 80 voxels. See *Appendix 3—figure 1* for additional lags.

*and Verfaellie, 2014*; *Renoult et al., 2012*). In this study, we directly compared four types of memory: one prototypical of semantic memory, one prototypical of episodic memory, and two intermediate (i.e., two types of personal semantics). We closely matched the four memory conditions: We varied the stimuli only in self-relevance (i.e., not self-relevant for general facts and self-relevant for other conditions) and temporal specificity (atemporal/general for both types of facts, more specific for repeated events, and very specific for unique events), in keeping with the distinctions across these four memory types. The general facts condition was thus operationalized as concerning knowledge of people in general (versus the self for the other conditions) and what they generally do (versus what they do at specific times). In that respect, the general facts condition takes the typical operationalization of general semantic memory as reflecting general knowledge of the world. In future investigations, one could consider comparing knowledge of the self to knowledge of specific individuals, which would allow one to evaluate a more specific aspect of semantic memory, but may be associated with other challenges (e.g., either systematically comparing the self to a specific individual, limiting generalizability, or thinking about a different individual in different trials, which would add a task-switching element). All conditions peaked at around the same time in key fMRI analyses.

Personal semantic, general semantic, and episodic memory had both shared and unique neural correlates. The shared neural correlates were revealed when contrasting the four memory conditions with a control condition (see Appendix 4). Activity in several regions of the 'core memory network'

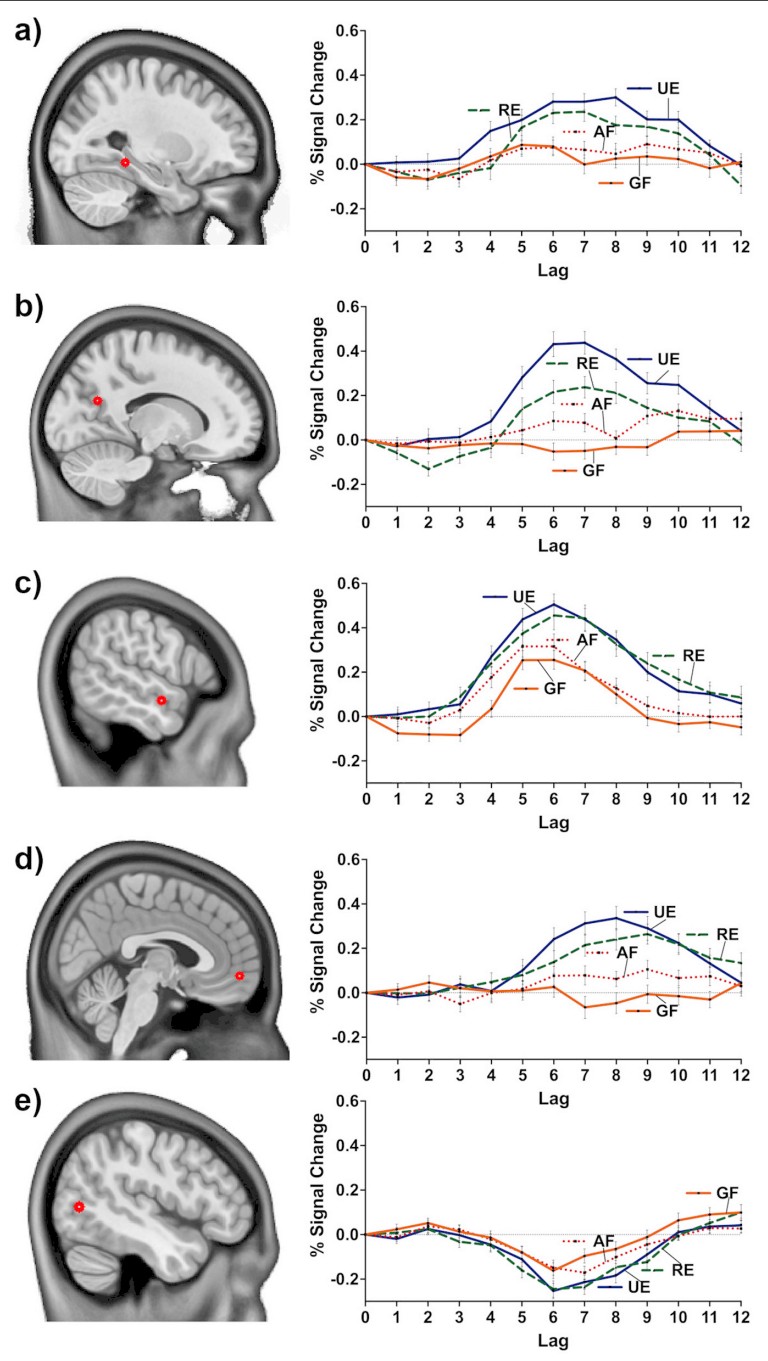

**Figure 5.** Percent BOLD signal change at MNI coordinates: (**a**) –27.5 –40 –10 (posterior division of the left parahippocampal gyrus), (**b**) –15.0 –62.5 22.5 (left precuneus cortex), (**c**) –57.5 –5 –12.5 (anterior division of the left middle temporal gyrus), (**d**) 2.5 55.0 –7.5 (right frontal pole), (**e**) –47.5 –70 5 (inferior division of the left lateral occipital cortex). (**a** and **c**) were common peaks for the two contrasts (linear and facts vs. events). (**b**, **d**, and **e**) were peaks of the linear contrast, but facts vs. events had a peak at nearby location (i.e., –47.5 –70 2.5 to e). GF = general facts, AF = autobiographical facts; RE = repeated events; UE = unique events.

(*Addis et al., 2016*; *Burianova and Grady, 2007*; *Rugg and Vilberg, 2013*; *Svoboda et al., 2006*) dissociated the four memory conditions from the control condition, including in the inferior/middle frontal gyrus, caudate, lingual gyrus, parahippocampal gyrus, and hippocampus bilaterally, and the left middle/superior temporal gyrus.

**Table 3.** Peaks of clusters for the linear contrast at lag 7.

| Bootstrap ratio | Cluster size (voxels) | X (mm) | Y (mm) | Z (mm) | Harvard-Oxford, probability atlas |
|---|---|---|---|---|---|
| Negative saliences | | | | | |
| −5.21 | 362 | 40.0 | −80.0 | 25.0 | 72.0% Right Lateral Occipital Cortex Superior Division |
| −4.76 | 188 | 42.5 | −50.0 | 57.5 | 46.0% Right Superior Parietal Lobule; 20.0% Right Angular Gyrus |
| −4.75 | 103 | −47.5 | −70.0 | 5.0 | 84.0% Left Lateral Occipital Cortex inferior division |
| −4.61 | 102 | 50.0 | 15.0 | 17.5 | 46.0% Right Inferior Frontal Gyrus pars opercularis |
| Positive saliences | | | | | |
| 5.03 | 113 | −5.0 | 30.0 | 37.5 | 63.0% Left Paracingulate Gyrus; 13.0% Left Superior Frontal Gyrus |
| 5.29 | 116 | 25.0 | 20.0 | 42.5 | 22.0% Right Middle Frontal Gyrus; 19.0% Right Superior Frontal Gyrus |
| 6.10 | 94 | −57.5 | −5.0 | −12.5 | 37.0% Left Middle Temporal Gyrus Anterior Division; 23.0% Left Superior Temporal Gyrus Anterior Division; 9.0% Left Superior Temporal Gyrus Posterior Division |
| 6.12 | 105 | −52.5 | 15.0 | 37.5 | 54.0% Left Middle Frontal Gyrus; 5.0% Left Inferior Frontal Gyrus pars opercularis |
| 6.24 | 110 | −27.5 | −40.0 | −10.0 | 29.0% Left Parahippocampal Gyrus Posterior Division; 27.0% Left Lingual Gyrus; 8.0% Left Temporal Occipital Fusiform Cortex; 7.0% Left Temporal Fusiform Cortex Posterior Division |
| 6.25 | 233 | −5.0 | −40.0 | 42.5 | 48.0% Left Cingulate Gyrus Posterior Division; 31.0% Left Precuneus Cortex |
| 6.65 | 137 | 45.0 | −75.0 | 37.5 | 68.0% Right Lateral Occipital Cortex Superior Division |
| 6.85 | 401 | −20.0 | 30.0 | 42.5 | 51.0% Left Superior Frontal Gyrus; 8.0% Left Middle Frontal Gyrus |
| 7.19 | 647 | 2.5 | 55.0 | −7.5 | 43.0% Right Frontal Pole; 28.0% Right Frontal Medial Cortex; 8.0% Right Paracingulate Gyrus |
| 8.87 | 444 | −30.0 | −82.5 | 45.0 | 44.0% Left Lateral Occipital Cortex Superior Division |
| 10.36 | 1118 | −15.0 | −62.5 | 22.5 | 54.0% Left Precuneus Cortex; 12.0% Left Supracalcarine Cortex |

Note: Thresholded at bootstrap ratio at ± 3 (*p* < .001), minimal cluster size of 80 voxels, and a minimal distance of 10 voxels. Labels from the Harvard-Oxford (*Desikan et al., 2006*) atlas obtained using AtlasReader (*Notter et al., 2019*).

In contrast, the unique neural correlates were evident when examining the four memory conditions on their own. The non-rotated PLS converges with the data-driven PLS (see Appendix 4) to suggest the facts vs. events contrast dominates to explain the spatiotemporal relations across memory conditions, although the difference in covariance explained between the two LVs of the non-rotated PLS was slight. In fact, both a priori contrasts captured aspects of the spatiotemporal relations adequately. The percentage of signal change (see *Figure 5*) illustrates the complementarity of the two perspectives to encapsulate the relation between memory conditions. That is, activity increased (or decreased) continuously across memory types, but the extent of the increase (or decrease) confers greater similarity between general and autobiographical facts, and between repeated and unique events. Thus, several regions showed a relatively small increase in activity from general facts to autobiographical facts, a relatively large increase from autobiographical facts to repeated events, and a relatively small increase from repeated events to unique events; these include the precuneus, posterior cingulate, angular gyrus and middle frontal gyrus bilaterally, and left parahippocampal gyrus, left hippocampus, and left middle/superior temporal gyrus (see *Appendix 2—figure 1* and *Appendix 3—figure 1*). [We obtained comparable non-rotated PLS results while including only voxels within the default mode and medial temporal networks from *Barnett et al., 2021*: 49.56% crossblock covariance (*p* < .001) for the linear contrast, and 50.44% crossblock covariance (*p* < .001) for the facts vs. events contrast. Within the selected networks, the same regions contributed to dissociate the memory conditions, and temporal brain scores peaked at lag 7. This supplementary analysis reinforces the importance of regions within the core memory network to determine the relation between memory conditions (see *Appendix 2—figure 1* and *Appendix 3—figure 1*), even though we found additional contributors at the whole brain level.] Activity instead decreased in a commensurate manner predominantly in the

**Table 4.** Peaks of clusters for the facts vs. events contrast at lag 7.

| Bootstrap ratio | Cluster size (voxels) | X (mm) | Y (mm) | Z (mm) | Harvard-Oxford, probability atlas |
|---|---|---|---|---|---|
| Negative saliences | | | | | |
| −5.94 | 251 | 42.5 | 45 | 5 | 85.0% Right Frontal Pole |
| −5.59 | 170 | 57.5 | 12.5 | 15 | 55.0% Right Inferior Frontal Gyrus pars opercularis; 23.0% Right Precentral Gyrus |
| −5.30 | 465 | 57.5 | −62.5 | −10 | 52.0% Right Lateral Occipital Cortex Inferior Division; 15.0% Right Inferior Temporal Gyrus temporooccipital part; 9.0% Right Middle Temporal Gyrus temporooccipital part |
| −5.17 | 100 | 67.5 | −42.5 | 22.5 | 19.0% Right Supramarginal Gyrus Posterior Division |
| −5.05 | 153 | 20 | −70 | 55 | 58.0% Right Lateral Occipital Cortex Superior Division |
| −5.05 | 300 | 60 | −35 | 45 | NA |
| −4.70 | 202 | −47.5 | −70 | 2.5 | 87.0% Left Lateral Occipital Cortex Inferior Division |
| Positive saliences | | | | | |
| 4.32 | 90 | −5 | 30 | 37.5 | 63.0% Left Paracingulate Gyrus; 13.0% Left Superior Frontal Gyrus |
| 4.87 | 101 | −10 | −95 | 0 | 65.0% Left Occipital Pole |
| 5.85 | 155 | 25 | 22.5 | 42.5 | 29.0% Right Superior Frontal Gyrus; 23.0% Right Middle Frontal Gyrus |
| 5.86 | 87 | 17.5 | −95 | 0 | 50.0% Right Occipital Pole |
| 6.21 | 153 | −52.5 | 15 | 37.5 | 54.0% Left Middle Frontal Gyrus; 5.0% Left Inferior Frontal Gyrus pars opercularis |
| 6.40 | 115 | −57.5 | −5 | −12.5 | 37.0% Left Middle Temporal Gyrus Anterior Division; 23.0% Left Superior Temporal Gyrus Anterior Division; 9.0% Left Superior Temporal Gyrus Posterior Division |
| 6.65 | 115 | −27.5 | −40 | −10 | 29.0% Left Parahippocampal Gyrus Posterior Division; 27.0% Left Lingual Gyrus; 8.0% Left Temporal Occipital Fusiform Cortex; 7.0% Left Temporal Fusiform Cortex Posterior Division |
| 6.88 | 487 | 0 | 55 | −5 | 17.0% Right Frontal Pole; 12.0% Left Frontal Pole; 9.0% Right Paracingulate Gyrus; 9.0% Right Frontal Medial Cortex; 6.0% Left Paracingulate Gyrus; 6.0% Left Frontal Medial Cortex |
| 7.08 | 421 | −20 | 17.5 | 47.5 | 37.0% Left Superior Frontal Gyrus; 6.0% Left Middle Frontal Gyrus |
| 7.09 | 163 | 45 | −75 | 37.5 | 68.0% Right Lateral Occipital Cortex Superior Division |
| 9.43 | 470 | −32.5 | −82.5 | 42.5 | 65.0% Left Lateral Occipital Cortex Superior Division |
| 9.55 | 1394 | −7.5 | −57.5 | 12.5 | 51.0% Left Precuneus Cortex; 7.0% Left Intracalcarine Cortex; 5.0% Left Supracalcarine Cortex |

Note: Thresholded at bootstrap ratio at ± 3 ($p < .001$), minimal cluster size of 80 voxels, and a minimal distance of 10 voxels. Labels from the Harvard-Oxford (***Desikan et al., 2006***) atlas obtained using AtlasReader (***Notter et al., 2019***).

right hemisphere, particularly the frontal pole, inferior frontal gyrus, and supramarginal cortex. These findings are compatible with a continuum perspective of declarative memory, because quantitative rather than qualitative variations in brain activity suffice to characterize the relation between these memory types (***Renoult et al., 2012***).

The different types of cues used in our experiment were used to trigger different 'retrieval modes' in our participants, 'a necessary condition for retrieval' (***Tulving, 1983***) that is maintained as a tonic state during a retrieval task (***Rugg and Wilding, 2000***). The behavioral data revealed the sentence cues induced typical phenomenological experience associated with these memory types. Self-relevance was rated lowest for general facts compared to all personal forms of memory, and lower for personal semantics (i.e., autobiographical facts and repeated events) than unique events. Additionally, the amount of visual details increased from general facts to autobiographical facts to repeated events to unique events. This is consistent with previous studies. For example, as compared to unique events, repeated events are generally remembered less vividly: they are associated with reduced temporal specificity, personal significance, emotionality, and amount of details (e.g., ***Addis et al., 2004b***; ***Holland et al., 2011***; ***Levine et al., 2004***). Lastly, the four memory conditions differed in the proportion of scenes that came to mind during retrieval. A smaller proportion of general facts and

autobiographical facts were categorized as scenes compared to repeated events and unique events, which did not differ in scene responses. This is consistent with the idea that both memories of unique and repeated events share a spatial organization that gives them their 'basic context', as well as a first person perspective, detailed visual imagery (*Hassabis and Maguire, 2007*; *Nadel, 1991*; *Robin, 2018*; *Robin et al., 2016*; *Rubin, 2022*; *Rubin and Umanath, 2015*). Thus, scenes are thought to be a dominant and integral feature of events (*Hassabis and Maguire, 2007*; *Nadel, 1991*; *Robin, 2018*; *Robin et al., 2016*; *Rubin, 2022*; *Rubin and Umanath, 2015*) in a way that has not been argued for 'facts'. Although context can shape semantic processing (*Yee and Thompson-Schill, 2016*), the representation of general and experience-far autobiographical facts (*Grilli and Verfaellie, 2016*) may rely less on entities within a context or the context itself and more or as much on the conceptual representation of individual entities, such as objects and words, than repeated and unique events. Facts, in particular personal facts, may evoke episodes and scenes along with them, as in autobiographically significant concepts (*Renoult et al., 2015*; *Westmacott et al., 2004*; *Westmacott and Moscovitch, 2003*), for example, a person could automatically recall going to a gift store when trying to decide what is their favorite gift. Nevertheless, 'scenes' would not be an integral component to general/ autobiographical facts in the same way that they would be for repeated/unique events (*Rubin, 2022*; *Rubin et al., 2019*; *Rubin and Umanath, 2015*). In this study, the importance of visuospatial processes (in subjective ratings and corresponding brain regions) was not commensurate with characteristics of the stimuli, which differed little across conditions and primarily increased in temporal specificity (e.g., general facts: 'Most people wear jeans.'; autobiographical facts: 'Sometimes I wear jeans.'; repeated events: 'When at work, I have worn jeans.'; unique events: 'Yesterday, I wore jeans.').

The importance of situational or contextual elements featured strongly in the neuroimaging data as in the behavioral data. The core network, also known as the default mode network, has been subdivided into an anterior temporal network linked to 'entities' (*Ranganath and Ritchey, 2012*; *Reagh and Ranganath, 2018*) or 'conceptual remembering' (*Sheldon et al., 2019*) and a posterior medial network linked to 'situational models' (*Ranganath and Ritchey, 2012*; *Reagh and Ranganath, 2018*) or 'perceptual remembering' (*Sheldon et al., 2019*). Although these networks process different kinds of information, neither is strictly dedicated to semantic or episodic memory (*Reagh and Ranganath, 2018*; *Sheldon et al., 2019*). For instance, knowledge can facilitate the search and construction of events (*Irish and Piguet, 2013*) and semantic memory can integrate contextual information (e.g., *Greenberg et al., 2009*; *Sheldon and Moscovitch, 2012*). Accordingly, in our study the dissociation between general/autobiographical facts and repeated/unique events did not have a clear posterior medial to anterior temporal demarcation (i.e., posterior medial activity for events and anterior temporal activity for facts). Instead, regions primarily within the posterior medial network (i.e., angular gyrus, posterior cingulate gyrus, precuneus, and parahippocampal gyrus; *Ritchey et al., 2015*) dissociated memory conditions, whereas those within the anterior temporal network (i.e., frontal orbital cortex, inferior anterior temporal gyrus, temporal pole, bilateral amygdala, and perirhinal cortex; *Ritchey et al., 2015*) did not. Indeed, events contained greater contextual information, as suggested by the increased proportion of scenes they evoked, as compared to factual memories. Further, cues were more temporally specific for repeated/unique events than general/autobiographical facts. Consistent with this, many regions of the posterior medial network were associated with greater activity for repeated/unique events than general/autobiographical facts (similar to *Ford et al., 2011*; *Levine et al., 2004*; *Maguire and Mummery, 1999*), and showed a linear increase from the most general type of memory (i.e., general facts) to the most specific memory (i.e., unique events). Therefore, activity in regions associated with visuospatial processing (e.g., precuneus) and scenes (e.g., medial temporal regions) coheres with behavioral data to support the prominence of contextual specificity in determining the relation across memory types. Activity in medial frontal regions is in harmony with ratings of visual details and scene perception, likewise increasing along a continuum of contextual specificity (see *Figure 5d*). However, this anterior portion of the medial frontal cortex may correspond best to self-processing rather than 'situational' processing or mental time travel (*Lieberman et al., 2019*). If activity in the medial frontal cortex in our study reflected exclusively self-processing, one would expect a greater proximity between autobiographical facts and repeated events on the basis of subjective ratings of self-relevance. The additional concordance of medial frontal cortex with a continuum of contextual specificity could be a corollary of the strong links between aspects of self-relevance and episodic simulation (*Grysman*

*et al., 2013*; *King et al., 2022*; *Tulving, 2002*; *Verfaellie et al., 2019*), in addition to this region's role in modulating recollection (*McCormick et al., 2020*), for example through engaging schema-related information (*Gilboa and Marlatte, 2017*).

Taken together, our data correspond with a continuum perspective of declarative memory, with the different memory types varying in magnitude of activation within a common network of brain regions. What underlies this overlap in the neural substrates of semantic and episodic memory? A parsimonious explanation is that semantic and episodic memory rely on similar elementary component processes (*Cabeza and Moscovitch, 2013*; *Larsen, 1992*; *Moscovitch, 1992*; *Rajah and McIntosh, 2005*; *Renoult et al., 2012*; *Rubin, 2022*). All types of memories would depend on a similar network of brain regions but with different weighting of certain nodes in the network. The identification of the relative contribution of different component processes is a critical next step. *Some* of the characteristics that would influence differences in hippocampal activity and other regions of the core memory network include: the number of details (*Thakral et al., 2020*), their association (*Duff et al., 2019*; *Solomon and Schapiro, 2020*), their integration within a scene (*Nadel, 1991*; *Robin, 2018*; *Robin et al., 2016*; *Rubin and Umanath, 2015*) or within a situational model (*Reagh and Ranganath, 2018*; *Summerfield et al., 2010*), their coarseness and precision (*Craik, 2020*; *Ekstrom and Yonelinas, 2020*), their type and modality (e.g., perceptual, spatial, temporal, social; *Binder and Desai, 2011*; *Grilli and Verfaellie, 2016*; *Sheldon et al., 2019*), their stability (*Auger and Maguire, 2018*), as well as their projection into a temporally distant time (*Andrews-Hanna et al., 2010*), the open-endedness of the representation (*Sheldon and Moscovitch, 2012*), the demands on pattern separation to construct unique representations or identify distinguishing features (*Rolls and Kesner, 2006*; *Schapiro et al., 2017*), and the likelihood of eliciting a specific event (*Renoult et al., 2015*; *Westmacott et al., 2004*; *Westmacott and Moscovitch, 2003*). For instance, episodic memory would typically rely to a greater degree than semantic memory on rich sensory-perceptual imagery, complex situational models or scenes, spatial and temporal features, and self-reflection. Accordingly, instead of activating different networks of brain regions, semantic and episodic processes may activate a similar network but with different degrees of magnitude, or recruit these brain regions in a complementary manner (*Sherman et al., 2023*). How each component is involved would also depend on the task at hand (e.g., *Gurguryan and Sheldon, 2019*); each component could be more or less engaged regardless of the memory type (e.g., semantic details can be thought of in rich detail, in relation to the self, or in relation to a spatial context). Therefore, there would be a 'neural-psychological representation correspondence' (*Moscovitch and Gilboa, 2022*) that includes elements of consciousness (e.g., feeling of being transported in time; *Tulving, 2002*) and that transcends categories of memory. Our neuroimaging data, and complementary behavioral data obtained outside the scanner, are compatible with this component process view and inconsistent with strictly separated memory systems. The operationalization of different types of personal semantics and their inclusion in a model of declarative memory does not promote fragmentation. Rather, a taxonomy of personal semantics offers an opportunity to explore what brings all forms of memory together and what can sometimes pull them apart.

## Materials and methods
### fMRI study participants

Fifty-three recruited participants (35 women, 18 men) were aged 24.89 years on average (*SD* = 4.51; range: 19–34) and attained a mean of 16.43 years of education (*SD* = 2.42). This sample size was the largest that was possible to achieve. The sample size is larger than similar studies (e.g., *N* ~12–28; *Addis et al., 2011*; *Burianova and Grady, 2007*; *Ford et al., 2011*; *Holland et al., 2011*). No formal power analysis was conducted. Participants responded to ads on university campuses and on social media. Candidates were retained if they were right-handed, native English speakers, free of any contraindication for MRI (e.g., ferromagnetic metal, back pain, claustrophobia), aged between 18 and 35 years, and had not experienced head injury, or neurological and psychiatric disorders. We excluded three participants because of an inadvertent phase encoding switch, one for an incidental finding, and one for ineligibility. Participants received a compensation of 30 CAD. The study received REB approval at the University of Ottawa (H08-16-32) and the Royal Ottawa Mental Health Centre (ROH; 2016023).

## Behavioral study participants

The total sample included 181 participants (143 women, 34 men, 2 non-binary; 2 missing values); they were 18.79 years old on average (*SD* = 1.58; range: 17–34; 2 missing values) and had attained a mean of 12.67 years of education (*SD* = 1.02; 3 missing values). The possible small differences between proximal memory types (e.g., *Addis et al., 2004b*; *Maguire and Mummery, 1999*; *Renoult et al., 2016*) justified the large sample size. We recruited participants from a pool of students who received a credit toward their introductory or research methods course. The ad specified the eligibility criteria, which were the same as the fMRI study apart from those for MRI safety. We excluded participants who disclosed information that conflicted with the criteria (*n* for primary reason listed): 18–35 years of age (*n* = 1), right-handed (*n* = 2), normal or corrected-to-normal vision (*n* = 1), no neurological or psychiatric disorder (past/present) or no loss of consciousness lasting more than 10 min (*n* = 26), and native English speakers (*n* = 35). We also excluded 1 participant for being inattentive during testing (i.e., looking at a cellphone during the task), and 9 for implausible data (i.e., always responding 'yes') or incomplete data (e.g., entire task or questionnaire). After exclusions we had a total of 106 participants (87 women, 19 men) with a mean age of 18.74 (*SD* = 1.05, range 18–23) and 12.67 years of education on average (*SD* = 0.97). The study received REB approval at the University of Ottawa (H08-16-32).

## fMRI task and procedure

All tasks described below were administered via E-Prime 2.0 (*Psychology Software Tools, 2012*). We optimized the sentence verification task used in an electrophysiological study (*Foster et al., 2012*; *Renoult et al., 2016*) for fMRI. The four experimental conditions (general facts, autobiographical facts, repeated events, unique events) consisted of the same 70 main clauses (material available in Appendix 5). The conditions differed in two aspects: (1) The tense changed from past tense for unique events, to present perfect for repeated events, to present for facts (general and autobiographical). The type of awareness associated with semantic memory is indeed thought to be centered in the present, whereas episodic recollection is oriented toward the past (*Tulving, 2001*; *Tulving, 2002*). (2) We added distinct cue words that preceded each condition and gave different degrees of temporal specificity. We used 6–7 cues per condition. In the unique events condition, we used specific time cues (last night, last week-end, this morning, this week, today, yesterday) to promote access to specific instances of events (e.g., 'Yesterday, I took a picture'). In the repeated events condition, we used script-like cues (when alone, when at a clinic, when at work, when on the bus, when on vacation, when shopping, when with friends), and constrained their temporal scope by asking participants to verify sentences by thinking about events that happened repeatedly within the last year (e.g., 'When shopping, I have taken a picture'). In the autobiographical facts condition, we used general time cues (everyday, often, rarely, sometimes, usually, very often) for participants to report what is usual for them ('Very often, I take pictures'). For the general facts condition, the first person personal pronoun (I) and the 6 cues were replaced by six distinct third person perspectives (everyone, few people, many people, most people, no one, some people), and participants had to report what they thought was generally true for people in their country ('Few people take pictures'). The task involved pressing one of two buttons to produce a 'yes' or 'no' response to indicate whether the statements were true.

The cue (e.g., last weekend) was presented for ~2 s, followed with the main clause (e.g., I went to the pharmacy) for ~4 s. Participants responded after each statement during a response screen that lasted ~3.5 s and which displayed the response options. The separate response screen minimized the contribution of response-related motor activity during the experimental tasks (see Appendix 5 for the analysis of response time). Participants completed an odd/even task during a jittered interval that lasted from 0 to 12 s (mean of 4.6 s; adapted from *Madore et al., 2016*). Participants indicated whether a digit was even using the same response options and buttons as the main task (i.e., yes, no). Each digit was shown in letters for 2 s; thus, participants performed the task for up to six digits (see *Figure 6*). An odd/even task is frequently used in autobiographical memory and future thinking research (*Parlar et al., 2018*; *Svoboda and Levine, 2009*; *Thakral et al., 2020*) to reveal the core memory network (*Stark and Squire, 2001*). We obtained duration for the intervals from Optseq2, with some adaptations (*Dale, 1999*).

We blocked the trials of a given memory condition together within each run. We selected a mixed design to minimize switch costs and to repeat the instructions briefly prior to the series of trials. We asked participants to think about what is usually true for people in general in this country (for general

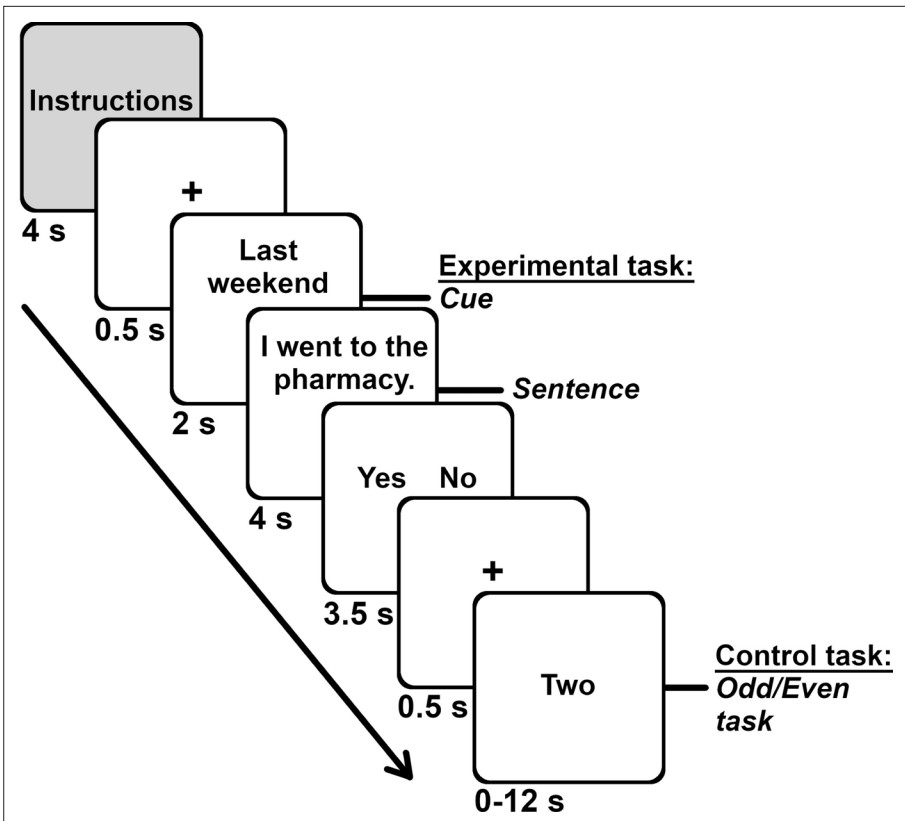

**Figure 6.** Task structure and durations. Each of the seven runs included the four memory conditions (general facts, autobiographical facts, repeated events, unique events). Each memory block started with instructions, followed with the 10 trials for that memory condition. Each trial unfolded in this order: fixation cross (0.5 s), cue (2 s), sentence (4 s), response screen (3.5 s). We presented the odd/even task (range 0–12 s, *M* = 4.6 s, 1 number per 2 s) during the interstimulus interval.

facts) or usually true for oneself (for autobiographical facts), and whether events happened repeatedly within the last year (for repeated events) or at a specific time within the last week (for unique events). Each of the seven runs included the four memory conditions, each comprising of 10 trials. A run lasted 10.6 min. We attributed the stimuli to a run to maximize the likelihood of obtaining a 'yes' response for each memory condition within each run. The likelihood was determined based on the frequency of yes responses per item in *Renoult et al., 2016*, and pilot data. The aim of the piloting was to obtain a comparable number of yes and no responses (in other words approximately 35 in each condition; see Appendix 5 for the description and analysis of the proportion of yes responses). The attribution of stimuli to a run was fixed. All participants received the same randomized order of the memory conditions, but a different randomized order of stimuli within each memory block. The stimuli were displayed in Arial in size 30, in white color over a black background. The task was presented visually on a mirror mounted over the head coil, and responses were made by pressing one of two buttons (for yes or no) on an MR safe response box.

All data were acquired within a single session, except for one participant who completed the study in two sessions because of a scanner issue. Participants practiced with a short version of the task outside and inside the scanner. In a post-scan interview, participants described briefly what they were thinking about during a few trials. Scanner time was 90 min. For the other steps, starting with consent and ending with debriefing, the study required an additional 60 min approximately.

## Behavioral study task

We tested participants in groups of 1–4. They completed individually the sentence verification task (described above), but we replaced the control task with subjective ratings, and participants completed only two runs (i.e., 20 stimuli per memory condition). The stimuli were randomized such

that each participants received 20 randomly selected stimuli per condition and all stimuli should be represented equally across all participants. After their yes/no response, participants performed a scene rating, details rating, and a self-relevance rating (no time limit). The order of the ratings was counterbalanced across participants. Participants indicated whether what they pictured in their mind was: (1) nothing/vague, (2) objects only, or (3) a whole scene (based on *Palombo et al., 2018*). Participants rated the amount of visual details that came to mind from 1 (none) to 5 (a lot; based on *D'Argembeau and Van der Linden, 2006*). We provided examples of visual details and emphasized the distinction between scenes and details. ('Note that you could remember a lot of bits and pieces about each individual object or people without putting them into an integrated whole'.) We specified that participants should consider both the amount of details of individual objects or people and the overall number of objects and people. Participants also rated how closely their thoughts were related to themselves from 1 (very remote) to 5 (very close). We explained that 'very close' meant that the thoughts were directly self-related, 'very remote' signified those thoughts concerned strangers or people in general. Intermediate ratings applied to acquaintances. After the practice, the researcher monitored the participants' attentiveness from a control room. After task completion, participants filled the demographic questionnaire and the Center for Epidemiologic Studies Depression Scale (*Radloff, 1977*). The study lasted 60 min.

## fMRI image acquisition

We acquired the images on a Siemens Biograph mMR (Siemens Healthineers, Erlangen, Germany; *Delso et al., 2011*) with a 32-channel head coil (Ceresensa, London, Canada) at the Brain Imaging Centre (BIC) of the ROH. We collected the functional data with a multiband accelerated EPI sequence from CMRR that is sensitive to BOLD (*Moeller et al., 2010*; *Xu et al., 2013*; aligned to anterior and posterior commissure, TR = 1200 ms, TE = 33, flip angle = 65 degrees, FOV = 200 mm, voxel size = 2.5 mm$^3$, slice thickness = 2.5 mm, 530 volumes per run, multiband factor = 6, covering the whole brain). The phase encoding alternated between anterior to posterior (for the first, third, fifth, seventh run) and posterior to anterior (for the second, fourth, sixth run).

Due to a technical error, the TE was 40 ms instead of the intended 33 ms for the second, fourth, and sixth run of the first 28 participants. For this reason, we opted for the field map-free method for distortion correction that is implemented in FMRIPREP (*Esteban et al., 2018*; *Esteban et al., 2019*) instead of correcting with the reverse phase encoding measurements. Moreover, the encoding phase direction was kept constant for subsequent participants (anterior to posterior) as alternating between phase encoding was originally intended for distortion correction (with reverse phase encoding), which would not be possible for the whole sample. Phase encoding or TE differences cannot explain differences between conditions. The task was divided into seven runs which each included the four memory conditions, and the run order was randomized across all participants.

The parameters for the anatomical scan, a T1-weighted pulse sequence (MPRAGE; *Deichmann et al., 2000*), of the first 28 participants had the following parameters: sagittal orientation, TR = 2530 ms, TE = 3.36 ms, TI = 1100 ms, flip angle = 7 degrees, slice thickness = 1 mm, FOV = 256 mm, voxel size = 1 mm$^3$, with an acceleration factor of 2 using GRAPPA. For the latter group of 20 participants, we acquired a T1-weighted pulse sequence (MEMPRAGE; *van der Kouwe et al., 2008*) to enhance the quality of the T1w: sagittal orientation, TR = 2500 ms, TE 1 = 1.69 ms, TE 2 = 3.55 ms, TE 3 = 5.41, TE 4 = 7.27, flip angle = 10 degrees, slice thickness = 1 mm, FOV = 256 mm, voxel size = 1 mm$^3$, with an acceleration factor of 2 using GRAPPA.

## fMRI analyses
### FMRIPREP pipeline
The MRI data were preprocessed using FMRIPREP (*Esteban et al., 2018*; *Esteban et al., 2019*) . The processing steps are described below verbatim as intended by FMRIPREP authors (https://fmriprep. readthedocs.io/; made available through CCO license).

"Results included in this manuscript come from preprocessing performed using fMRIPprep 1.2.3 (*Esteban et al., 2018*; *Esteban et al., 2019*), which is based on Nipype 1.1.6-dev (*Gorgolewski et al., 2011*; *Gorgolewski et al., 2018*; RRID:SCR_002502). Anatomical Data Preprocessing. The T1-weighted (T1w) image was corrected for intensity non-uniformity using N4BiasFieldCorrection (*Tustison et al., 2010*, ANTs 2.2.0), and used as T1w reference throughout the workflow. The T1w

reference was then skull-stripped using antsBrainExtraction.sh (ANTs 2.2.0), using OASIS as target template. Brain surfaces were reconstructed using recon-all (FreeSurfer 6.0.1, RRID:SCR_001847, *Dale et al., 1999*), and the brain mask estimated previously was refined with a custom variation of the method to reconcile ANTs-derived and FreeSurfer-derived segmentations of the cortical gray matter (GM) of Mindboggle (RRID:SCR_002438, *Klein et al., 2017*). Spatial normalization to the ICBM 152 Nonlinear Asymmetrical template version 2009c (*Fonov et al., 2009*; RRID:SCR_008796) was performed through nonlinear registration with antsRegistration (ANTs 2.2.0, RRID:SCR_004757, *Avants et al., 2008*), using brain-extracted versions of both T1w volume and template. Brain tissue segmentation of cerebrospinal fluid (CSF), white matter (WM), and GM was performed on the brain-extracted T1w using fast (FSL 5.0.9, RRID:SCR_002823, *Zhang et al., 2001*). For each of the seven BOLD runs found per subject (across all tasks and sessions), the following preprocessing was performed. First, a reference volume and its skull-stripped version were generated using a custom methodology of fMRIPrep. A deformation field to correct for susceptibility distortions was estimated based on fMRIPrep's fieldmap-less approach. The deformation field is that resulting from co-registering the BOLD reference to the same-subject T1w reference with its intensity inverted (*Huntenburg, 2014*; *Wang et al., 2017*). Registration is performed with antsRegistration (ANTs 2.2.0), and the process regularized by constraining deformation to be nonzero only along the phase-encoding direction, and modulated with an average fieldmap template (*Treiber et al., 2016*). Based on the estimated susceptibility distortion, an unwarped BOLD reference was calculated for a more accurate co-registration with the anatomical reference. The BOLD reference was then co-registered to the T1w reference using bbregister (FreeSurfer) which implements boundary-based registration (*Greve and Fischl, 2009*). Co-registration was configured with nine degrees of freedom to account for distortions remaining in the BOLD reference. Head-motion parameters with respect to the BOLD reference (transformation matrices, and six corresponding rotation and translation parameters) are estimated before any spatiotemporal filtering using mcflirt (FSL 5.0.9, *Jenkinson et al., 2002*). The BOLD time-series (including slice-timing correction when applied) were resampled onto their original, native space by applying a single, composite transform to correct for head-motion and susceptibility distortions. These resampled BOLD time-series will be referred to as preprocessed BOLD in original space, or just preprocessed BOLD. The BOLD time-series were resampled to MNI152N-Lin2009cAsym standard space, generating a preprocessed BOLD run in MNI152NLin2009cAsym space. First, a reference volume and its skull-stripped version were generated using a custom methodology of fMRIPrep. Several confounding time-series were calculated based on the *preprocessed BOLD*: framewise displacement (FD), DVARS, and three region-wise global signals. FD and DVARS are calculated for each functional run, both using their implementations in *Nipype* (following the definitions by *Power et al., 2014*). The three global signals are extracted within the CSF, the WM, and the whole-brain masks. Additionally, a set of physiological regressors were extracted to allow for component-based noise correction (*CompCor*, *Behzadi et al., 2007*). Principal components are estimated after high-pass filtering the *preprocessed BOLD* time-series (using a discrete cosine filter with 128 s cut-off) for the two *CompCor* variants: temporal (tCompCor) and anatomical (aCompCor). Six tCompCor components are then calculated from the top 5% variable voxels within a mask covering the subcortical regions. This subcortical mask is obtained by heavily eroding the brain mask, which ensures it does not include cortical GM regions. For aCompCor, six components are calculated within the intersection of the aforementioned mask and the union of CSF and WM masks calculated in T1w space, after their projection to the native space of each functional run (using the inverse BOLD-to-T1w transformation). The head-motion estimates calculated in the correction step were also placed within the corresponding confounds file. The BOLD time-series were resampled to surfaces on the following spaces: *fsnative*, *fsaverage*. All resamplings can be performed with *a single interpolation step* by composing all the pertinent transformations (i.e., head-motion transform matrices, susceptibility distortion correction when available, and co-registrations to anatomical and template spaces). Gridded (volumetric) resamplings were performed using antsApplyTransforms (ANTs), configured with Lanczos interpolation to minimize the smoothing effects of other kernels (*Lanczos, 1964*). Non-gridded (surface) resamplings were performed using mri_vol2surf (FreeSurfer). Many internal operations of fMRIPrep use Nilearn 0.4.2 (*Abraham et al., 2014*; RRID:SCR_001362), mostly within the functional processing workflow. For more details of the pipeline, see the section corresponding to workflows in fMRIPrep's documentation."

## FSL

The fMRIPrep output (in MNI152NLin2009cAsym space) was spatially smoothed using a Gaussian kernel of FWHM 5 mm and grand-mean intensity normalized using FEAT (FMRI Expert Analysis Tool) Version 6.00 from FSL (FMRIB's Software Library, https://www.fmrib.ox.ac.uk; *Jenkinson et al., 2012*). We did not apply slice timing correction with fMRIPrep or FSL because of the rapid multi-band acquisition.

## PLS

We used a multivariate approach, PLS correlation (*Krishnan et al., 2011*; *McIntosh et al., 2004a*; *McIntosh and Lobaugh, 2004b*), implemented in the PLSgui/PLScmd toolbox for Matlab (https://www.rotman-baycrest.on.ca/index.php?section=84; *McIntosh et al., 2011*). PLS is a technique developed for chemometrics (*Krishnan et al., 2011*) and widely adopted for neuroimaging use. Strengths of this analysis are that it is well suited for designs with multiple sets of measures and collinearity, and it mitigates power issues encountered in univariate analyses (*O'Toole et al., 2007*). It performs a singular value decomposition of the relation between two data matrices, X (brain) and Y (task design) to identify how tasks and voxels covary maximally together. PLS produces LVs that represent the similarities and differences in covariance patterns between the two matrices (akin to PCA eigenvectors). Saliences indicate the strength of the contribution of tasks and voxels to an LV (as indexed through bootstrap values, described below).

We specified that each trial began at cue onset and comprised 12 TRs (or 14.4 s) to encompass the typical time-window of a BOLD response (*McIntosh et al., 2004a*). We retained only yes responses like *Renoult et al., 2016*, because they presumably reflect access to information consistent with the memory condition. Memory accuracy is often difficult to assess for autobiographical memory (*Cabeza and St Jacques, 2007*), and so was not considered for any of the conditions. We retained trials that had 3–6 numbers for the control task, thus lasting 6–12 s. We assessed the significance of LVs ($p <$ .05) via 1000 permutations (randomizing the labels of conditions without replacement). We tested the stability of each voxel's contribution to the LV via 500 iterations of bootstrap estimation (resampling participants with replacement; *Krishnan et al., 2011*; *McIntosh et al., 2004a*; *McIntosh and Lobaugh, 2004b*). The threshold of ± 3 bootstrap ratio (equivalent to $p <$ .001; as in *Hirshhorn et al., 2012*; *Ziaei et al., 2016*) was used to determine whether a voxel made a reliable contribution to the LV. A reliable voxel contributes to the overall task and brain pattern. The PLS tests the association between the task and all voxels at all TRs in a single analysis, and so does not require correction for multiple comparison (*McIntosh and Lobaugh, 2004b*). An additional threshold of a minimum of 80 voxels with a minimum distance of 10 voxels was used to facilitate the summary of findings in the results section and the tables (for a similar cluster size, see *Bowen et al., 2019*; *Hirshhorn et al., 2012*; *Robin et al., 2015*; *Ziaei et al., 2016*). The bootstrap percentile estimates are unreliable. Thus, we reported the standard error of the bootstrap estimation instead of the confidence intervals along with *t*-tests (see Appendices 2 and 3); the interpretation is similar.

## Statistical analyses of brain scores and behavioral data

We compared the four memory conditions (general facts, autobiographical facts, repeated events, unique events) on brain scores, the proportion of 'yes' responses and response time during the fMRI task, and on mean self-relevance, mean visual details and proportion of trials associated with one of the three scene categories during the behavioral task. We followed up on significant main effects with paired samples *t*-tests, reporting Hedges' *g* as the measure of effect size, and correcting each set of post hoc tests (i.e., six tests) with the Holm-Bonferroni procedure (*Holm, 1979*). We attributed the value equivalent to $z \pm 2.58$ to bring univariate outliers closer to the mean. Scene ratings were excessively skewed (see *Figure 2*), but parametric tests gave similar results than nonparametric and are reported for simplicity. We applied a Greenhouse-Geisser correction (*Greenhouse and Geisser, 1959*) when the sphericity assumption was violated. All these analyses focused on trials associated with 'yes' responses (as in fMRI analyses) and were executed in SPSS v. 28 (*Corp, 2021*).

## Open science statement

The instructions and stimuli are available in Appendix 5. End-stage data and some scripts are available on https://osf.io/ (https://osf.io/py5k6/). Additional information may be requested to the corresponding authors. fMRI standards guided the reporting in this paper (*Poldrack et al., 2008*).

## Acknowledgements

We gratefully acknowledge the participants who generously took part in the study; Sara Trincao-Batra and Talia Chin for help with recruitment, testing or preparing the data for analysis; Melina Langevin and Andrew Wilson for collecting the data for the behavioral analysis; Kirandeep K Dogra for the validation of sample descriptives and analyses in SI; Emily Wang with help preparing tables, Emma Haight for feedback on the manuscript; Drs. Donna Rose Addis, Samuel Fynes-Clinton, Andra Smith for advice, and particularly Dr. Randy McIntosh for advice on many aspects of the PLS analyses. Preparation of this manuscript was supported by grants from Natural Sciences and Engineering Research Council of Canada (NSERC; to PSRD), from the British Academy (grant number SG142524 to LR), funds from the Faculty of Social Sciences of the University of Ottawa (to PSRD), an Ontario Graduate Scholarship (to AFNT). DJP is supported by an NSERC Discovery Grant (RGPIN-2019-04596).

## Additional information

### Funding

| Funder | Grant reference number | Author |
| --- | --- | --- |
| British Academy | SG142524 | Louis Renoult |
| Natural Sciences and Engineering Research Council of Canada | RGPIN-2019-04596 | Daniela J Palombo |
| Natural Sciences and Engineering Research Council of Canada | | Patrick SR Davidson |

The funders had no role in study design, data collection and interpretation, or the decision to submit the work for publication.

### Author contributions

Annick FN Tanguay, Conceptualization, Formal analysis, Investigation, Visualization, Writing - original draft, Project administration; Daniela J Palombo, Conceptualization, Supervision, Validation, Writing - original draft, Writing - review and editing; Brittany Love, Investigation, Writing - review and editing; Rafael Glikstein, Writing - review and editing; Patrick SR Davidson, Conceptualization, Resources, Supervision, Funding acquisition, Writing - original draft, Project administration, Writing - review and editing; Louis Renoult, Conceptualization, Supervision, Funding acquisition, Writing - original draft, Project administration, Writing - review and editing

### Author ORCIDs

Annick FN Tanguay (iD) http://orcid.org/0000-0002-6964-3077
Louis Renoult (iD) http://orcid.org/0000-0001-7861-0552

### Ethics

Human subjects: The study received REB approval at the University of Ottawa (H08-16-32), and the Royal Ottawa Mental Health Centre (ROH; 2016023).

### Decision letter and Author response

Decision letter https://doi.org/10.7554/eLife.83645.sa1
Author response https://doi.org/10.7554/eLife.83645.sa2

# Additional files

## Supplementary files
• MDAR checklist

## Data availability
End-stage data (i.e., PLS BSR nifti files) and some scripts (e.g., for event files) are available on OSF (https://osf.io/py5k6/). The authors did not receive approval from the ethics committee to deposit data in a public repository. Additional information may be requested to the corresponding authors. Deidentified data may be shared (i.e., converted dicom files to nifti without identifiers, defaced T1w, other data with minimal demographic/health information to avoid identification). Researchers will be asked to sign a confidentiality agreement and to describe briefly the purpose to ensure it fits with the general purpose of the study described in the consent form. No formal application will be required. Data may not be analyzed for commercial purposes.

The following dataset was generated:

| Author(s) | Year | Dataset title | Dataset URL | Database and Identifier |
| --- | --- | --- | --- | --- |
| Tanguay A, Palombo D, Love B, Glikstein R, Davidson P, Renoult L | 2022 | Personal Semantic, General Semantic, and Episodic Memory: Shared and Unique Neural Correlates | https://osf.io/py5k6/ | Open Science Framework, py5k6 |

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

# Appendix 1

## Behavioural study

### Scene rating: nothing/vague

The four memory conditions differed in their proportion of memory that elicited nothing or a vague image, $F(2.79, 293.34) = 6.82$, $p < .001$, $\eta_p^2 = 0.06$ (see **Figure 2**). A greater proportion of general facts ($M = 0.16$, $SE = 0.02$) evoked nothing or something vague compared to repeated events ($M = 0.10$, $SE = 0.01$), $t(105) = 3.86$, $p < .001$, $g = 0.37$, CI 95% [0.18, 0.57], and unique events ($M = 0.10$, $SE = 0.01$), $t(105) = 3.14$, $p = .002$, $g = 0.30$, CI 95% [0.11, 0.50]. Similarly, a greater proportion of autobiographical facts ($M = 0.14$, $SE = 0.02$) were associated with nothing or a vague image compared to repeated events, $t(105) = 2.94$, $p = .004$, $g = 0.29$, CI 95% [0.09, 0.48]. No other comparisons were significant after correction: general facts vs. autobiographical facts, $t(105) = 1.47$, $p = .144$, $g = 0.14$, CI 95% [–0.05, 0.33]; autobiographical facts vs. unique events, $t(105) = 2.20$, $p = .030$, $g = 0.21$, CI 95% [0.02, 0.40], and repeated events vs. unique events, $t(105) = –0.24$, $p = .814$, $g = –0.02$, CI 95% [–0.21, 0.17].

### Scene rating: object

The four memory conditions differed in the proportion of memory perceived as an object during retrieval, $F(3, 315) = 17.11$, $p < .001$, $\eta_p^2 = 0.14$ (see **Figure 2**). A greater proportion of general facts ($M = 0.44$, $SE = 0.02$) and autobiographical facts ($M = 0.42$, $SE = 0.02$) were perceived as objects compared to repeated events ($M = 0.34$, $SE = 0.02$) and unique events ($M = 0.30$, $SE = 0.02$): general facts vs. repeated events, $t(105) = 4.17$, $p < .001$, $g = 0.40$, CI 95% [0.21, 0.60]; general facts vs. unique events, $t(105) = 5.94$, $p < .001$, $g = 0.57$, CI 95% [0.37, 0.78]; autobiographical facts vs. repeated events, $t(105) = 3.36$, $p = .001$, $g = 0.33$, CI 95% [0.13, 0.52]; autobiographical facts vs. unique events, $t(105) = 5.53$, $p < .001$, $g = 0.54$, CI 95% [0.33, 0.74]. General facts did not differ from autobiographical facts, $t(105) = 1.22$, $p = .226$, $g = 0.12$, CI 95% [–0.07, 0.31], and repeated events did not differ from unique events, $t(105) = 1.82$, $p = .071$, $g = 0.18$, CI 95% [–0.02, 0.37]. Therefore, facts were more frequently associated with the visualization of isolated objects than events.

# Appendix 2

## Non-rotated PLS with a priori contrasts: linear contrast

### Statistical comparison of brain scores

All memory conditions differed from one another: general facts vs. autobiographical facts, $t(47) = -6.97$, $p < .001$, $g = -1.00$, CI 95% [−1.34, −0.65]; general facts vs. repeated events, $t(47) = -16.99$, $p < .001$, $g = -2.43$, CI 95% [−2.99, −1.86]; general events vs. unique events, $t(47) = -18.19$, $p < .001$, $g = -2.61$, CI 95% [−3.20, −2.01]; autobiographical facts vs. repeated events, $t(47) = -12.35$, $p < .001$, $g = -1.77$, CI 95% [−2.22, −1.31]; autobiographical facts vs. unique events, $t(47) = -16.02$, $p < .001$, $g = -2.29$, CI 95% [−2.83, −1.75]; repeated events vs. unique events, $t(47) = -6.53$, $p < .001$, $g = -0.94$, CI 95% [−1.27, −0.59]. Thus, unique events were more robustly associated with activity as described in the main text than repeated events, and repeated events compared to autobiographical facts, and autobiographical facts compared to general facts.

**Appendix 2—table 1.** Peaks of clusters for the linear contrast at lags 5–6 and 8–9.

| Lag | Bootstrap ratio | Cluster size (voxels) | X (mm) | Y (mm) | Z (mm) | Harvard-Oxford, probability atlas |
|---|---|---|---|---|---|---|
| Negative saliences | | | | | | |
| 6 | −5.92 | 99 | 60.0 | 10.0 | 12.5 | 39.0% Right Precentral Gyrus; 28.0% Right Inferior Frontal Gyrus pars opercularis |
| 6 | −5.47 | 204 | 45.0 | −82.5 | 15.0 | 46.0% Right Lateral Occipital Cortex Superior Division; 18.0% Right Lateral Occipital Cortex inferior division |
| 6 | −5.35 | 279 | 55.0 | −37.5 | 30.0 | 29.0% Right Supramarginal Gyrus Posterior Division; 9.0% Right Parietal Operculum Cortex; 8.0% Right Planum Temporale; 6.0% Right Supramarginal Gyrus Anterior Division |
| 6 | −4.41 | 104 | −55.0 | −37.5 | 25.0 | 46.0% Left Parietal Operculum Cortex; 18.0% Left Planum Temporale; 8.0% Left Supramarginal Gyrus Anterior Division; 6.0% Left Supramarginal Gyrus Posterior Division |
| 8 | −4.65 | 189 | 45.0 | −82.5 | 17.5 | 59.0% Right Lateral Occipital Cortex Superior Division |
| 9 | −5.51 | 387 | 50.0 | −75.0 | 10.0 | 68.0% Right Lateral Occipital Cortex inferior division; 8.0% Right Lateral Occipital Cortex Superior Division |
| 9 | −5.38 | 124 | 32.5 | −47.5 | 60.0 | 55.0% Right Superior Parietal Lobule |
| Positive saliences | | | | | | |
| 5 | 5.22 | 129 | −35.0 | −82.5 | 37.5 | 77.0% Left Lateral Occipital Cortex Superior Division |
| 5 | 5.58 | 208 | −20.0 | 30.0 | 42.5 | 51.0% Left Superior Frontal Gyrus; 8.0% Left Middle Frontal Gyrus |
| 5 | 5.88 | 151 | 17.5 | −52.5 | 7.5 | 56.0% Right Precuneus Cortex; 9.0% Right Cingulate Gyrus Posterior Division |
| 5 | 6.08 | 237 | −12.5 | −60.0 | 20.0 | 49.0% Left Precuneus Cortex; 11.0% Left Supracalcarine Cortex |
| 6 | 6.01 | 167 | −30.0 | −35.0 | −15.0 | 49.0% Left Parahippocampal Gyrus Posterior Division; 29.0% Left Temporal Fusiform Cortex Posterior Division |
| 6 | 6.50 | 312 | −20.0 | 32.5 | 42.5 | 53.0% Left Superior Frontal Gyrus; 10.0% Left Frontal Pole; 6.0% Left Middle Frontal Gyrus |
| 6 | 7.11 | 379 | −5.0 | 47.5 | −2.5 | 70.0% Left Paracingulate Gyrus; 15.0% Left Cingulate Gyrus Anterior Division |
| 6 | 7.18 | 114 | 45.0 | −75.0 | 37.5 | 68.0% Right Lateral Occipital Cortex Superior Division |
| 6 | 8.14 | 234 | −32.5 | −82.5 | 42.5 | 65.0% Left Lateral Occipital Cortex Superior Division |
| 6 | 9.21 | 856 | −15.0 | −62.5 | 22.5 | 54.0% Left Precuneus Cortex; 12.0% Left Supracalcarine Cortex |
| 8 | 5.05 | 152 | 25.0 | 30.0 | 47.5 | 34.0% Right Superior Frontal Gyrus; 22.0% Right Middle Frontal Gyrus |
| 8 | 5.83 | 115 | 45.0 | −75.0 | 37.5 | 68.0% Right Lateral Occipital Cortex Superior Division |
| 8 | 6.40 | 261 | −2.5 | −35.0 | 45.0 | 60.0% Left Cingulate Gyrus Posterior Division; 25.0% Left Precuneus Cortex |

*Appendix 2—table 1 Continued on next page*

*Appendix 2—table 1 Continued*

| Lag | Bootstrap ratio | Cluster size (voxels) | X (mm) | Y (mm) | Z (mm) | Harvard-Oxford, probability atlas |
|---|---|---|---|---|---|---|
| 8 | 6.62 | 91 | −57.5 | −5.0 | −12.5 | 37.0% Left Middle Temporal Gyrus Anterior Division; 23.0% Left Superior Temporal Gyrus Anterior Division; 9.0% Left Superior Temporal Gyrus Posterior Division |
| 8 | 8.25 | 449 | −22.5 | 32.5 | 45.0 | 47.0% Left Superior Frontal Gyrus; 10.0% Left Middle Frontal Gyrus; 10.0% Left Frontal Pole |
| 8 | 9.97 | 367 | −32.5 | −82.5 | 42.5 | 65.0% Left Lateral Occipital Cortex Superior Division |
| 8 | 10.03 | 961 | −2.5 | 50.0 | −7.5 | 43.0% Left Frontal Medial Cortex; 38.0% Left Paracingulate Gyrus |
| 8 | 10.43 | 1096 | −5.0 | −62.5 | 15.0 | 42.0% Left Precuneus Cortex; 16.0% Left Intracalcarine Cortex; 12.0% Left Supracalcarine Cortex |
| 9 | 5.66 | 134 | −10.0 | 25.0 | 42.5 | 11.0% Left Superior Frontal Gyrus; 9.0% Left Paracingulate Gyrus |
| 9 | 6.16 | 196 | −2.5 | −35.0 | 42.5 | 81.0% Left Cingulate Gyrus Posterior Division; 16.0% Left Precuneus Cortex |
| 9 | 7.72 | 910 | −7.5 | 55.0 | 10.0 | 43.0% Left Paracingulate Gyrus; 25.0% Left Frontal Pole; 5.0% Left Superior Frontal Gyrus |
| 9 | 7.98 | 476 | −20.0 | 17.5 | 47.5 | 37.0% Left Superior Frontal Gyrus; 6.0% Left Middle Frontal Gyrus |
| 9 | 8.94 | 862 | −5.0 | −65.0 | 15.0 | 31.0% Left Precuneus Cortex; 23.0% Left Supracalcarine Cortex; 21.0% Left Intracalcarine Cortex |
| 9 | 9.80 | 243 | −32.5 | −82.5 | 42.5 | 65.0% Left Lateral Occipital Cortex Superior Division |

Thresholded at bootstrap ratio at ± 3 (*p* < .001), minimal cluster size of 80 voxels, and a minimal distance of 10 voxels. Labels from the Harvard-Oxford (**Desikan et al., 2006**) atlas obtained using AtlasReader (**Notter et al., 2019**).

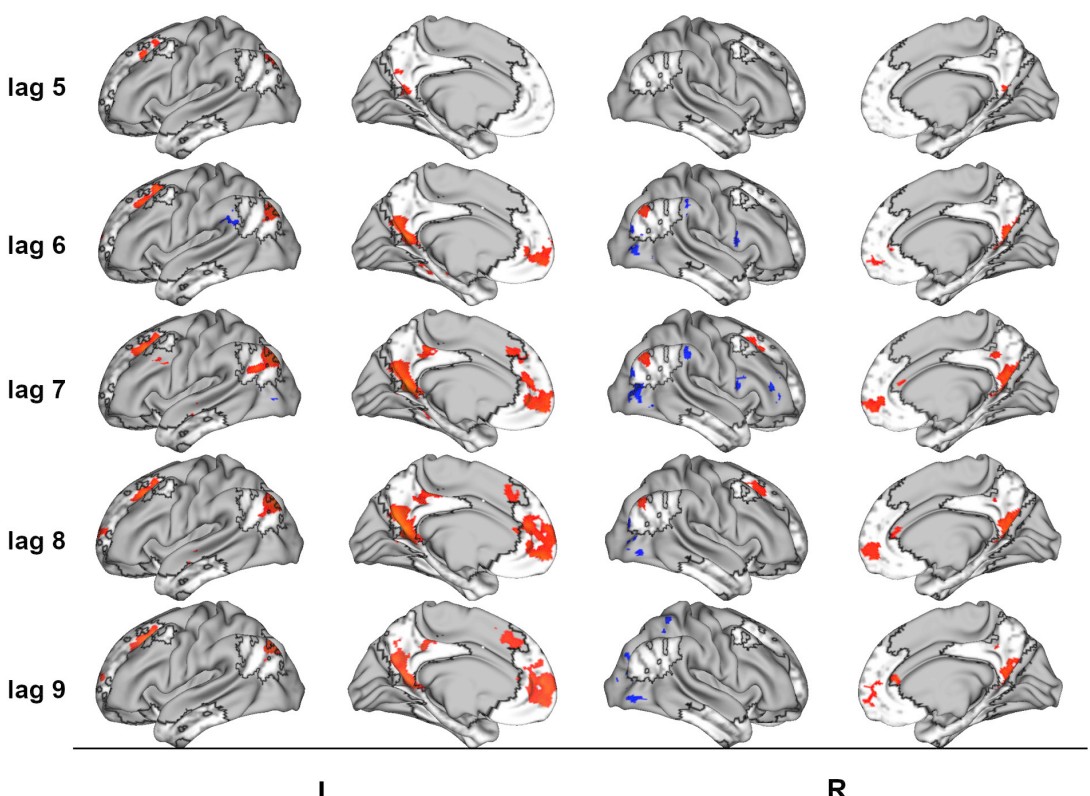

**Appendix 2—figure 1.** Brain scores in orange and blue tones for the latent variable (LV) 1 of the non-rotated partial least squares (PLS) with the linear contrast between memory conditions overlaid on the default mode and medial temporal networks from **Barnett et al., 2021** shown in white with black contour and projected onto a
*Appendix 2—figure 1 continued on next page*

*Appendix 2—figure 1 continued*
surface template from the Human Connectome Project (*Van Essen et al., 2012*) using Connectome Workbench (*Marcus et al., 2011*).

# Appendix 3

## Non-rotated PLS with a priori contrasts: facts vs. events

### Statistical comparison of brain scores

The brain scores of general and autobiographical facts did not differ from one another, $t(47) =$ –0.72, $p$ = .477, $g$ = –0.10, CI 95% [–0.38, 0.18], and those of repeated and unique events did not differ from one another, $t(47) =$ –0.65, $p$ = .517, $g$ = –0.09, CI 95% [–0.37, 0.19]. Hence, general and autobiographical facts did not differ in how well they expressed the 'facts' portion of the LV, and repeated and unique events did not differ in how well they expressed the 'events' portion of the LV. All other comparisons were significant: general facts and repeated events, $t(47) =$ –15.53, $p <$ .001, $g$ = –2.22, CI 95% [–2.75, –1.69]; general facts and unique events, $t(47) =$ –14.76, $p <$ .001, $g$ = –2.11, CI 95% [–2.62, –1.60]; autobiographical facts and repeated events, $t(47) =$ –15.25, $p <$ .001, $g$ = –2.18, CI 95% [–2.70, –1.66]; autobiographical facts and unique events, $t(47) =$ –14.97, $p <$ .001, $g$ = –2.14, CI 95% [–2.66, –1.63].

**Appendix 3—table 1.** Peaks of clusters for the facts vs. events contrast at lags 5–6 and 8–9.

| Lag | Bootstrap ratio | Cluster size (voxels) | X (mm) | Y (mm) | Z (mm) | Harvard-Oxford, probability atlas |
|---|---|---|---|---|---|---|
| Negative saliences | | | | | | |
| 5 | −5.82 | 140 | 57.5 | 10 | 12.5 | 43.0% Right Inferior Frontal Gyrus pars opercularis; 39.0% Right Precentral Gyrus |
| 5 | −4.92 | 100 | 67.5 | −42.5 | 27.5 | 14.0% Right Supramarginal Gyrus Posterior Division |
| 6 | −7.13 | 182 | 60 | 10 | 12.5 | 39.0% Right Precentral Gyrus; 28.0% Right Inferior Frontal Gyrus pars opercularis |
| 6 | −6.18 | 267 | 55 | −37.5 | 30 | 29.0% Right Supramarginal Gyrus Posterior Division; 9.0% Right Parietal Operculum Cortex; 8.0% Right Planum Temporale; 6.0% Right Supramarginal Gyrus Anterior Division |
| 6 | −5.69 | 303 | 45 | −47.5 | 55 | NA |
| 6 | −5.63 | 363 | 52.5 | −75 | −2.5 | 80.0% Right Lateral Occipital Cortex Inferior Division |
| 6 | −5.13 | 294 | −57.5 | −47.5 | 25 | 46.0% Left Supramarginal Gyrus Posterior Division; 13.0% Left Angular Gyrus |
| 6 | −4.96 | 213 | 45 | 42.5 | 22.5 | 87.0% Right Frontal Pole; 5.0% Right Middle Frontal Gyrus |
| 6 | −4.64 | 145 | −60 | 5 | 12.5 | 79.0% Left Precentral Gyrus |
| 6 | −4.59 | 204 | −50 | −75 | −2.5 | 78.0% Left Lateral Occipital Cortex Inferior Division |
| 6 | −4.52 | 85 | −37.5 | 47.5 | 27.5 | 74.0% Left Frontal Pole |
| 8 | −5.56 | 137 | 45 | 37.5 | 10 | 54.0% Right Frontal Pole; 21.0% Right Inferior Frontal Gyrus pars triangularis |
| 8 | −5.00 | 111 | 12.5 | −62.5 | 70 | 29.0% Right Lateral Occipital Cortex Superior Division; 6.0% Right Superior Parietal Lobule |
| 8 | −4.91 | 288 | 45 | −82.5 | 17.5 | 59.0% Right Lateral Occipital Cortex Superior Division |
| 8 | −4.88 | 136 | −47.5 | −75 | 7.5 | 62.0% Left Lateral Occipital Cortex Inferior Division; 11.0% Left Lateral Occipital Cortex Superior Division |
| 8 | −4.76 | 85 | 60 | 12.5 | 12.5 | 43.0% Right Inferior Frontal Gyrus pars opercularis; 27.0% Right Precentral Gyrus |
| 8 | −4.64 | 140 | 42.5 | −47.5 | 45 | 27.0% Right Angular Gyrus; 27.0% Right Supramarginal Gyrus Posterior Division; 11.0% Right Superior Parietal Lobule |
| 8 | −4.42 | 120 | 57.5 | −35 | 32.5 | 22.0% Right Parietal Operculum Cortex; 18.0% Right Supramarginal Gyrus Posterior Division; 10.0% Right Supramarginal Gyrus Anterior Division; 8.0% Right Planum Temporale |
| 9 | −5.53 | 466 | 47.5 | −80 | 12.5 | 46.0% Right Lateral Occipital Cortex Inferior Division; 34.0% Right Lateral Occipital Cortex Superior Division |
| 9 | −4.81 | 106 | −40 | −82.5 | 12.5 | 40.0% Left Lateral Occipital Cortex Inferior Division; 35.0% Left Lateral Occipital Cortex Superior Division |
| 9 | −4.80 | 187 | 32.5 | −52.5 | 50 | 44.0% Right Superior Parietal Lobule; 14.0% Right Angular Gyrus; 8.0% Right Lateral Occipital Cortex Superior Division |

*Appendix 3—table 1 Continued on next page*

*Appendix 3—table 1 Continued*

| Lag | Bootstrap ratio | Cluster size (voxels) | X (mm) | Y (mm) | Z (mm) | Harvard-Oxford, probability atlas |
|---|---|---|---|---|---|---|
| Positive saliences | | | | | | |
| 5 | 5.54 | 131 | −32.5 | −82.5 | 42.5 | 65.0% Left Lateral Occipital Cortex Superior Division |
| 5 | 5.60 | 193 | −27.5 | 15 | 52.5 | 37.0% Left Middle Frontal Gyrus; 15.0% Left Superior Frontal Gyrus |
| 5 | 5.79 | 378 | −12.5 | −60 | 20 | 49.0% Left Precuneus Cortex; 11.0% Left Supracalcarine Cortex |
| 5 | 7.25 | 101 | 17.5 | −92.5 | 0 | 48.0% Right Occipital Pole; 6.0% Right Lateral Occipital Cortex Inferior Division |
| 5 | 7.29 | 150 | −12.5 | −95 | 0 | 55.0% Left Occipital Pole |
| 6 | 5.58 | 152 | −22.5 | −22.5 | −17.5 | 44.0% Left Hippocampus; 12.0% Left Parahippocampal Gyrus Posterior Division; 6.0% Left Parahippocampal Gyrus Anterior Division |
| 6 | 5.76 | 228 | −5 | 57.5 | −10 | 61.0% Left Frontal Pole; 24.0% Left Frontal Medial Cortex |
| 6 | 6.10 | 96 | 25 | 25 | 45 | 35.0% Right Middle Frontal Gyrus; 34.0% Right Superior Frontal Gyrus |
| 6 | 6.38 | 100 | 45 | −75 | 37.5 | 68.0% Right Lateral Occipital Cortex Superior Division |
| 6 | 6.44 | 120 | −12.5 | −95 | −2.5 | 62.0% Left Occipital Pole |
| 6 | 6.92 | 100 | 17.5 | −95 | 0 | 50.0% Right Occipital Pole |
| 6 | 7.52 | 333 | −22.5 | 15 | 50 | 40.0% Left Superior Frontal Gyrus; 14.0% Left Middle Frontal Gyrus |
| 6 | 8.26 | 261 | −32.5 | −82.5 | 42.5 | 65.0% Left Lateral Occipital Cortex Superior Division |
| 6 | 8.68 | 823 | −15 | −62.5 | 22.5 | 54.0% Left Precuneus Cortex; 12.0% Left Supracalcarine Cortex |
| 8 | 4.60 | 84 | −5 | 32.5 | 40 | 47.0% Left Paracingulate Gyrus; 35.0% Left Superior Frontal Gyrus |
| 8 | 4.64 | 110 | −52.5 | 15 | 37.5 | 54.0% Left Middle Frontal Gyrus; 5.0% Left Inferior Frontal Gyrus pars opercularis |
| 8 | 5.60 | 183 | 25 | 30 | 50 | 43.0% Right Superior Frontal Gyrus; 18.0% Right Middle Frontal Gyrus |
| 8 | 5.99 | 143 | 45 | −75 | 37.5 | 68.0% Right Lateral Occipital Cortex Superior Division |
| 8 | 6.34 | 94 | −57.5 | −5 | −12.5 | 37.0% Left Middle Temporal Gyrus Anterior Division; 23.0% Left Superior Temporal Gyrus Anterior Division; 9.0% Left Superior Temporal Gyrus Posterior Division |
| 8 | 6.55 | 255 | −2.5 | −35 | 45 | 60.0% Left Cingulate Gyrus Posterior Division; 25.0% Left Precuneus Cortex |
| 8 | 7.14 | 438 | −22.5 | 25 | 45 | 47.0% Left Superior Frontal Gyrus; 20.0% Left Middle Frontal Gyrus |
| 8 | 10.43 | 1115 | −5 | −62.5 | 15 | 42.0% Left Precuneus Cortex; 16.0% Left Intracalcarine Cortex; 12.0% Left Supracalcarine Cortex |
| 8 | 10.58 | 424 | −35 | −82.5 | 42.5 | 57.0% Left Lateral Occipital Cortex Superior Division |
| 8 | 11.07 | 668 | −2.5 | 50 | −7.5 | 43.0% Left Frontal Medial Cortex; 38.0% Left Paracingulate Gyrus |
| 9 | 5.15 | 153 | 27.5 | 25 | 52.5 | 29.0% Right Middle Frontal Gyrus; 24.0% Right Superior Frontal Gyrus |
| 9 | 5.36 | 85 | −57.5 | −5 | −12.5 | 37.0% Left Middle Temporal Gyrus Anterior Division; 23.0% Left Superior Temporal Gyrus Anterior Division; 9.0% Left Superior Temporal Gyrus Posterior Division |
| 9 | 6.20 | 188 | −7.5 | 27.5 | 45 | 25.0% Left Superior Frontal Gyrus; 6.0% Left Paracingulate Gyrus |
| 9 | 7.35 | 1051 | 0 | 57.5 | −5 | 25.0% Right Frontal Pole; 20.0% Left Frontal Pole; 5.0% Right Frontal Medial Cortex |
| 9 | 7.50 | 554 | −25 | 27.5 | 45 | 30.0% Left Superior Frontal Gyrus; 29.0% Left Middle Frontal Gyrus |
| 9 | 7.85 | 1075 | 0 | −62.5 | 25 | 74.0% Left Precuneus Cortex; 16.0% Right Precuneus Cortex |
| 9 | 9.38 | 280 | −32.5 | −82.5 | 42.5 | 65.0% Left Lateral Occipital Cortex Superior Division |

Thresholded at bootstrap ratio at ± 3 ($p < .001$), minimal cluster size of 80 voxels, and a minimal distance of 10 voxels. Labels from the Harvard-Oxford (*Desikan et al., 2006*) atlas obtained using AtlasReader (*Notter et al., 2019*).

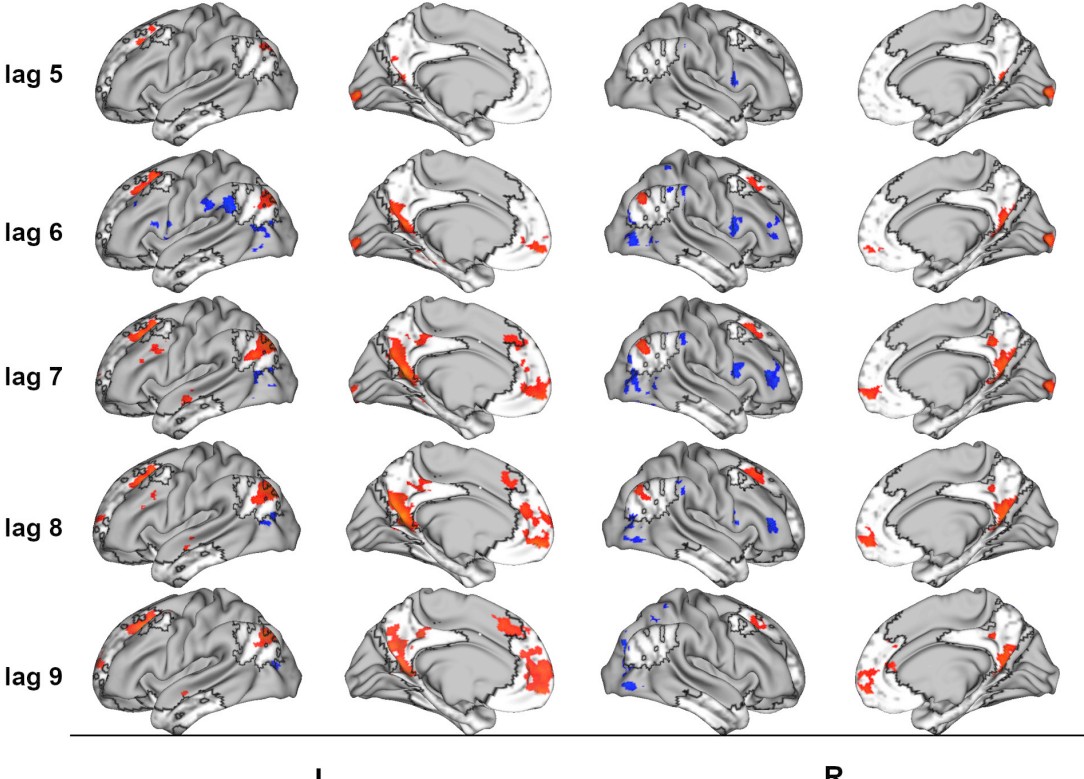

**Appendix 3—figure 1.** Brain scores for the latent variable (LV) 1 of the non-rotated partial least squares (PLS) with the facts vs. events contrast between memory conditions overlayed on the default mode and medial temporal networks from *Barnett et al., 2021* shown in white with black contour and projected onto a surface template from the Human Connectome Project (*Van Essen et al., 2012*) using Connectome Workbench (*Marcus et al., 2011*).

# Appendix 4

## Mean-centered PLS with memory and control condition

### PLS results

PLS produces LVs; in this case, LVs were variables that explained a maximum of the covariance between the four memory conditions (general facts, autobiographical facts, repeated events, unique event) and the spatiotemporal aspects of brain activity (**Abdi and Williams, 2013**; **Krishnan et al., 2011**; **McIntosh et al., 2004a**; **McIntosh and Lobaugh, 2004b**). The first significant LV from the PLS analysis distinguished the four memory conditions from a control condition (i.e., odd/even judgments), explaining 84.94% of the cross-block covariance ($p < .001$, see **Appendix 4—figure 1**). The inclusion of a control condition—involving little memory processes like odd/even judgments—along with memory conditions in a PLS tends to pull out the memory network because the dissociation between the control and memory conditions explains a maximum of the covariance in the data (**Addis et al., 2004a**; **Burianova and Grady, 2007**; **Sheldon and Levine, 2013**; **Spreng and Grady, 2010**).

At lag 7, increased activity in regions from the 'core memory network' dissociated the memory conditions from the control condition, including large regions of the prefrontal cortex and parietal cortex bilaterally, the medial temporal lobe, and the left lateral temporal cortex. The voxels that contributed the most to this brain LV, named 'saliences', can be found in **Appendix 4—table 1** and **Appendix 4—table 2** (**Abdi and Williams, 2013**).

A second significant LV ($p = .028$) accounted for 6.41% of the covariance. As this LV is similar to the non-rotated PLS described in the main text (i.e., general/autobiographical facts vs. repeated/unique events), we do not discuss it further. No other LVs were significant, $p > .05$.

**Appendix 4—table 1.** Peaks of clusters for latent variable (LV) 1 of the mean-centered partial least squares (PLS) (memory and control) at lag 7.

| Bootstrap ratio | Cluster size (voxels) | X (mm) | Y (mm) | Z (mm) | Harvard-Oxford, probability atlas |
|---|---|---|---|---|---|
| Negative saliences | | | | | |
| −19.56 | 9172 | −5.0 | −57.5 | 15.0 | 49% Left Precuneus Cortex; 8% Left Cingulate Gyrus Posterior Division |
| −15.77 | 5015 | −45.0 | 27.5 | −10.0 | 67% Left Frontal Orbital Cortex; 7% Left Inferior Frontal Gyrus Pars Triangularis |
| −13.30 | 260 | 5.0 | −55.0 | −47.5 | No Label |
| −13.21 | 2622 | −2.5 | 30.0 | 42.5 | 45% Left Paracingulate Gyrus; 30% Left Superior Frontal Gyrus |
| −12.42 | 364 | 47.5 | −72.5 | 37.5 | 57% Right Lateral Occipital Cortex Superior Division |
| −12.25 | 289 | 0.0 | 60.0 | −10.0 | 36% Right Frontal Pole; 31% Left Frontal Pole |
| −11.48 | 846 | 40.0 | −75.0 | −37.5 | No Label |
| −8.03 | 805 | 37.5 | −17.5 | 45.0 | 50% Right Precentral Gyrus; 26% Right Postcentral Gyrus |
| −7.96 | 221 | 40.0 | 40.0 | −10.0 | 36% Right Frontal Pole; 7% Right Frontal Orbital Cortex |
| −5.23 | 130 | 47.5 | −20.0 | 17.5 | 45% Right Parietal Operculum Cortex; 31% Right Central Opercular Cortex; 5% Right Postcentral Gyrus |
| −5.10 | 192 | 55.0 | 25.0 | 17.5 | 41% Right Inferior Frontal Gyrus Pars Triangularis; 21% Right Inferior Frontal Gyrus Pars Opercularis; 5% Right Middle Frontal Gyrus |
| Positive saliences | | | | | |
| 5.77 | 143 | −30.0 | −65.0 | −22.5 | No Label |
| 6.37 | 231 | 20.0 | 5.0 | 52.5 | 18% Right Superior Frontal Gyrus |
| 6.84 | 587 | 7.5 | −75.0 | 45.0 | 42% Right Precuneus Cortex; 17% Right Cuneal Cortex; 10% Right Lateral Occipital Cortex Superior Division |
| 7.86 | 329 | −27.5 | −12.5 | 57.5 | 39% Left Precentral Gyrus; 9% Left Superior Frontal Gyrus |
| 7.87 | 224 | −20.0 | −75.0 | −47.5 | No Label |

*Appendix 4—table 1 Continued on next page*

*Appendix 4—table 1 Continued*

| Bootstrap ratio | Cluster size (voxels) | X (mm) | Y (mm) | Z (mm) | Harvard-Oxford, probability atlas |
|---|---|---|---|---|---|
| 8.51 | 284 | 40.0 | −10.0 | −10.0 | 32% Right Planum Polare; 25% Right Insular Cortex |
| 8.63 | 417 | 7.5 | −32.5 | 47.5 | 30% Right Cingulate Gyrus Posterior Division; 27% Right Precentral Gyrus; 16% Right Precuneus Cortex; 9% Right Postcentral Gyrus |
| 8.74 | 1477 | −30.0 | −65.0 | 12.5 | No Label |
| 9.21 | 536 | 50.0 | 7.5 | 20.0 | 35% Right Precentral Gyrus; 31% Right Inferior Frontal Gyrus Pars Opercularis |
| 10.10 | 2341 | −15.0 | 32.5 | 5.0 | 8% Left Lateral Ventricle |
| 10.43 | 2352 | −60.0 | −32.5 | 52.5 | 20% Left Supramarginal Gyrus Anterior Division |
| 12.24 | 1112 | 57.5 | −35.0 | 52.5 | 43% Right Supramarginal Gyrus Posterior Division; 18% Right Supramarginal Gyrus Anterior Division |
| 12.40 | 3966 | 20.0 | −52.5 | −25.0 | No Label |

Note: Thresholded at bootstrap ratio at ± 3 (*p* < .001), minimal cluster size of 80 voxels, and a minimal distance of 10 voxels. Labels from the Harvard-Oxford (**Desikan et al., 2006**) atlas obtained using AtlasReader (**Notter et al., 2019**).

**Appendix 4—table 2.** Peaks of clusters for latent variable (LV) 1 of the mean-centered partial least squares (PLS) (memory and control) at lags 5–6 and 8–9.

| Lag | Bootstrap ratio | Cluster size (voxels) | X (mm) | Y (mm) | Z (mm) | Harvard-Oxford, probability atlas |
|---|---|---|---|---|---|---|
| Negative saliences | | | | | | |
| 5 | −17.08 | 2903 | −2.5 | −57.5 | 17.5 | 74% Left Precuneus Cortex; 19% Left Cingulate Gyrus Posterior Division |
| 5 | −15.47 | 5464 | −45.0 | 25.0 | −10.0 | 74% Left Frontal Orbital Cortex; 6% Left Frontal Operculum Cortex |
| 5 | −14.49 | 436 | −12.5 | −95.0 | −5.0 | 55% Left Occipital Pole |
| 5 | −13.18 | 1486 | −45.0 | −65.0 | 30.0 | 54% Left Lateral Occipital Cortex Superior Division; 8% Left Angular Gyrus |
| 5 | −12.36 | 555 | −55.0 | −7.5 | −12.5 | 28% Left Superior Temporal Gyrus Anterior Division; 24% Left Middle Temporal Gyrus Anterior Division; 14% Left Superior Temporal Gyrus Posterior Division; 5% Left Middle Temporal Gyrus Posterior Division |
| 5 | −12.16 | 1856 | 20.0 | −95.0 | 0.0 | 44% Right Occipital Pole |
| 5 | −10.94 | 380 | 50.0 | −72.5 | 35.0 | 47% Right Lateral Occipital Cortex Superior Division |
| 5 | −10.07 | 160 | 37.5 | 40.0 | −12.5 | 57% Right Frontal Pole; 11% Right Frontal Orbital Cortex |
| 5 | −8.82 | 416 | 35.0 | −65.0 | −30.0 | No Label |
| 5 | −8.51 | 105 | 60.0 | −5.0 | −17.5 | 48% Right Middle Temporal Gyrus Anterior Division; 14% Right Middle Temporal Gyrus Posterior Division; 5% Right Superior Temporal Gyrus Posterior Division |
| 5 | −8.11 | 134 | 5.0 | −57.5 | −50.0 | No Label |
| 5 | −7.17 | 658 | 47.5 | −15.0 | 52.5 | 44% Right Postcentral Gyrus; 23% Right Precentral Gyrus |
| 5 | −4.62 | 124 | 57.5 | 27.5 | 15.0 | 50% Right Inferior Frontal Gyrus Pars Triangularis; 14% Right Inferior Frontal Gyrus Pars Opercularis |
| 6 | −19.95 | 7309 | −2.5 | −57.5 | 17.5 | 74% Left Precuneus Cortex; 19% Left Cingulate Gyrus Posterior Division |
| 6 | −16.49 | 7663 | −45.0 | 27.5 | −10.0 | 67% Left Frontal Orbital Cortex; 7% Left Inferior Frontal Gyrus Pars Triangularis |
| 6 | −13.44 | 675 | −45.0 | −72.5 | 35.0 | 78% Left Lateral Occipital Cortex Superior Division |
| 6 | −12.30 | 388 | 50.0 | −70.0 | 35.0 | 78% Right Lateral Occipital Cortex Superior Division |
| 6 | −11.98 | 208 | 5.0 | −52.5 | −45.0 | No Label |
| 6 | −9.92 | 649 | 35.0 | −65.0 | −30.0 | No Label |
| 6 | −8.61 | 228 | 37.5 | 40.0 | −10.0 | 44% Right Frontal Pole; 11% Right Frontal Orbital Cortex |

*Appendix 4—table 2 Continued on next page*

*Appendix 4—table 2 Continued*

| Lag | Bootstrap ratio | Cluster size (voxels) | X (mm) | Y (mm) | Z (mm) | Harvard-Oxford, probability atlas |
|---|---|---|---|---|---|---|
| 6 | −8.51 | 757 | 45.0 | −22.5 | 52.5 | 44% Right Postcentral Gyrus; 5% Right Precentral Gyrus |
| 6 | −8.42 | 135 | 60.0 | −5.0 | −17.5 | 48% Right Middle Temporal Gyrus Anterior Division; 14% Right Middle Temporal Gyrus Posterior Division; 5% Right Superior Temporal Gyrus Posterior Division |
| 6 | −4.73 | 95 | 57.5 | 25.0 | 20.0 | 40% Right Inferior Frontal Gyrus Pars Triangularis; 16% Right Inferior Frontal Gyrus Pars Opercularis; 7% Right Middle Frontal Gyrus |
| 8 | −17.60 | 11,337 | −5.0 | −55.0 | 17.5 | 51% Left Precuneus Cortex; 20% Left Cingulate Gyrus Posterior Division |
| 8 | −14.98 | 8569 | −45.0 | 27.5 | −10.0 | 67% Left Frontal Orbital Cortex; 7% Left Inferior Frontal Gyrus Pars Triangularis |
| 8 | −13.29 | 299 | 5.0 | −52.5 | −45.0 | No Label |
| 8 | −11.48 | 315 | 50.0 | −70.0 | 37.5 | 59% Right Lateral Occipital Cortex Superior Division |
| 8 | −8.13 | 135 | −10.0 | −30.0 | −30.0 | 97% Brain-Stem |
| 8 | −7.75 | 238 | 35.0 | 25.0 | −2.5 | 40% Right Insular Cortex; 36% Right Frontal Orbital Cortex; 5% Right Frontal Operculum Cortex |
| 8 | −7.45 | 93 | −10.0 | 7.5 | 17.5 | 58% Left Lateral Ventricle; 41% Left Caudate |
| 8 | −7.06 | 285 | 45.0 | −25.0 | 17.5 | 76% Right Parietal Operculum Cortex; 6% Right Planum Temporale; 5% Right Heschl's Gyrus (includes H1 and H2) |
| 8 | −7.02 | 788 | 37.5 | −15.0 | 45.0 | 61% Right Precentral Gyrus; 13% Right Postcentral Gyrus |
| 8 | −5.23 | 157 | −2.5 | −12.5 | 62.5 | 50% Left Juxtapositional Lobule Cortex; 34% Left Precentral Gyrus |
| 8 | −4.85 | 96 | −35.0 | −32.5 | 22.5 | 44% Left Parietal Operculum Cortex |
| 9 | −17.53 | 11,252 | −22.5 | −80.0 | −7.5 | 43% Left Occipital Fusiform Gyrus; 13% Left Lingual Gyrus |
| 9 | −14.91 | 9678 | −47.5 | 40.0 | −5.0 | 70% Left Frontal Pole; 8% Left Inferior Frontal Gyrus Pars Triangularis; 5% Left Frontal Orbital Cortex |
| 9 | −14.14 | 332 | 7.5 | −52.5 | −45.0 | No Label |
| 9 | −9.50 | 728 | −60.0 | −45.0 | −5.0 | 33% Left Middle Temporal Gyrus Temporooccipital Part; 18% Left Middle Temporal Gyrus Posterior Division |
| 9 | −9.11 | 251 | 50.0 | −70.0 | 37.5 | 59% Right Lateral Occipital Cortex Superior Division |
| 9 | −8.04 | 581 | 37.5 | −17.5 | 12.5 | 69% Right Insular Cortex; 9% Right Heschl's Gyrus (includes H1 and H2) |
| 9 | −7.82 | 210 | 40.0 | 42.5 | −10.0 | 47% Right Frontal Pole |
| 9 | −6.74 | 120 | 17.5 | 5.0 | 17.5 | 72% Right Caudate |
| 9 | −6.15 | 195 | 37.5 | −15.0 | 42.5 | 57% Right Precentral Gyrus; 16% Right Postcentral Gyrus |
| 9 | −6.06 | 262 | −37.5 | −30.0 | 22.5 | 53% Left Parietal Operculum Cortex; 6% Left Central Opercular Cortex |
| 9 | −5.81 | 105 | 55.0 | 20.0 | 7.5 | 47% Right Inferior Frontal Gyrus Pars Opercularis; 25% Right Inferior Frontal Gyrus Pars Triangularis |
| 9 | −5.73 | 302 | 25.0 | −25.0 | 70.0 | 45% Right Precentral Gyrus; 15% Right Postcentral Gyrus |
| Positive saliences | | | | | | |
| 5 | 7.42 | 671 | 42.5 | 47.5 | 15.0 | 78% Right Frontal Pole |
| 5 | 7.64 | 380 | −35.0 | −50.0 | −30.0 | No Label |
| 5 | 8.93 | 189 | −22.5 | −75.0 | −52.5 | No Label |
| 5 | 10.12 | 1257 | 42.5 | 5.0 | 0.0 | 71% Right Insular Cortex; 7% Right Central Opercular Cortex |
| 5 | 13.79 | 5490 | 25.0 | −55.0 | −20.0 | No Label |
| 5 | 16.45 | 11,000 | −42.5 | −17.5 | 60.0 | 46% Left Precentral Gyrus; 15% Left Postcentral Gyrus |

*Appendix 4—table 2 Continued*

| Lag | Bootstrap ratio | Cluster size (voxels) | X (mm) | Y (mm) | Z (mm) | Harvard-Oxford, probability atlas |
|---|---|---|---|---|---|---|
| 6 | 6.87 | 84 | −15.0 | −27.5 | 37.5 | 23% Left Precentral Gyrus; 18% Left Cingulate Gyrus Posterior Division; 5% Left Precuneus Cortex |
| 6 | 7.61 | 318 | −35.0 | −50.0 | −30.0 | No Label |
| 6 | 7.95 | 672 | 45.0 | 47.5 | 15.0 | 88% Right Frontal Pole |
| 6 | 9.13 | 132 | 10.0 | −35.0 | 45.0 | 39% Right Cingulate Gyrus Posterior Division; 32% Right Precuneus Cortex; 5% Right Postcentral Gyrus; 5% Right Precentral Gyrus |
| 6 | 10.23 | 1203 | 40.0 | −10.0 | −7.5 | 55% Right Insular Cortex; 20% Right Planum Polare |
| 6 | 13.30 | 945 | 15.0 | −65.0 | −45.0 | No Label |
| 6 | 13.58 | 10,698 | −50.0 | −35.0 | 50.0 | 36% Left Supramarginal Gyrus Anterior Division; 23% Left Postcentral Gyrus; 14% Left Superior Parietal Lobule |
| 6 | 15.35 | 4346 | 17.5 | −55.0 | −17.5 | No Label |
| 8 | 5.27 | 123 | 22.5 | 5.0 | 60.0 | 43% Right Superior Frontal Gyrus |
| 8 | 5.35 | 132 | −25.0 | −62.5 | 32.5 | 24% Left Lateral Occipital Cortex Superior Division |
| 8 | 6.13 | 322 | −45.0 | −70.0 | −7.5 | 75% Left Lateral Occipital Cortex Inferior Division |
| 8 | 6.45 | 271 | 15.0 | −62.5 | −45.0 | No Label |
| 8 | 6.71 | 140 | −22.5 | −75.0 | −52.5 | No Label |
| 8 | 7.23 | 90 | −32.5 | −92.5 | −5.0 | 53% Left Occipital Pole; 21% Left Lateral Occipital Cortex Inferior Division |
| 8 | 7.35 | 331 | 12.5 | −65.0 | 40.0 | 43% Right Precuneus Cortex |
| 8 | 7.99 | 230 | −17.5 | −5.0 | 30.0 | No Label |
| 8 | 8.04 | 159 | 5.0 | −27.5 | 45.0 | 62% Right Cingulate Gyrus Posterior Division; 15% Right Precentral Gyrus; 7% Right Precuneus Cortex |
| 8 | 8.10 | 331 | 45.0 | 45.0 | 10.0 | 87% Right Frontal Pole |
| 8 | 8.14 | 455 | 50.0 | 7.5 | 22.5 | 41% Right Precentral Gyrus; 33% Right Inferior Frontal Gyrus Pars Opercularis |
| 8 | 8.52 | 384 | −25.0 | −55.0 | 15.0 | 10% Left Lateral Ventricle |
| 8 | 9.08 | 840 | −55.0 | −35.0 | 52.5 | 48% Left Supramarginal Gyrus Anterior Division |
| 8 | 9.84 | 1103 | −17.5 | 32.5 | 5.0 | 6% Left Lateral Ventricle |
| 8 | 11.53 | 3742 | 57.5 | −37.5 | 55.0 | 22% Right Supramarginal Gyrus Posterior Division |
| 9 | 5.45 | 152 | 22.5 | 7.5 | 60.0 | 40% Right Superior Frontal Gyrus; 6% Right Middle Frontal Gyrus |
| 9 | 6.37 | 131 | −22.5 | −75.0 | −47.5 | No Label |
| 9 | 7.02 | 444 | −55.0 | −37.5 | 55.0 | 33% Left Supramarginal Gyrus Anterior Division; 8% Left Supramarginal Gyrus Posterior Division |
| 9 | 7.02 | 123 | 7.5 | −32.5 | 47.5 | 30% Right Cingulate Gyrus Posterior Division; 27% Right Precentral Gyrus; 16% Right Precuneus Cortex; 9% Right Postcentral Gyrus |
| 9 | 7.70 | 150 | −17.5 | −7.5 | 30.0 | No Label |
| 9 | 7.89 | 284 | 45.0 | 45.0 | 10.0 | 87% Right Frontal Pole |
| 9 | 8.30 | 532 | −25.0 | −55.0 | 15.0 | 10% Left Lateral Ventricle |
| 9 | 9.07 | 714 | −12.5 | 30.0 | 2.5 | 12% Left Lateral Ventricle |
| 9 | 9.20 | 433 | 42.5 | 10.0 | 30.0 | 25% Right Precentral Gyrus; 21% Right Middle Frontal Gyrus; 17% Right Inferior Frontal Gyrus Pars Opercularis |
| 9 | 9.23 | 553 | −30.0 | −92.5 | −5.0 | 46% Left Occipital Pole; 14% Left Lateral Occipital Cortex Inferior Division |

*Appendix 4—table 2 Continued*

| Lag | Bootstrap ratio | Cluster size (voxels) | X (mm) | Y (mm) | Z (mm) | Harvard-Oxford, probability atlas |
|---|---|---|---|---|---|---|
| 9 | 10.06 | 2805 | 55.0 | −37.5 | 52.5 | 64% Right Supramarginal Gyrus Posterior Division; 11% Right Supramarginal Gyrus Anterior Division |
| 9 | 10.78 | 840 | 32.5 | −95.0 | −7.5 | 62% Right Occipital Pole; 10% Right Lateral Occipital Cortex Inferior Division |

Note: Thresholded at bootstrap ratio at ± 3 (*p* < .001), minimal cluster size of 80 voxels, and a minimal distance of 10 voxels. Labels from the Harvard-Oxford (*Desikan et al., 2006*) atlas obtained using AtlasReader (*Notter et al., 2019*).

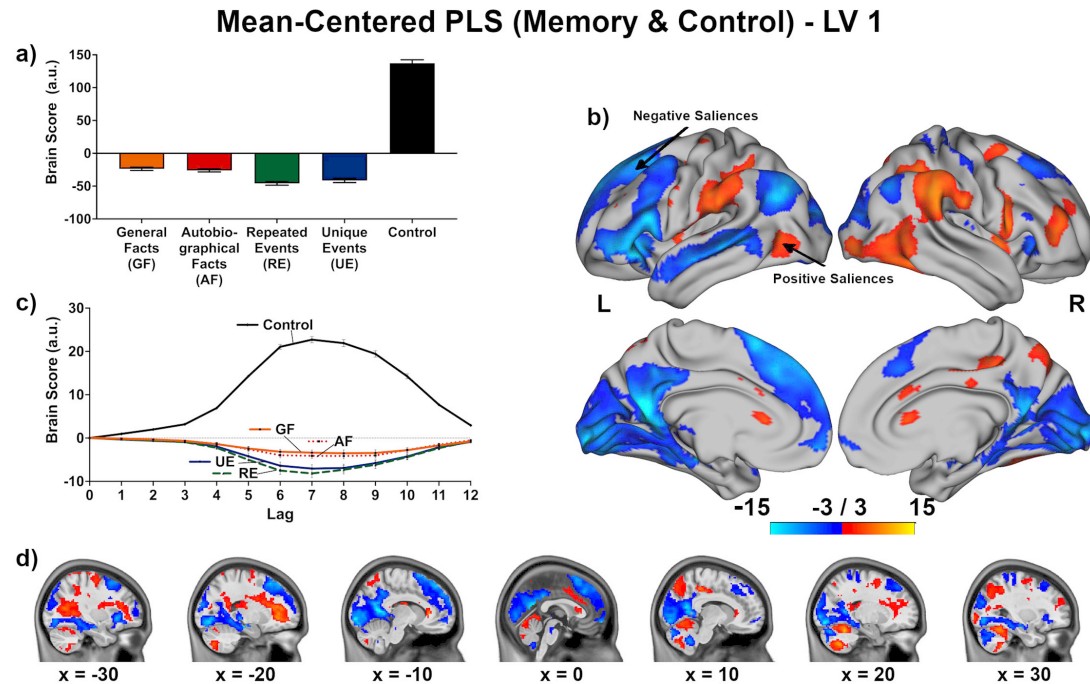

**Appendix 4—figure 1.** This mean-centered partial least squares (PLS) included the control task in addition to the memory conditions. The first significant latent variable (LV) identified activation that dissociated all memory conditions from the control task. (a) Average brain score. Error bars are ±1 *SE* of bootstrap estimates. (c) Brain scores shown at each lag (i.e., each TR/1.2 s). Error bars are ±1 *SE*. (b and d) Brain scores at lag 7 with positive saliences shown in warm colors (increased activity for the control task relative to memory conditions) and negative saliences shown in cold colors (increased activity for the memory conditions relative to the control task). Brain scores are projected onto a surface from the Human Connectome Project (S1200; *Van Essen et al., 2012*) using Connectome Workbench (*Marcus et al., 2011*) in (b) and the MNI152NLin2009cAsym volume using FSLeyes (*McCarthy, 2021*) in (d). Bootstrap ratios are thresholded at ± 3, *p* < .001, cluster size ≥ 80 voxels.

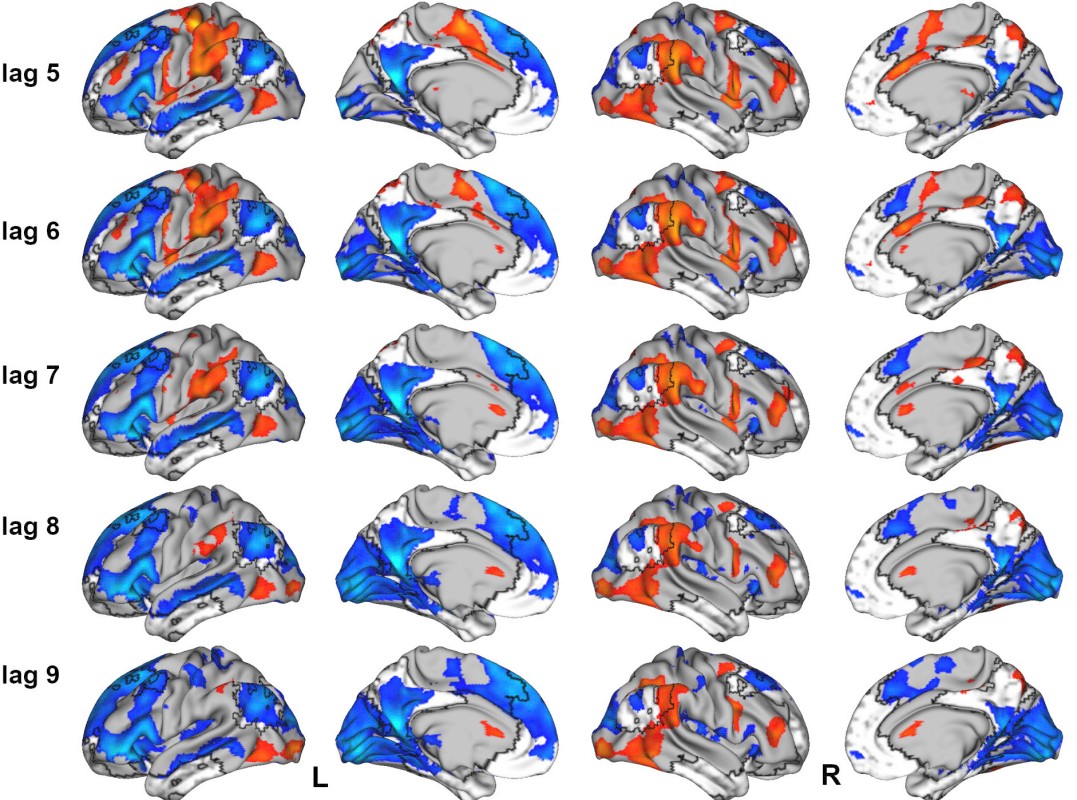

**Appendix 4—figure 2.** Brain scores for the latent variable (LV) 1 of the mean-centered partial least squares (PLS) that comprised the control task and the memory conditions overlayed on the default mode and medial temporal networks from *Barnett et al., 2021* shown in white with black contour and projected onto a surface template from the Human Connectome Project (*Van Essen et al., 2012*) using Connectome Workbench (*Marcus et al., 2011*).

# Appendix 5

## Stimuli list

**Appendix 5—table 1.** Practice task.

| Stimuli ID | Unique events (UE) *'Please respond (yes or no) if the following events happened to you.'* [1] | | Repeated events (RE) *'Please respond (yes or no) if the following events have happened to you repeatedly in the last year.'* | | Autobiographical facts (AF) *'Please respond (yes or no) according to what is usually true for you.'* | | General facts (GF) *'Please respond (yes or no) according to what is usually true for people in this country.'* | |
|---|---|---|---|---|---|---|---|---|
| | Cue | Sentence | Cue | Sentence | Cue | Sentence | Cue | Sentence |
| Ex. 1 | Yesterday | I bought a lottery ticket. | When with friends | I have bought a lottery ticket. | Often | I buy lottery tickets. | Few people | Buy lottery tickets. |
| Ex. 2 | Last weekend | I attended a wedding. | When on vacation | I attended a wedding. | Usually | I attend weddings. | Everyone | Attends weddings. |

Note: Participants practiced the task outside and inside the scanner using the trials listed in *Appendix 5—table 1*. The researcher explained the task: 'This task is going to present four kinds of statements to which you will answer yes or no. These statements are presented on a timer, so if you answer and the slide doesn't change right away, that's ok. You will be asked to answer questions in one of four ways.'

For the first one, please answer yes or no according to what happened to you once. For example, it could say: This morning, I paid bills. If that happened once, you would say yes.

For the second one, please answer yes or no accord to what happened to you repeatedly in the last year. For example, it could say: When at work, I paid bills. The key word is 'repeatedly'. If that happened repeatedly, you would say yes.

For the third one, please answer yes or no according to what is usually true to you. For example, it could say: Often, I pay bills. If that were true for you, you would say yes.

Last, please answer yes or no according to what you think is true of most people in this country. For example, a statement could be: Everyone pays bills. If you agree with this, you would say 'yes'.

## Response time

We entered response time in a repeated-measures ANOVA with memory (general facts, autobiographical facts, repeated events, unique events) and response (yes, no) as factors. Participants were faster to respond yes ($M = 730.97$, $SE = 24.58$) than to respond no ($M = 757.38$, $SE = 27.25$), $F(1, 47) = 6.21$, $p = .016$, $\eta_p^2 = 0.12$. We followed up on a main effect of memory, $F(3, 141) = 3.38$, $p = .020$, $\eta_p^2 = 0.07$, with paired samples $t$-tests corrected with a Holm-Bonferroni procedure. We computed the mean of yes and no responses for each condition prior to that step. Unique events ($M = 724.35$, $SE = 25.85$) had a faster response time than repeated events ($M = 750.55$, $SE = 26.21$), $t(47) = 2.93$, $p = .005$, $g = 0.42$, CI 95% [0.13, 0.71], and general facts ($M = 753.92$, $SE = 25.43$), $t(47) = 2.75$, $p = .009$, $g = 0.39$, CI 95% [0.10, 0.68], but not autobiographical facts ($M = 747.89$, $SE = 27.21$), $t(47) = 2.15$, $p = .037$, $g = 0.31$, CI 95% [0.02, 0.59]. None of the other comparisons were significant: general facts vs. autobiographical facts: $t(47) = 0.62$, $p = .541$, $g = 0.09$, CI 95% [–0.19, 0.37]; general facts vs. repeated events: $t(47) = 0.31$, $p = .758$, $g = 0.04$, CI 95% [–0.24, 0.33]; and autobiographical facts vs. repeated events: $t(47) = –0.25$, $p = .804$, $g = –0.04$, CI 95% [–0.32, 0.25]. The interaction between memory and response was not significant, $F(3, 141) = 0.97$, $p = .411$, $\eta_p^2 = 0.02$. Nevertheless, as we included only yes responses in the fMRI analyses, we repeated the ANOVA with yes responses and keeping only the memory factor. The main effect of memory was not significant, $F(2.56, 120.09) = 1.23$, $p = .302$, $\eta_p^2 = 0.03$, suggesting that all memory conditions produced similar response times when considering yes responses only. Even though seemingly similar to *Renoult et al., 2016*, for yes responses, participants waited until a response screen to produce their response and their speed does not reflect a true reaction time.

## Proportion of yes responses

We entered the proportion of yes responses in a repeated-measures ANOVA with memory as a factor (general facts, autobiographical facts, repeated events, unique events). The main effect of memory was significant, $F(2.01, 94.45) = 41.50$, $p < .001$, $\eta_p^2 = 0.47$ (see *Appendix 5—figure 1*). We followed up with paired sample $t$-tests that were corrected using a Holm-Bonferroni procedure. All $t$-tests were significant. As expected and as in *Renoult et al., 2016*, unique events ($M = 0.42$, $SE = 0.01$) had a smaller proportion of yes responses than general facts ($M = 0.55$, $SE = 0.01$), $t(47) = 10.03$, $p < .001$, $g = 1.44$, CI 95% [1.03, 1.84], autobiographical facts ($M = 0.59$, $SE = 0.01$), $t(47) = 14.09$, $p < .001$, $g = 2.02$, CI 95% [1.52, 2.51], and repeated events ($M = 0.50$, $SE = 0.02$) $t(47) = 5.29$, $p < .001$, $g = 0.76$, CI 95% [0.44, 1.07]. Further, repeated events had a smaller proportion of yes responses than general facts, $t(47) = 2.22$, $p = .031$, $g = 0.32$, CI 95% [0.03, 0.61], and autobiographical facts,

**Appendix 5—table 2.** Stimuli list.

| Stimuli ID | Unique events (UE) 'Please respond (yes or no) if the following events happened to you.'[1] | | | Repeated events (RE) 'Please respond (yes or no) if the following events have happened to you repeatedly in the last year.' | | | Autobiographical facts (AF) 'Please respond (yes or no) according to what is usually true for you.' | | | General facts (GF) 'Please respond (yes or no) according to what is usually true for people in this country.' | | |
|---|---|---|---|---|---|---|---|---|---|---|---|---|
| | Run | Cue | Sentence | Run | Cue | Sentence | Run | Cue | Sentence | Run | Cue | Sentence |
| 1 | B | Yesterday | I wore white socks. | G | When shopping | I have worn white socks. | F | Usually | I wear white socks. | C | Everyone | Wears white socks. |
| 2 | E | Last night | I took a shower. | F | When on vacation | I have taken a shower. | D | Often | I take showers. | D | Most people | Take showers. |
| 3 | C | Last night | I used a computer. | E | When at work | I have used a computer. | B | Very often | I use a computer. | G | Few people | Use a computer. |
| 4 | D | Today | I ate breakfast. | F | When with friends | I have eaten breakfast. | F | Usually | I eat breakfast. | F | Some people | Skip breakfast. |
| 5 | G | This morning | I made my bed. | F | When with friends | I have made my bed. | F | Every day | I make my bed. | B | Few people | Make their bed. |
| 6 | G | Last week-end | I drove on a highway. | B | When going to a clinic | I have driven on a highway. | G | Rarely | I drive on a highway. | G | Most people | Drive on the highway. |
| 7 | A | Yesterday | I ate a chocolate. | E | When shopping | I have eaten a chocolate. | D | Rarely | do I eat chocolate. | A | Most people | Eat chocolate. |
| 8 | G | This week | I read a book. | A | When on the bus | I have read books. | G | Rarely | I read books. | C | Few people | Read books. |
| 9 | C | This week | I ate at a restaurant. | C | When alone | I have eaten at a restaurant. | G | Often | I eat at restaurants. | G | No one | Eats at restaurants. |
| 10 | E | Last night | I watched tv. | A | When at work | I have watched TV. | G | Very often | I watch TV. | C | Everyone | Watches TV. |
| 11 | E | This week | I went shopping. | B | When on vacation | I have gone shopping. | C | Very often | I go shopping. | F | Few people | Go shopping. |
| 12 | D | This morning | I drank coffee. | E | When shopping | I have drunk coffee. | D | Every day | I drink coffee. | C | Many people | Drink coffee. |
| 13 | D | Today | I talked on the phone. | A | When on the bus | I have talked on the phone. | E | Very often | I talk on the phone. | F | Everyone | Talks on the phone. |
| 14 | E | Last night | I ate pizza. | E | When shopping | I have eaten pizza. | F | Every day | I eat pizza. | A | Most people | Eat pizza. |
| 15 | A | This week | I went to the movies. | C | When alone | I have gone to the movies. | G | Sometimes | I go to the movies. | D | Many people | Go to the movies. |
| 16 | A | Today | I read a newspaper. | G | When on the bus | I have read a newspaper. | A | Rarely | I read a newspaper. | B | Everyone | Reads a newspaper. |
| 17 | A | Today | I spent money. | C | When at work | I have spent money. | C | Every day | I spend money. | D | No one | Spends money. |

*Appendix 5—table 2 continued on next page*

Appendix 5—table 2 continued

| Stimuli ID | Unique events (UE) 'Please respond (yes or no) if the following events happened to you.'[1] | | | Repeated events (RE) 'Please respond (yes or no) if the following events have happened to you repeatedly in the last year.' | | | Autobiographical facts (AF) 'Please respond (yes or no) according to what is usually true for you.' | | | General facts (GF) 'Please respond (yes or no) according to what is usually true for people in this country.' | | |
|---|---|---|---|---|---|---|---|---|---|---|---|---|
| | Run | Cue | Sentence | Run | Cue | Sentence | Run | Cue | Sentence | Run | Cue | Sentence |
| 18 | D | Last week-end | I rented a movie. | C | When alone | I have rented a movie. | F | Rarely | I rent movies. | G | Some people | Rent movies. |
| 19 | A | Yesterday | I read a magazine. | F | When with friends | I have read a magazine. | B | Often | I read magazines. | G | Most people | Read magazines. |
| 20 | B | Yesterday | I listened to music. | C | When at work | I have listened to music. | D | Often | I listen to music. | G | Most people | Listen to music. |
| 21 | B | Yesterday | I washed dishes. | A | When at work | I have washed dishes. | D | Every day | I wash the dishes. | G | No one | Washes the dishes. |
| 22 | E | This week | I talked to a family member. | D | When at a clinic | I have talked to a family member. | C | Often | I talk to a family member. | D | Most people | Talk to family members. |
| 23 | F | Yesterday | I wore jeans. | A | When at work | I have worn jeans. | A | Sometimes | I wear jeans. | E | Most people | Wear jeans. |
| 24 | C | Last night | I slept well. | A | When alone | I have slept well. | C | Usually | I sleep well. | C | No one | Sleeps well. |
| 25 | F | This morning | I woke up early. | G | When on vacation | I have woken up early. | D | Usually | I wake up early. | C | No one | Wakes up early. |
| 26 | F | Today | I had chicken. | D | When at work | I have eaten chicken. | F | Sometimes | I eat chicken. | C | Everyone | Eats chicken. |
| 27 | G | Today | I listened to the radio. | D | When at a clinic | I have listened to the radio. | C | Every day | I listen to the radio. | F | Few people | Listen to the radio. |
| 28 | F | Last night | I went to bed early. | F | When with friends | I have gone to bed early. | B | Usually | I go to bed early. | A | Some people | Go to bed early. |
| 29 | F | Yesterday | I took a nap. | E | When on vacation | I have taken naps. | A | Often | I take naps. | E | No one | Takes naps. |
| 30 | B | Last night | I cooked dinner. | F | When with friends | I have cooked dinner. | G | Rarely | I cook dinner. | F | Many people | Cook dinner. |
| 31 | C | Last week-end | I went dancing. | C | When alone | I have gone dancing. | E | Often | I go dancing. | D | Few people | Go dancing. |
| 32 | D | This week | I watched sports. | D | When alone | I have watched sports. | B | Often | I watch sports. | D | Everyone | Watches sports. |
| 33 | G | This morning | I checked my email. | D | When on the bus | I have checked my email. | G | Often | I check my email. | G | Everyone | Checks their email. |
| 34 | B | This week | I played with a dog. | C | When alone | I have played with a dog. | G | Rarely | I play with dogs. | D | Some people | Play with dogs. |

Appendix 5—table 2 continued on next page

Appendix 5—table 2 continued

| Stimuli ID | Unique events (UE) 'Please respond (yes or no) if the following events happened to you.' [1] | | | Repeated events (RE) 'Please respond (yes or no) if the following events have happened to you repeatedly in the last year.' | | | Autobiographical facts (AF) 'Please respond (yes or no) according to what is usually true for you.' | | | General facts (GF) 'Please respond (yes or no) according to what is usually true for people in this country.' | | |
|---|---|---|---|---|---|---|---|---|---|---|---|---|
| | Run | Cue | Sentence | Run | Cue | Sentence | Run | Cue | Sentence | Run | Cue | Sentence |
| 35 | C | This week | I bought a CD. | G | When on vacation | I have bought CDs. | C | Rarely | I buy CDs. | D | No one | Buys CDs. |
| 36 | D | This week | I ate fries. | B | When on the bus | I have eaten fries. | F | Rarely | I eat fries. | C | Many people | Eat fries. |
| 37 | G | Last week-end | I went to the mall. | B | When on vacation | I have gone to the mall. | C | Often | I go to the mall. | E | No one | Goes to the mall. |
| 38 | F | This morning | I drank juice. | C | When shopping | I have drunk juice. | A | Very often | I drink juice. | B | Many people | Drink juice. |
| 39 | G | Today | I went on a walk. | A | When alone | I have gone on a walk. | A | Sometimes | I go on walks. | F | Some people | Go on walks. |
| 40 | F | Yesterday | I ate candy. | D | When at work | I have eaten candy. | B | Rarely | I eat candy. | E | Some people | Eat candy. |
| 41 | B | This morning | I went to the bank. | G | When with friends | I have gone to the bank. | G | Sometimes | I go to the bank. | A | Few people | Go to the bank. |
| 42 | C | Last week-end | I played a video game. | C | When alone | I have played a video game. | D | Very often | I play video games. | D | Most people | Play video games. |
| 43 | B | This week | I worked out. | E | When on vacation | I have worked out. | E | Very often | I work out. | C | Some people | Work out. |
| 44 | G | Last week-end | I did my laundry. | E | When with friends | I have done my laundry. | B | Very often | I do my laundry. | F | Everyone | Does their laundry. |
| 45 | E | Last week-end | I ate pancakes. | G | When with friends | I have eaten pancakes. | E | Sometimes | I eat pancakes. | B | Some people | Eat pancakes. |
| 46 | D | Yesterday | I logged on to Facebook. | D | When at work | I have logged on to Facebook. | C | Very often | I log on to Facebook. | A | Everyone | Logs on to Facebook. |
| 47 | F | Yesterday | I sent a text message. | D | When on the bus | I have sent a text message. | D | Every day | I send text messages. | A | Some people | Send text messages. |
| 48 | A | This week | I ate a sandwich. | B | When at work | I have eaten a sandwich. | F | Rarely | I eat sandwiches. | B | Some people | Eat sandwiches. |
| 49 | A | Last week-end | I hugged a friend. | F | When on vacation | I have hugged a friend. | E | Sometimes | I hug a friend. | E | Many people | Hug friends. |
| 50 | F | Last week-end | I watered a plant. | G | When on vacation | I have watered a plant. | G | Often | I water a plant. | A | Many people | Water plants. |
| 51 | A | Last night | I sang a tune. | E | When shopping | I have sung a tune. | F | Sometimes | I sing a tune. | F | No one | Sings tunes. |

*Appendix 5—table 2 continued*

| Stimuli ID | Unique events (UE) 'Please respond (yes or no) if the following events happened to you.'[1] | | | Repeated events (RE) 'Please respond (yes or no) if the following events have happened to you repeatedly in the last year.' | | | Autobiographical facts (AF) 'Please respond (yes or no) according to what is usually true for you.' | | | General facts (GF) 'Please respond (yes or no) according to what is usually true for people in this country.' | | |
|---|---|---|---|---|---|---|---|---|---|---|---|---|
| | Run | Cue | Sentence | Run | Cue | Sentence | Run | Cue | Sentence | Run | Cue | Sentence |
| 52 | A | This morning | I kissed somebody. | B | When on vacation | I have kissed somebody. | B | Every day | I kiss somebody. | B | Most people | Kiss others. |
| 53 | A | Last week-end | I bought a gift. | F | When with friends | I have bought a gift. | D | Rarely | I buy gifts. | E | Everyone | Buys gifts. |
| 54 | E | This morning | I went to the gym. | B | When on vacation | I have been to the gym. | E | Sometimes | I go to the gym. | B | Few people | Go to the gym. |
| 55 | C | Today | I missed a meeting. | A | When at work | I have missed a meeting. | D | Often | I miss a meeting. | A | No one | Misses meetings. |
| 56 | B | Last night | I had a drink. | C | When alone | I have had a drink. | A | Very often | I have a drink. | B | Many people | Gave a drink. |
| 57 | D | Yesterday | I took a picture. | F | When shopping | I have taken a picture. | A | Very often | I take pictures. | E | Few people | Take pictures. |
| 58 | D | Today | I went to a pharmacy. | B | When on vacation | I have been to the pharmacy. | A | Rarely | I go to the pharmacy. | C | Everyone | Goes to the pharmacy. |
| 59 | C | This morning | I prayed. | F | When with friends | I have prayed. | A | Sometimes | I pray. | A | Few people | Pray. |
| 60 | F | Last night | I heard jokes. | E | When shopping | I have heard jokes. | C | Every day | I hear jokes. | F | Everyone | Hears jokes. |
| 61 | C | This week | I gave to a charity. | E | When shopping | I have given to charity. | E | Usually | I give to charity. | B | Few people | Give to charity. |
| 62 | E | Last week-end | I visited a museum. | G | When with friends | I have visited a museum. | B | Rarely | I visit a museum. | D | Many people | Visit museums. |
| 63 | G | Yesterday | I took a course. | G | When on vacation | I have taken a course. | A | Every day | I take a course. | E | Many people | Take a course. |
| 64 | E | Last week-end | I had a cold. | G | When on vacation | I have had a cold. | B | Rarely | I have a cold. | B | Many people | Have a cold. |
| 65 | D | This week | I went swimming. | A | When at work | I have gone swimming. | B | Sometimes | I go swimming. | E | No one | Goes swimming. |
| 66 | B | Today | I checked the news online. | D | When at a clinic | I have checked the news online. | E | Every day | I check the news online. | F | Few people | Check the news online. |
| 67 | B | Last night | I drove my car. | A | When going to a clinic | I have driven my car. | F | Very often | I drive my car. | G | Most people | Drive their car. |
| 68 | G | Yesterday | I brushed my teeth. | D | When at a clinic | I brushed my teeth. | E | Very often | I brush my teeth. | E | Everyone | Brushes their teeth. |
| 69 | E | Yesterday | I ate fruit. | B | When shopping | I ate fruit. | C | Every day | I eat fruits. | A | Most people | Eat fruit. |

*Appendix 5—table 2 continued on next page*

*Appendix 5—table 2 continued*

| Stimuli ID | Unique events (UE) 'Please respond (yes or no) if the following events happened to you.' [1] | | Repeated events (RE) 'Please respond (yes or no) if the following events have happened to you repeatedly in the last year.' | | Autobiographical facts (AF) 'Please respond (yes or no) according to what is usually true for you.' | | General facts (GF) 'Please respond (yes or no) according to what is usually true for people in this country.' | |
| | Run | Cue | Sentence | Run | Cue | Sentence | Run | Cue | Sentence | Run | Cue | Sentence |
|---|---|---|---|---|---|---|---|---|---|---|---|---|
| 70 | C | Today | I washed my face. | B | When alone | I washed my face. | E | Usually | I wash my face. | G | No one | Washes their face. |

Note: [1]Brief instructions displayed at the beginning of each block within each run. [2]The memory order within each run was based on the Latin Square technique. Run A and G: RE, UE, AF, GF; Run B and D: UE, AF, GF, RE; Run C: AF, GF, RE, UE; Run E and F: GF, RE, UE, AF. Run order was randomized for each participant. Stimuli were randomized within each block per participant.

$t(47) = 4.24$, $p < .001$, $g = 0.61$, CI 95% [0.30, 0.91]. The proportion of yes responses was larger for autobiographical facts than general facts, $t(47) = -3.24$, $p = .002$, $g = -0.46$, CI 95% [−0.76, −0.17]. In raw numbers, this translates into an $M$ of 37.06 trials ($SE = 0.77$, range from 22 to 51) for general facts, $M$ of 39.81 trials ($SE = 0.76$, range from 24 to 56) for autobiographical facts, $M$ of 33.77 trials ($SE = 1.34$, range from 14 to 57) for repeated events, and $M$ of 28.02 trials ($SE = 0.70$, range from 18 to 38) for unique events.

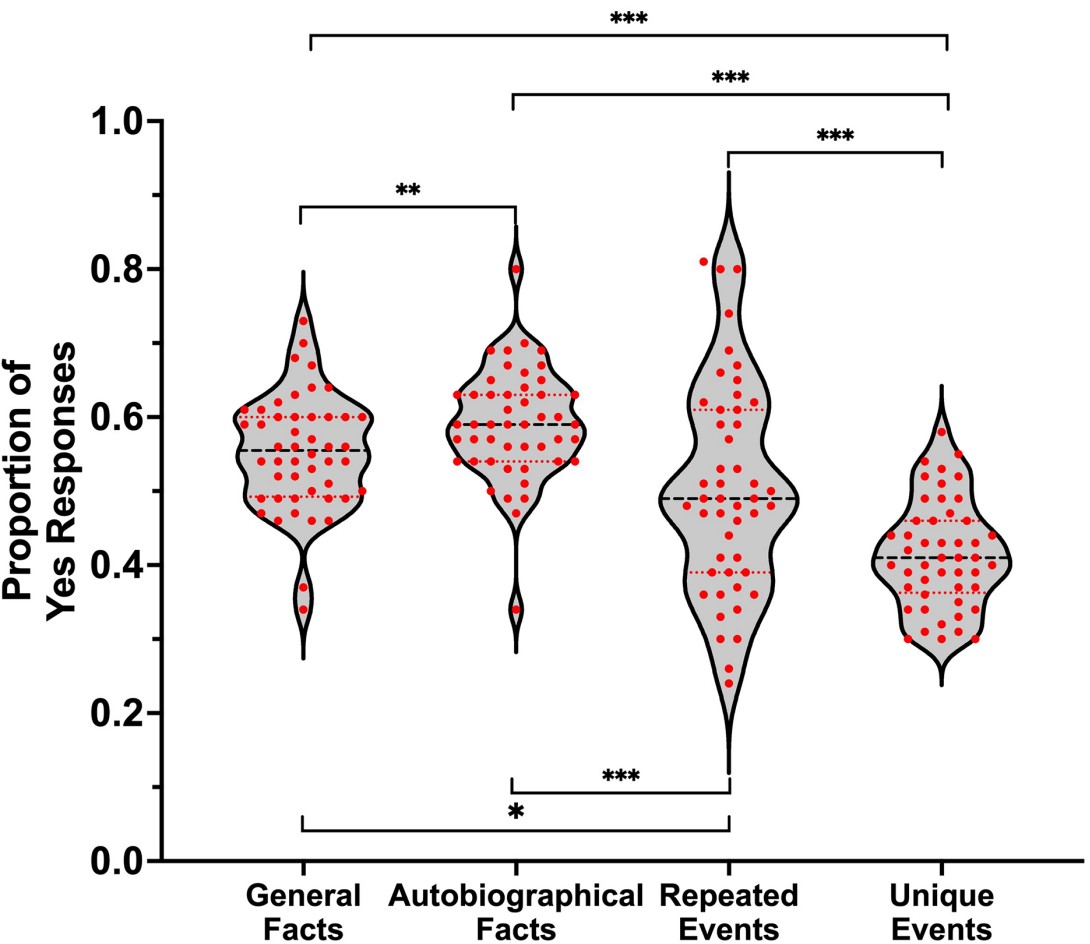

**Appendix 5—figure 1.** Proportion of yes responses relative to the total number of responses a participant made. Red points represent scores of individual participants ($N = 48$). A black line shows the median and red lines show the quartiles. *$p < .05$, **$p < .01$, ***$p < .001$.

