## [Editor Report]

There has been much theoretical and empirical work focused on the distinctions between episodic and semantic memory. The current work is important because it provides convincing and compelling evidence that this two-part model is not sufficient to explain the range of memory retrieval modes, particularly when considering personal semantic memory. This will be of great interest to memory researchers and will become a benchmark for future studies of autobiographical memory.

---

## [Decision Letter]

**Decision letter after peer review:**

Thank you for submitting your article "Personal Semantic, General Semantic, and Episodic Memory: Shared and Unique Neural Correlates" for consideration by *eLife*. Your article has been reviewed by 2 peer reviewers, and the evaluation has been overseen by a Reviewing Editor and Chris Baker as the Senior Editor. The following individual involved in the review of your submission has agreed to reveal their identity: Reece Roberts (Reviewer #1).

Both reviewers felt that the paper tackles an incredibly important distinction between episodic and semantic memory by considering some forms of memory that lie in between like personal semantics (self-knowledge). Instead of a dichotomy between episodic and semantic memory, the authors propose that the distinctions lie along a continuum. The reviewers both supported the use of PLS for addressing the questions proposed. The reviews did have several helpful considerations for the authors to address in a revision and I list them below (as well as include the comments at the end of the letter).

First, both reviewers noted that the sheer amount of results reported is overwhelming and they agreed (as do I) that the authors should limit reporting to the hypothesis-driven results (non-rotated) and leave the data-driven results for Supplemental data. We would not want the main targeted hypothesis-driven results to be lost in a sea of reporting, as they are close to doing now.

Second, it was suggested that the authors consider a greater link between the neuroimaging dataset and the behavioral dataset. This can be largely discussed in the discussion since it would be hard to propose a formal cross-experiment analysis. But comparing and contrasting the two results a bit more would be helpful and allow readers to generate new hypotheses to test in future studies.

Third, Reviewer 1 pointed out correctly that the contrast weights used to define a linear contrast do not do so in its current form. This analysis should be redone if the authors want to claim it is a linear (equidistant between conditions) effect. If not, please restate the purpose and interpretation of the analysis.

Fourth, there were several important detailed analysis issues brought up also by Reviewer #2 that should be addressed. One in particular related to the modeling of each trial with the inclusion of the cue itself, rather than the question. This reviewer rightly raises an important question about how the inclusion of the cue itself may be resulting in more overlap between conditions. Please address this and other questions about blocking trials.

Fifth, both reviewers felt the paper could be strengthened by not simply discarding questions with 'no' responses but rather using them to contrast with the 'yes' trials. Could the authors consider these additional analyses and what they might add to the paper?

Sixth, in terms of added discussion it was noted that reference to how dementias characterized by semantic or episodic memory loss have overlapping atrophy, particularly in the anterior hippocampus, would be an important empirical datapoint to include.

Chapleau, M., Aldebert, J., Montembeault, M., and Brambati, S. M. (2016). Atrophy in Alzheimer's Disease and Semantic Dementia: An ALE Meta-Analysis of Voxel-Based Morphometry Studies. Journal of Alzheimer's disease: JAD, 54(3), 941-955. https://doi.org/10.3233/JAD-160382

*Reviewer #1 (Recommendations for the authors):*

Using both neuroimaging and behavioural data, the current study investigates the relationship between personal semantics (i.e., self-knowledge) and the classical distinction between episodic and semantic memory. In the fMRI study, the authors employed a design in which participants were required to make yes/no decisions to a range of questions that either involved memory for unique events, memory for repeated events, autobiographical knowledge, or general knowledge. In a separate behavioural experiment, a range of subjective/phenomenological ratings (e.g., self-relevance, visual detail, scene rating) was made to the same conditions.

The manuscript is well-written and tackles a topic of central importance in the cognitive neuroscience of memory. While episodic and semantic memory are routinely thought of as two separate, interacting systems, the authors persuasively argue against such a clear-cut dichotomy in the introduction. Instead, they propose a continuum/spectrum ranging from the recollection of specific and repeated events (episodic) to personal semantics and general facts (semantic). The use of PLS is appropriate as it allows for the detection of whole-brain patterns associated with underlying latent variables and is frequently used in autobiographical memory and future imagination literature for this reason. The sample size is appropriate (final N = 48), and substantially larger than that of similar studies. A side note: the authors cite Grady et al. (2021) to claim that the ideal N is 80 in PLS analyses, but the Grady study specifically investigated how sample sizes affected the stability of brain-behaviour correlations so is not relevant to the current study.

Despite the many positive aspects of the current study, there are several issues that, if addressed, would benefit the clarity and impact of the manuscript. First, the authors use both a combination of data-driven (i.e., mean-centered) and hypothesis-driven (i.e., non-rotated) PLS analyses, and the volume of results reported could be substantially reduced if, instead, a series of hypothesis-driven analyses were carried out that mapped onto the authors' hypotheses. While data-driven PLS approaches are useful for exploring data, a drawback is that patterns of covariance are often detected that are not related to any specific hypotheses. In addition, the number of significant latent variables produces a large volume of results that are spread across the main manuscript and appendices. As the authors do have specific hypotheses, a series of targeted, non-rotated hypotheses would be more appropriate in delineating the relationship between the experimental conditions.

Second, I think greater emphasis could have been placed on outlining the relationship between neuroimaging and behavioural experiments. While there is probably no way to link the two statistically, as the two datasets involve separate sets of participants, I do think the Discussion section would benefit greatly from the authors theorising about how the differences in conditions that appear in the behavioural experiment map onto brain regions apparent in the fMRI results. For example, there is a linear increase in visual details in the behavioural experiment and a set of regions that show the same effect in the neuroimaging experiment – which of those brain regions do the authors propose is associated with the increase in visual detail? Which are associated with the increase in self-relevance? It should be added that the interpretation of functional roles of brain regions would be greatly enhanced if percent signal change values were extracted and plotted for key regions in the core network.

Regarding the non-rotated PLS analysis testing for a contrast across the four memory conditions, the authors coded for the following contrast: [-2,-1, 1, 2]. This is described as a linear contrast, but this is not the case, as it does not code for an equal difference between conditions. It codes for a smaller difference between two "episodic" conditions and the two "semantic" conditions and a larger difference between the repeated events and autobiographical facts conditions. This can be observed in Figures5A and 5C. In addition, the BSR images (Figure 5B and Figure 5D) look very similar to those observed in the first mean-centered "Memory only" LV, which distinguishes between episodic and semantic conditions (Figure 3). A true linear contrast would be: [-3 -1 1 3]. If the goal was to run a linear contrast, this analysis should be redone. Alternatively, the authors should provide a rationale about why the contrast was coded in the way that it was-what are the reasons for predicting larger differences between repeated events and autobiographical facts conditions than, for example, the unique and repeated events?

Lastly, the central idea of the current manuscript is that within the extended core memory network (negatively weighted regions in Figure 1), a set of elementary operations are weighted differently depending on the type of memory being recalled. For this framework to hold, it's important to know if the differences observed between the memory conditions (e.g., Figure 3, Figure 5) are observed in a set of regions within the core memory network or if there are regions outside the core network that show differences between the memory conditions.

As stated in the public review, I think this manuscript has the potential to be of interest to many researchers in the cognitive neuroscience of memory. In addition to the issues raised in the public review, I think the design of the neuroimaging experiment allows for supplementary analyses that could produce results that corroborate the main findings. If I understand the design correctly, only the "Yes" responses were included in the fMRI analyses and the "No" responses were discarded. I wonder if there is an opportunity to directly contrast the Yes and No responses for each condition, as presumably, these should isolate the neural correlates associated with each memory condition and there should be notable differences in those correlates for each memory condition (e.g. greater hippocampal responses in the unique event Yes > No contrast relative to the general facts Yes > No contrast).

I think the addition of the large volume of results in the appendix section is commendable in some respects, I am skeptical if a reader will actually consume the large volume of data in the tables. Perhaps linking to the overlay nifti files hosted on NeuroVault may be more appropriate?

*Reviewer #2 (Recommendations for the authors):*

This study uses advanced fMRI methods to address important and theoretically-motivated questions about memory retrieval as it occurs in the real world. Central to this study is the notion that, when studying how people recall information for their lives, we can understand a potential intermediary form of memory that exists between episodic and semantic memory: personal semantics. The main study was an fMRI study in which participants responded to questions that involved accessing information from episodic memories, two forms of personal semantics – repeated events and autobiographical knowledge – as well as semantic memory. A parallel behavioural experiment on a different sample of participants was conducted to examine how these forms of information differ with respect to subjective ratings.

A major strength of this paper is that it questions the classic division between episodic and semantic memory, proposing alternate views for how to consider retrieval from declarative memory. The use of PLS to analyze the data was a smart choice, albeit there are some considerations for these analyses. It is unfortunate that the neuroimaging and behavioural experiments were not collected on the sample. It might be useful to consider how the behavioural results can inform the neuroimaging experiment rather than considering the behavioural experiment solo.

I have some suggestions to help strengthen the manuscript, which are listed below in order of presentation rather than importance.

The introduction reviews all the key pieces of literature. When noting the overlap between episodic and semantic memory wrt the hippocampus, you may wish to include literature that notes how dementias characterised by semantic or episodic memory loss do have overlapping atrophy, particularly in the anterior hippocampus

Chapleau, M., Aldebert, J., Montembeault, M., and Brambati, S. M. (2016). Atrophy in Alzheimer's Disease and Semantic Dementia: An ALE Meta-Analysis of Voxel-Based Morphometry Studies. Journal of Alzheimer's disease: JAD, 54(3), 941-955. https://doi.org/10.3233/JAD-160382

In the introduction. I think the rationale for the study could be strengthened by explicitly stating how the resulting work with contribute to new memory theories and extend from the reviewed literature. Simply said – note exactly what knowledge gap is being filled.

In terms of the methods, the choice to block the conditions within each run raises the question of whether the results are due to differences in the content being accessed (semantic or episodic) or retrieval state (retrieval mode, in the words of Tulving) effects. Could the authors speak to this? (I come back to this with some suggestions for the discussion)

For the analysis, the authors note that the PLS analysis began at cue onset, and they estimated activity for 12 TR (14.4. seconds). What was the choice to analyze at the cue onset rather than the question? It seems to me that including the BOLD response associated with the cue means that your signal included both activities related to that cue, which is different across the conditions (some are temporal cues, some are activity cues, and some reflect people), as well as activity in accessing the intended information (a fact or event).

I also found that the stimuli for the general facts condition seem to reflect opinions (Few people take pictures) rather than facts (The Queen of England is dead). Can the authors speak to this?

The PLS analysis also seems to be focused on yes responses, however, from Appendix 6 it seems like this means that between.6 and.4 proportion of the trials are not being analyzed. Would it not be useful for the authors in the trials by yes or no responses to see how ts drove neural activity?

In terms of the control task, the authors note why they included a control task e (page 7, line 11), but I would like to know why the authors selected the odd/even task. That is, what processes are being controlled for with this task? It seems to me that it would be general internally directed attention vs externally directed attention rather than specifying a memory network, per se. Does the choice of this task alter the pattern pulled out by PLS?

In terms of results, from Figure 1 it also seems that the negative brain pattern was being represented more significantly by RE and UE than GF or AF (as the CI for the brain scores does not overlap) – is this correct? If so, this would suggest to me that there is more dissociation between facts vs events than commonalities.

In how the results are presented, I had a hard time integrating the behavioural and neuroimaging data, perhaps the paper would benefit from presenting the behavioural study first to show the distinctions in the memory cues and then the neuroimaging data to examine neural mechanisms.

The authors introduce the idea of the results reflecting the engagement of the DMN and different subsystems, but their results present patterns of activity that are not constrained to these networks or reflect connectivity, a better metric of network engagement. Perhaps the authors could constrain the PLS analysis to the DMN network so that they map their findings to the reviewed work about DMN contributions to autobiographical memory.

I also think it would be great to focus the discussion on specific theories that fall within a component process framework to suggest why the resulting patterns emerged, such as theories that result reflect accessing context dependent vs independent information (Grilli et al.,) or different levels of consciousness (Tulving). In fact, I was curious if the authors thought that the distinctions are because different content is being accessed or the same content at different levels of representation. Both have relevance to theories on accessing information from memory that are noted. For example, the PMAT model discussing distinctions in content versus the TTT (Gilboa and Moscovitch) would note distinctions in representation, at least that is my understanding.

---

## [Author Response]

Essential revisions:Reviewer #1 (Recommendations for the authors):Using both neuroimaging and behavioural data, the current study investigates the relationship between personal semantics (i.e., self-knowledge) and the classical distinction between episodic and semantic memory. In the fMRI study, the authors employed a design in which participants were required to make yes/no decisions to a range of questions that either involved memory for unique events, memory for repeated events, autobiographical knowledge, or general knowledge. In a separate behavioural experiment, a range of subjective/phenomenological ratings (e.g., self-relevance, visual detail, scene rating) was made to the same conditions.The manuscript is well-written and tackles a topic of central importance in the cognitive neuroscience of memory. While episodic and semantic memory are routinely thought of as two separate, interacting systems, the authors persuasively argue against such a clear-cut dichotomy in the introduction. Instead, they propose a continuum/spectrum ranging from the recollection of specific and repeated events (episodic) to personal semantics and general facts (semantic). The use of PLS is appropriate as it allows for the detection of whole-brain patterns associated with underlying latent variables and is frequently used in autobiographical memory and future imagination literature for this reason.1. The sample size is appropriate (final N = 48), and substantially larger than that of similar studies. A side note: the authors cite Grady et al. (2021) to claim that the ideal N is 80 in PLS analyses, but the Grady study specifically investigated how sample sizes affected the stability of brain-behaviour correlations so is not relevant to the current study.

Thank you for pointing out the important nuance of Grady et al. We have edited this statement in the methods. “This sample size was the largest that was possible to achieve. The sample size is larger than similar studies (e.g. *N* ~12-28; Addis et al., 2011; Burianova and Grady, 2007; Ford et al., 2011; Holland et al., 2011). No formal power analysis was conducted.”.

2. Despite the many positive aspects of the current study, there are several issues that, if addressed, would benefit the clarity and impact of the manuscript. First, the authors use both a combination of data-driven (i.e., mean-centered) and hypothesis-driven (i.e., non-rotated) PLS analyses, and the volume of results reported could be substantially reduced if, instead, a series of hypothesis-driven analyses were carried out that mapped onto the authors' hypotheses. While data-driven PLS approaches are useful for exploring data, a drawback is that patterns of covariance are often detected that are not related to any specific hypotheses. In addition, the number of significant latent variables produces a large volume of results that are spread across the main manuscript and appendices. As the authors do have specific hypotheses, a series of targeted, non-rotated hypotheses would be more appropriate in delineating the relationship between the experimental conditions.

We now report only hypotheses-driven analyses in the paper to reduce the volume of results. Given the conceptual similarity of the non-rotated (NR) PLS contrasts and mean-centered (MC) PLS with memory conditions only, we omitted the latter. Nevertheless, the ability to elucidate the relation between memory types with little constraints, in a data-driven manner, was one of the strongest appeals of PLS. Hence, we moved the MC PLS with memory conditions and the control task to the Supplementary Information. The benefit of preserving this MC PLS is twofold. First, a potential pitfall is that readers may interpret PLS results like univariate analyses. The first mean-centered PLS clarifies the narrative through highlighting the commonalities across memory conditions and this follows in the footsteps of many PLS papers in the literature (e.g., Addis et al., 2004; Sheldon and Levine, 2005). Incidentally, the LV reinforces confidence in the validity of our findings because the typical declarative memory network emerges from the memory conditions vs. control LV (i.e., the first LV). As described in the original submission of the manuscript, the second LV from that MC PLS was relatively similar to the general/autobiographical facts vs. repeated/unique events LV (simplified to facts vs. events) of the second MC PLS, now omitted. The finding using a data-driven analysis bolsters confidence in the NR PLS findings and informs the interpretation of data.

3. Second, I think greater emphasis could have been placed on outlining the relationship between neuroimaging and behavioural experiments. While there is probably no way to link the two statistically, as the two datasets involve separate sets of participants, I do think the Discussion section would benefit greatly from the authors theorising about how the differences in conditions that appear in the behavioural experiment map onto brain regions apparent in the fMRI results. For example, there is a linear increase in visual details in the behavioural experiment and a set of regions that show the same effect in the neuroimaging experiment – which of those brain regions do the authors propose is associated with the increase in visual detail? Which are associated with the increase in self-relevance?

We followed the suggestion from Comment R2.9 to invert the order of fMRI and behavioural results, and further anchored the a-priori contrasts in behavioural findings. We also discussed the behavioural results before some of the neuroimaging findings to better contrast their relation in the discussion, refining the links as appropriate.

In results: “We used a non-rotated PLS to test two theoretically plausible relations between the four memory conditions: a linear contrast (-3, -1, 1, 3) and one comparing general/autobiographical facts and repeated/unique events (-1, -1, 1, 1; abbreviated to facts vs. events subsequently). A linear contrast would be consistent with the continuum perspective of personal semantics (see Box 3; Renoult et al., 2012), which would predict an increase in activity from general facts to autobiographical facts, from autobiographical facts to repeated events, and from repeated events to unique events. The increase in visual details (described above) followed precisely this pattern. Similarly, personal relevance increased from general facts to personal semantics (autobiographical facts and repeated events) to unique events, suggesting similar dynamics between component processes (e.g., contextual specificity may increase along with personal relevance). The facts vs. events contrast would favour the view of personal semantics as a subtype of general semantics (Box 2; Renoult et al., 2012). Of all personal semantics, autobiographical facts corresponds best with this view due to its abstraction from events, its more objective quality than other forms of personal knowledge (e.g., trait knowledge), and the feeling of “knowing” the “facts” rather than recollecting events (i.e., “noetic” consciousness; Tulving, 2002). Repeated events would instead group with unique events as “event memory” due to the common construction of a scene (Rubin and Umanath, 2015). Indeed, participants in the behavioural study perceived scenes as frequently for repeated and unique events. However, it is less clear how one would accommodate the greater number of scenes evoked for autobiographical than general facts in a way that aligns with that perspective. Thus, the conjunction of the visualization of scenes, amount of visual details, and personal relevance agrees most with a continuum of contextual specificity.”

In discussion: “The importance of situational or contextual elements featured strongly in the neuroimaging data as in the behavioural data. The core network, also known as the default mode network, has been subdivided into an anterior temporal network linked to “entities” (Ranganath and Ritchey, 2012; Reagh and Ranganath, 2018) or “conceptual remembering” (Sheldon et al., 2019) and a posterior medial network linked to “situational models” (Ranganath and Ritchey, 2012; Reagh and Ranganath, 2018) or “perceptual remembering” (Sheldon et al., 2019). Although these networks process different kinds of information, neither is strictly dedicated to semantic or episodic memory (Reagh and Ranganath, 2018; Sheldon et al., 2019). For instance, knowledge can facilitate the search and construction of events (Irish and Piguet, 2013) and semantic memory can integrate contextual information (e.g., Greenberg et al., 2009; Sheldon and Moscovitch, 2012). Accordingly, in our study the dissociation between general/autobiographical facts and repeated/unique events did not have a clear posterior medial to anterior temporal demarcation (i.e., posterior medial activity for events and anterior temporal activity for facts). Instead, regions primarily within the posterior medial network (i.e., angular gyrus, posterior cingulate gyrus, precuneus, and parahippocampal gyrus; Ritchey et al., 2015) dissociated memory conditions, whereas those within the anterior temporal network (i.e., frontal orbital cortex, inferior anterior temporal gyrus, temporal pole, bilateral amygdala, and perirhinal cortex; Ritchey et al., 2015) did not. Indeed, events contained greater contextual information, as suggested by the increased proportion of scenes they evoked, as compared to factual memories. Further, cues were more temporally specific for repeated/unique events than general/autobiographical facts. Consistent with this, many regions of the posterior medial network were associated with greater activity for repeated/unique events than general/autobiographical facts (similar to Ford et al., 2011; Levine et al., 2004; Maguire and Mummery, 1999), and showed a linear increase from the most general type of memory (i.e., general facts) to the most specific memory (i.e., unique events). Therefore, activity in regions associated with visuospatial processing (e.g., precuneus) and scenes (e.g., medial temporal regions) coheres with behavioural data to support the prominence of contextual specificity in determining the relation across memory types. Activity in medial frontal regions is in harmony with ratings of visual details and scene perception, likewise increasing along a continuum of contextual specificity (see Figure 5d). However, this anterior portion of the medial frontal cortex may correspond best to self-processing rather than “situational” processing or mental time travel (Lieberman et al., 2019). If activity in the medial frontal cortex in our study reflected exclusively self-processing, one would expect a greater proximity between autobiographical facts and repeated events on the basis of subjective ratings of self-relevance. The additional concordance of medial frontal cortex with a continuum of contextual specificity could be a corollary of the strong links between aspects of self-relevance and episodic simulation (Tulving et al., 2002; King et al., 2022; Grysman et al., 2013; Verfaellie et al., 2019), in addition to this region’s role in modulating recollection (McCormick et al., 2020), for example through engaging schema-related information (Gilboa and Marlatte, 2017).”

4. It should be added that the interpretation of functional roles of brain regions would be greatly enhanced if percent signal change values were extracted and plotted for key regions in the core network.

We added percent signal change for peaks within key regions of the core network in Figure 5. Two of the peaks (i.e., a and c) were common between the two LVs, and so also serve to illustrate the compatibility of the two contrasts.

5. Regarding the non-rotated PLS analysis testing for a contrast across the four memory conditions, the authors coded for the following contrast: [-2,-1, 1, 2]. This is described as a linear contrast, but this is not the case, as it does not code for an equal difference between conditions. It codes for a smaller difference between two "episodic" conditions and the two "semantic" conditions and a larger difference between the repeated events and autobiographical facts conditions. This can be observed in Figures5A and 5C. In addition, the BSR images (Figure 5B and Figure 5D) look very similar to those observed in the first mean-centered "Memory only" LV, which distinguishes between episodic and semantic conditions (Figure 3). A true linear contrast would be: [-3 -1 1 3]. If the goal was to run a linear contrast, this analysis should be redone. Alternatively, the authors should provide a rationale about why the contrast was coded in the way that it was-what are the reasons for predicting larger differences between repeated events and autobiographical facts conditions than, for example, the unique and repeated events?

Thank you for noting this error. We repeated this PLS analysis with the correct contrasts: [-3 -1 1 3]. The corrected results are now reported in the paper with a pattern of results that are similar (albeit not identical) to the erroneous contrast. Critically, the new PLS results do not change the interpretations made in our initial draft.

6. Lastly, the central idea of the current manuscript is that within the extended core memory network (negatively weighted regions in Figure 1), a set of elementary operations are weighted differently depending on the type of memory being recalled. For this framework to hold, it's important to know if the differences observed between the memory conditions (e.g., Figure 3, Figure 5) are observed in a set of regions within the core memory network or if there are regions outside the core network that show differences between the memory conditions.

In response to this comment, we repeated the non-rotated PLS using a mask of the default mode network (DMN; anterior temporal, medial prefrontal, posterior medial) and medial temporal network from Barnett et al. (2021) to test whether these regions would suffice to dissociate the memory conditions. Note that Barnett et al. conceptualize an MTL network as dissociable from the DMN, however the authors reference this subnetwork as a critical DMN bridge between DMN and the hippocampus and other networks. In light of this interpretation and the more general disagreement to date about the exact neuroanatomical boundaries of the DMN, we felt it was important to include Barnett's MTL network in our mask. Using Barnett et al.’s (2021) mask rather than a mask of the extended core network from our data (i.e., negatively weighted regions now shown in Appendix 4**—**figure 1) helped to avoid circularity.

The results of this ancillary non-rotated PLS were comparable to those from the main text. To further highlight the importance of the core memory network in determining the relation between memory conditions, we now display brain scores overlayed on Barnett et al.’s (2021) mask in Appendices 2 and 3 Figure**—**1, Appendix 4**—**Figure 1. The regions associated with a linear increase and showing increased activity for repeated/unique events than general/autobiographical facts predominantly fall within Barnett et al.’s (2021) networks. Altogether our data are consistent primarily with a quantitative rather than qualitative difference between memory conditions, that is, with mostly the same regions being engaged but to a different extent. Nevertheless, our data do not exclude the contribution of other regions. Regions associated with less decreased activity for general/autobiographical facts than repeated/unique events (i.e., negative saliences in Table 4) fall largely outside of the extended core network shown in Appendix 4**—**figure 1 and Barnett et al.’s network: the right frontal pole, the right inferior frontal gyrus, the right lateral occipital cortex, the right supramarginal gyrus, and the left occipital cortex. We nuanced the conclusions accordingly.

We added the following footnote in the discussion: “We obtained comparable non-rotated PLS results while including only voxels within the default mode and medial temporal networks from Barnett et al. (2021): 49.56% crossblock covariance (*p* < .001) for the linear contrast, and 50.44% crossblock covariance (*p* < .001). Within the selected networks, the same regions contributed to dissociate the memory conditions, and temporal brain scores peaked at lag 7. This supplementary analysis reinforces the importance of regions within the core memory network to determine the relation between memory conditions (see Appendices 2 and 3, figure 1), even though we found additional contributors at the whole brain level.”

7. As stated in the public review, I think this manuscript has the potential to be of interest to many researchers in the cognitive neuroscience of memory. In addition to the issues raised in the public review, I think the design of the neuroimaging experiment allows for supplementary analyses that could produce results that corroborate the main findings. If I understand the design correctly, only the "Yes" responses were included in the fMRI analyses and the "No" responses were discarded. I wonder if there is an opportunity to directly contrast the Yes and No responses for each condition, as presumably, these should isolate the neural correlates associated with each memory condition and there should be notable differences in those correlates for each memory condition (e.g. greater hippocampal responses in the unique event Yes > No contrast relative to the general facts Yes > No contrast).

We weighed the need to simplify and reduce to avoid “a sea of reporting” with the benefits of including ‘no’ responses in the analyses. The inclusion of ‘no’ diverges substantially from planned analyses and from the approach previously taken in Renoult et al. (2016). Our approach follows many memory researchers who retain only responses for which a memory trace is presumably present (e.g., Addis et al., 2011; Burianova and Grady, 2007).

The interpretation of ‘no’ responses remains ambiguous. For example, Maguire et al. (2001) explored the thought processes associated with ‘false’ (akin to ‘no’) responses in a pilot study about autobiographical memory, public events and general knowledge. The ‘false’ statements elicited recollection much like ‘true’ statements (Maguire et al., 2001). The literature on recall-to-reject further cautions that recollection is not specific to the recollection of a true event. In Maguire et al. (2001), the ‘false’ statements were modifications of ‘true’ statements, and so were associated with the recollection of elements associated with the ‘true’ statements. In our case, the statements could have additionally produced a mixture of absence of recall, and poor subjective and/or objective recollection of events. For example, for the statement “Yesterday evening, I took a shower”, a person could: 1) not recall anything, 2) know they didn’t or think about a relevant personal semantics (e.g., I always take showers in the morning, it’s always sunny when I take a shower), 3) think they may have taken a shower but be unsure about the timing or other aspects of the event; 4) recall going to the gym in the morning and taking a shower right after (i.e., recall a related event), or 5) recollect an outing at the movie theatre that fully occupied the evening (i.e., recall an unrelated event). Even more problematic for our purpose (i.e., contrasting the four memory conditions), the meaning of ‘no’ response could vary based on the condition. A ‘no’ response for Repeated events could mean the event never happened or it happened once but not repeatedly, for example. Subsequent research could take on the endeavour of contrasting “yes” and “no” responses, equipped with clear hypotheses about the differences between yes and no responses, and how that could alter the relation across memory conditions. In such a study, one might wish to equate the number of ‘yes’ and ‘no’ responses within each condition, measure subjective memory (e.g., confidence, vividness) after each response, and use alternative designs to account for the contribution of a retrieval mode.

In sum, we opt to not analyze ‘no’ responses to closely follow planned analyses, extending prior findings with ERPs now with fMRI (Renoult et al., 2016), and we retain only responses that generate greater interpretative clarity to illuminate the relation between memory conditions.

We added in the methods: “We retained only yes responses like Renoult et al. (2016) because they presumably reflect access to information consistent with the memory condition.”

9. I think the addition of the large volume of results in the appendix section is commendable in some respects, I am skeptical if a reader will actually consume the large volume of data in the tables. Perhaps linking to the overlay nifti files hosted on NeuroVault may be more appropriate?

We now make available the thresholded (80 voxels cluster) and unthresholded nifti files with bootstrap ratios for all lags on osf.io. Accordingly, we simplified all the tables to include only the Harvard-Oxford atlas (because of its importance in the in-text summary) for lag 7 and for additional lags in the appendices (i.e., Supplementary information).

Reviewer #2 (Recommendations for the authors):This study uses advanced fMRI methods to address important and theoretically-motivated questions about memory retrieval as it occurs in the real world. Central to this study is the notion that, when studying how people recall information for their lives, we can understand a potential intermediary form of memory that exists between episodic and semantic memory: personal semantics. The main study was an fMRI study in which participants responded to questions that involved accessing information from episodic memories, two forms of personal semantics – repeated events and autobiographical knowledge – as well as semantic memory. A parallel behavioural experiment on a different sample of participants was conducted to examine how these forms of information differ with respect to subjective ratings.A major strength of this paper is that it questions the classic division between episodic and semantic memory, proposing alternate views for how to consider retrieval from declarative memory. The use of PLS to analyze the data was a smart choice, albeit there are some considerations for these analyses. It is unfortunate that the neuroimaging and behavioural experiments were not collected on the sample. It might be useful to consider how the behavioural results can inform the neuroimaging experiment rather than considering the behavioural experiment solo.I have some suggestions to help strengthen the manuscript, which are listed below in order of presentation rather than importance.1. The introduction reviews all the key pieces of literature. When noting the overlap between episodic and semantic memory wrt the hippocampus, you may wish to include literature that notes how dementias characterised by semantic or episodic memory loss do have overlapping atrophy, particularly in the anterior hippocampusChapleau, M., Aldebert, J., Montembeault, M., and Brambati, S. M. (2016). Atrophy in Alzheimer's Disease and Semantic Dementia: An ALE Meta-Analysis of Voxel-Based Morphometry Studies. Journal of Alzheimer's disease: JAD, 54(3), 941-955. https://doi.org/10.3233/JAD-160382

Thank you for suggesting the addition of this highly relevant paper. We added a sentence and the reference in the introduction. “Anterior hippocampal atrophy features in neurodegenerative diseases affecting semantic and episodic memory alike (Chapleau et al., 2016).”

2. In the introduction. I think the rationale for the study could be strengthened by explicitly stating how the resulting work with contribute to new memory theories and extend from the reviewed literature. Simply said – note exactly what knowledge gap is being filled.

In the introduction, we justified the novelty of our approach through these points:

– Inferences about the differences between memory types rely predominantly on data from distinct studies or indirect comparisons (e.g., studies of semantic memory and of episodic memory).

– Further, even in direct comparisons, qualitative gaps exist between the typical conditions used to measure episodic versus semantic retrieval (e.g., gaps in task difficulty, reaction time [RT], and executive demands). Our study used stimuli that were closely matched across conditions and produced similar RTs across conditions in a prior ERP study (Renoult et al., 2016).

– Personal semantics is an understudied form of memory, despite evidence that it appears to be the more common for autobiographical memory elicited in free and cued recall (Barsalou, 1988) and in brain stimulation studies (Curot et al., 2017).

– Our review and taxonomy of personal semantics (Renoult et al., 2012) has generated quite a bit of interest (432 citations since) but very few empirical studies have investigated the neural correlates of the various types of personal semantics. Rarely, if ever, have two types of personal semantics been directly compared to both semantic and episodic memory. Their inclusion within a single study is key to understanding the neural substrates of declarative memory.

To address your comment, we added this paragraph to the introduction to clearly state the gap of knowledge: “In this study, we aimed to go beyond dichotomies commonly used in memory research (e.g., semantic vs. episodic, anterior vs. posterior brain regions, different vs. identical) to examine the multidimensional and complex relations across the spectrum in declarative memory, and importantly do so through direct comparisons. Our operationalization captures the prototypical definitions of several memory types, in close alignment with a taxonomy of personal semantics (Renoult et al., 2012), and possible characteristic function in daily life. Additionally, the analyses aim to uncover patterns that could act like heuristics to characterize declarative memory function. Critically, however, our additional focus on component processes in the behavioural study (i.e., amount of visual details, ability to evoke a scene, self-relevance) relies on the theoretical perspective that relations between memory types can be explained through the information accessed and cognitive processes engaged. Thus, it is implicit to our approach that the relations across memory types are not rigid and could be altered depending on task or personal goal (e.g., Grilli and Verfaellie, 2016). This study seeks to develop a framework suitable to bridge the divide in research about semantic and episodic aspects in declarative memory, and offers a complementary approach to explore the multiplicity of factors that coalescence to define the mnesic experience.”

3. In terms of the methods, the choice to block the conditions within each run raises the question of whether the results are due to differences in the content being accessed (semantic or episodic) or retrieval state (retrieval mode, in the words of Tulving) effects. Could the authors speak to this? (I come back to this with some suggestions for the discussion).

Thank you for this comment. The different types of cues used in our experiment indeed were used to trigger different “retrieval modes” in our participants, “a necessary condition for retrieval” (Tulving, 1983, page 169) that is maintained as a tonic state during a retrieval task (Rugg and Wilding, 2000). The behavioral data confirmed that this manipulation was successful in inducing typical phenomenological experience associated with these memory types. The self-relevance of the content being accessed was rated lowest for general facts compared to all personal forms of memory, and lower for personal semantics (i.e., autobiographical facts and repeated events) than unique events. Additionally, the amount of visual details retrieved increased from general facts to autobiographical facts to repeated events to unique events. Lastly, the four memory conditions differed in the proportion of scenes that came to mind during retrieval. A smaller proportion of general facts and autobiographical facts were categorized as scenes compared to repeated events and unique events, which did not differ in scene responses.

We added this statement in the discussion to clarify this distinction: “The different types of cues used in our experiment were used to trigger different “retrieval modes” in our participants, “a necessary condition for retrieval” (Tulving, 1983, p. 169) that is maintained as a tonic state during a retrieval task (Rugg and Wilding, 2000). The behavioral data revealed the sentence cues induced typical phenomenological experience associated with these memory types. Self-relevance was rated lowest for general facts compared to all personal forms of memory, and lower for personal semantics (i.e., autobiographical facts and repeated events) than unique events. Additionally, the amount of visual details increased from general facts to autobiographical facts to repeated events to unique events. This is consistent with previous studies. For example, as compared to unique events, repeated events are generally remembered less vividly: they are associated with reduced temporal specificity, personal significance, emotionality, and amount of details (e.g., Addis, Moscovitch, et al., 2004; Holland et al., 2011; Levine et al., 2004). Lastly, the four memory conditions differed in the proportion of scenes that came to mind during retrieval. A smaller proportion of general facts and autobiographical facts were categorized as scenes compared to repeated events and unique events, which did not differ in scene responses.

4. For the analysis, the authors note that the PLS analysis began at cue onset, and they estimated activity for 12 TR (14.4. seconds). What was the choice to analyze at the cue onset rather than the question? It seems to me that including the BOLD response associated with the cue means that your signal included both activities related to that cue, which is different across the conditions (some are temporal cues, some are activity cues, and some reflect people), as well as activity in accessing the intended information (a fact or event).

PLS does not make assumptions about the shape and timing of the hemodynamic response function (McIntosh et al. 2004). Therefore, for PLS analyses, it is recommended to include a large time-window to capture the entirety of the BOLD response (McIntosh et al. 2004). In our case, starting at cue onset signifies that the 12 TRs cover predominantly the cue, sentence and response periods. The percent signal change (see Figure 5) suggests the 12 TRs adequately encompass the rise, peak, and recovery of the BOLD response. Importantly, spatiotemporal PLS alleviates concern in interpretability as the temporal brain scores show the periods that contribute maximally to dissociate the memory conditions as set up in the contrasts, that is, in our case from TR 5 to TR 9 with the peak at TR 7. The peak occurs late in the epoch suggesting the cue period does not drive the findings. We think that the addition of a percent signal change figure (based on comment R1.4) significantly facilitates the interpretation of our findings.

5. I also found that the stimuli for the general facts condition seem to reflect opinions (Few people take pictures) rather than facts (The Queen of England is dead). Can the authors speak to this?

To begin, the General “facts” condition is reflective of semantic memory and its name mimics Autobiographical “facts”, much like Unique events is reflective of episodic memory and its name mirrors Repeated events. Several considerations motivated the operationalization of the General facts condition. First, the stimuli differed minimally in wording across conditions, being adjusted only in self-reference (i.e., referring to the self in all conditions but general facts) and temporal specificity (general for both types of facts, somewhat more specific for repeated events, and very specific for unique events). The general facts condition was thus operationalized as concerning knowledge of people in general (versus the self for the other conditions) and what they generally do (versus what they do at specific times). In that respect, the General facts condition takes the typical operationalization of general semantic memory as reflecting general knowledge of the world. For example, in the case of a “Few people take pictures”, a person could think of devices used to take pictures, occasions when people take pictures, and who usually takes pictures (e.g., parents, tourists).

According to the dictionary definition, opinions are “not based on evidence or ideas that everyone agrees on”, whereas facts are “undeniably true” (Antidote 9, Druide Informatique). To further match General facts with the other memory conditions, we did not manipulate the level of confidence or indisputability attached to statements. All conditions had some statements for which a “yes” response was more frequent (though “yes” was purposefully never unanimous), and some for which a “yes” response was less frequent. As you indicate, subjective feelings apply equally in semantic and episodic memory (cf. Mazancieux et al., 2020), and subjective feelings of confidence (for example) as well as accuracy may vary across statements within a condition, and this for all conditions. We did not consider accuracy because even if this were feasible for General facts statements, it would not be for the autobiographical memory conditions. Regardless of a statement’s truth in the absolute sense, we would expect a “yes” response to a statement to reflect a retrieval mode and access to information consistent with the memory condition.

Lastly, in future investigations one could consider comparing knowledge of the self to specific individuals (e.g., as you suggest “The Queen of England is dead”), which would certainly be interesting too to evaluate a more specific aspect of semantic memory, but may be associated with other challenges (e.g., either systematically comparing the self to a specific individual, limiting generalizability, or thinking about a different individual in different trials, which would add a task-switching element).

We added these sentences to the discussion: “The general facts condition was thus operationalized as concerning knowledge of people in general (versus the self for the other conditions) and what they generally do (versus what they do at specific times). In that respect, the general facts condition takes the typical operationalization of general semantic memory as reflecting general knowledge of the world. In future investigations, one could consider comparing knowledge of the self to knowledge of specific individuals, which would allow to evaluate a more specific aspect of semantic memory, but may be associated with other challenges (e.g., either systematically comparing the self to a specific individual, limiting generalizability, or thinking about a different individual in different trials, which would add a task-switching element).”

We edited the methods to include these statements: “We retained only yes responses like Renoult et al. (2016) because they presumably reflect access to information consistent with the memory condition. Memory accuracy is often difficult to assess for autobiographical memory (Cabeza and St Jacques, 2007), and so was not considered for any of the conditions.”

6. The PLS analysis also seems to be focused on yes responses, however, from Appendix 6 it seems like this means that between.6 and.4 proportion of the trials are not being analyzed. Would it not be useful for the authors in the trials by yes or no responses to see how ts drove neural activity?

Reviewer 1 raised this point (Comment R1.7). “We weighed the need to simplify and reduce to avoid “a sea of reporting” with the benefits of including ‘no’ responses in the analyses. The inclusion of ‘no’ diverges substantially from planned analyses and from the approach previously taken in Renoult et al. (2016). Our approach follows many memory researchers who retain only responses for which a memory trace is presumably present (e.g., Addis et al., 2011; Burianova and Grady, 2007).

The interpretation of ‘no’ responses remains ambiguous. For example, Maguire et al. (2001) explored the thought processes associated with ‘false’ (akin to ‘no’) responses in a pilot study about autobiographical memory, public events and general knowledge. The ‘false’ statements elicited recollection much like ‘true’ statements (Maguire et al., 2001). The literature on recall-to-reject further cautions that recollection is not specific to the recollection of a true event. In Maguire et al. (2001), the ‘false’ statements were modifications of ‘true’ statements, and so were associated with the recollection of elements associated with the ‘true’ statements. In our case, the statements could have additionally produced a mixture of absence of recall, and poor subjective and/or objective recollection of events. For example, for the statement “Yesterday evening, I took a shower”, a person could: 1) not recall anything, 2) know they didn’t or think about a relevant personal semantics (e.g., I always take showers in the morning, it’s always sunny when I take a shower), 3) think they may have taken a shower but be unsure about the timing or other aspects of the event; 4) recall going to the gym in the morning and taking a shower right after (i.e., recall a related event), or 5) recollect an outing at the movie theatre that fully occupied the evening (i.e., recall an unrelated event). Even more problematic for our purpose (i.e., contrasting the four memory conditions), the meaning of ‘no’ response could vary based on the condition. A ‘no’ response for Repeated events could mean the event never happened or it happened once but not repeatedly, for example. Subsequent research could take on the endeavour of contrasting “yes” and “no” responses, equipped with clear hypotheses about the differences between yes and no responses, and how that could alter the relation across memory conditions. In such a study, one might wish to equate the number of ‘yes’ and ‘no’ responses within each condition, measure subjective memory (e.g., confidence, vividness) after each response, and use alternative designs to account for the contribution of a retrieval mode.

In sum, we opt to not analyze ‘no’ responses to closely follow planned analyses, extending prior findings with ERPs now with fMRI (Renoult et al., 2016), and we retain only responses that generate greater interpretative clarity to illuminate the relation between memory conditions.

We added in the methods: “We retained only yes responses like Renoult et al. (2016) because they presumably reflect access to information consistent with the memory condition.”

7. In terms of the control task, the authors note why they included a control task e (page 7, line 11), but I would like to know why the authors selected the odd/even task. That is, what processes are being controlled for with this task? It seems to me that it would be general internally directed attention vs externally directed attention rather than specifying a memory network, per se. Does the choice of this task alter the pattern pulled out by PLS?

We concur with you and the authors that inspired the original design of the study (Foster et al., 2012) that mathematical operations or judgements on numbers involve “external attention” (Foster et al., 2012, p. 15514), whereas memory judgements are internally focused. Several studies on autobiographical memory and future thinking use odd/even judgements (e.g., Parlar et al., 2018; Svoboda and Levine, 2009; Thakral et al., 2020) as memory-related cognition functions permeate internally-focused cognition and a contrast with an “odd/even” task helps to reveal the memory-related brain activity (Stark and Squire, 2001). We modelled our approach of an odd/even task during the interstimulus interval after Madore et al. (2016). Thus, the control condition facilitated the comparison between memory conditions at minimal cost to scan time due to its insertion within the interstimulus interval. We added a justification for the odd and even task in the methods: “An odd/even task is frequently used in autobiographical memory and future thinking research (Parlar et al., 2018; Svoboda and Levine, 2009; Thakral et al., 2020) to reveal the core memory network (Stark and Squire, 2001).”

8. In terms of results, from Figure 1 it also seems that the negative brain pattern was being represented more significantly by RE and UE than GF or AF (as the CI for the brain scores does not overlap) – is this correct? If so, this would suggest to me that there is more dissociation between facts vs events than commonalities.

This mean-centered PLS showed that the four memory conditions engaged a common set of regions when compared to the odd/even task. You are correct that some memory conditions appear to express that network more than others. This is conceptually similar to the findings from the second LV from that PLS analysis (described briefly in text), the non-rotated PLS now reported in the main text, and was previously shown in the mean-centered PLS with the memory conditions only. The direct juxtaposition of the linear and facts vs. events contrasts in the non-rotated PLS along with the percent signal change in Figure 5 help to clarify the extent of similarities and differences across memory conditions.

We clarified the interpretation in the discussion:

“ In contrast, the unique neural correlates were evident when examining the four memory conditions on their own. The non-rotated PLS converges with the data-driven PLS (see Appendix 4) to suggest the facts vs. events contrast dominates to explain the spatiotemporal relations across memory conditions, although the difference in covariance explained between the two LVs of the non-rotated PLS was slight. In fact, both a-priori contrasts captured aspects of the spatiotemporal relations adequately. The percentage of signal change (see Figure 5) illustrates the complementarity of the two perspectives to encapsulate the relation between memory conditions. That is, activity increased (or decreased) continuously across memory types, but the extent of the increase (or decrease) confers greater similarity between general and autobiographical facts, and between repeated and unique events. Thus, several regions showed a relatively small increase in activity from general facts to autobiographical facts, a relatively large increase from autobiographical facts to repeated events, and a relatively small increase from repeated events to unique events; these include the precuneus, posterior cingulate, angular gyrus and middle frontal gyrus bilaterally, and left parahippocampal gyrus, left hippocampus, and left middle/superior temporal gyrus (see Appendices 2 and 3, figure 1). Activity instead decreased in a commensurate manner predominantly in the right hemisphere, particularly the frontal pole, inferior frontal gyrus, and supramarginal cortex. These findings are compatible with a continuum perspective of declarative memory, because quantitative rather than qualitative variations in brain activity suffice to characterize the relation between these memory types (Renoult et al., 2012).”

9. In how the results are presented, I had a hard time integrating the behavioural and neuroimaging data, perhaps the paper would benefit from presenting the behavioural study first to show the distinctions in the memory cues and then the neuroimaging data to examine neural mechanisms.

We present the behavioural results first as suggested. We also revised the results and discussion to better integrate behaviour and neuroimaging results throughout the manuscript. Notably:

In the Results section: “We used a non-rotated PLS to test two theoretically plausible relations between the four memory conditions: a linear contrast (-3, -1, 1, 3) and one comparing general/autobiographical facts and repeated/unique events (-1, -1, 1, 1; abbreviated to facts vs. events subsequently). A linear contrast would be consistent with the continuum perspective of personal semantics (see Box 3; Renoult et al., 2012), which would predict an increase in activity from general facts to autobiographical facts, from autobiographical facts to repeated events, and from repeated events to unique events. The increase in visual details (described above) followed precisely this pattern. Similarly, personal relevance increased from general facts to personal semantics (autobiographical facts and repeated events) to unique events, suggesting similar dynamics between component processes (e.g., contextual specificity may increase along with personal relevance). The facts vs. events contrast would favour the view of personal semantics as a subtype of general semantics (Box 2; Renoult et al., 2012). Of all personal semantics, autobiographical facts corresponds best with this view due to its abstraction from events, its more objective quality than other forms of personal knowledge (e.g., trait knowledge), and the feeling of “knowing” the “facts” rather than recollecting events (i.e., “noetic” consciousness; Tulving, 2002). Repeated events would instead group with unique events as “event memory” due to the common construction of a scene (Rubin and Umanath, 2015). Indeed, participants in the behavioural study perceived scenes as frequently for repeated and unique events. However, it is less clear how one would accommodate the greater number of scenes evoked for autobiographical than general facts in a way that aligns with that perspective. Thus, the conjunction of the visualization of scenes, amount of visual details, and personal relevance agrees most with a continuum of contextual specificity.”

In the discussion: “The importance of situational or contextual elements featured strongly in the neuroimaging data as in the behavioural data. The core network, also known as the default mode network, has been subdivided into an anterior temporal network linked to “entities” (Ranganath and Ritchey, 2012; Reagh and Ranganath, 2018) or “conceptual remembering” (Sheldon et al., 2019) and a posterior medial network linked to “situational models” (Ranganath and Ritchey, 2012; Reagh and Ranganath, 2018) or “perceptual remembering” (Sheldon et al., 2019). Although these networks process different kinds of information, neither is strictly dedicated to semantic or episodic memory (Reagh and Ranganath, 2018; Sheldon et al., 2019). For instance, knowledge can facilitate the search and construction of events (Irish and Piguet, 2013) and semantic memory can integrate contextual information (e.g., Greenberg et al., 2009; Sheldon and Moscovitch, 2012). Accordingly, in our study the dissociation between general/autobiographical facts and repeated/unique events did not have a clear posterior medial to anterior temporal demarcation (i.e., posterior medial activity for events and anterior temporal activity for facts). Instead, regions primarily within the posterior medial network (i.e., angular gyrus, posterior cingulate gyrus, precuneus, and parahippocampal gyrus; Ritchey et al., 2015) dissociated memory conditions, whereas those within the anterior temporal network (i.e., frontal orbital cortex, inferior anterior temporal gyrus, temporal pole, bilateral amygdala, and perirhinal cortex; Ritchey et al., 2015) did not. Indeed, events contained greater contextual information, as suggested by the increased proportion of scenes they evoked, as compared to factual memories. Further, cues were more temporally specific for repeated/unique events than general/autobiographical facts. Consistent with this, many regions of the posterior medial network were associated with greater activity for repeated/unique events than general/autobiographical facts (similar to Ford et al., 2011; Levine et al., 2004; Maguire and Mummery, 1999), and showed a linear increase from the most general type of memory (i.e., general facts) to the most specific memory (i.e., unique events). Therefore, activity in regions associated with visuospatial processing (e.g., precuneus) and scenes (e.g., medial temporal regions) coheres with behavioural data to support the prominence of contextual specificity in determining the relation across memory types. Activity in medial frontal regions is in harmony with ratings of visual details and scene perception, likewise increasing along a continuum of contextual specificity (see Figure 5d). However, this anterior portion of the medial frontal cortex may correspond best to self-processing rather than “situational” processing or mental time travel (Lieberman et al., 2019). If activity in the medial frontal cortex in our study reflected exclusively self-processing, one would expect a greater proximity between autobiographical facts and repeated events on the basis of subjective ratings of self-relevance. The additional concordance of medial frontal cortex with a continuum of contextual specificity could be a corollary of the strong links between aspects of self-relevance and episodic simulation (Tulving et al., 2002; King et al., 2022; Grysman et al., 2013; Verfaellie et al., 2019), in addition to this region’s role in modulating recollection (McCormick et al., 2020), for example through engaging schema-related information (Gilboa and Marlatte, 2017).”

10. The authors introduce the idea of the results reflecting the engagement of the DMN and different subsystems, but their results present patterns of activity that are not constrained to these networks or reflect connectivity, a better metric of network engagement. Perhaps the authors could constrain the PLS analysis to the DMN network so that they map their findings to the reviewed work about DMN contributions to autobiographical memory.

Your observation aligns with Comment R1.6. This was our response:

“In response to this comment, we repeated the non-rotated PLS using a mask of the default mode network (DMN; anterior temporal, medial prefrontal, posterior medial) and medial temporal network from Barnett et al. (2021) to test whether these regions would suffice to dissociate the memory conditions. Note that Barnett et al., conceptualize an MTL network as dissociable from the DMN, however the authors reference this subnetwork as a critical DMN bridge between DMN and the hippocampus and other networks. In light of this interpretation and the more general disagreement to date about the exact neuroanatomical boundaries of the DMN, we felt it was important to include Barnett's MTL network in our mask. Using Barnett et al.’s (2021) mask rather than a mask of the extended core network from our data (i.e., negatively weighted regions now shown in Appendix 4**—**figure 1) helped to avoid circularity.

The results of this ancillary non-rotated PLS were comparable to those from the main text. To further highlight the importance of the core memory network in determining the relation between memory conditions, we now display brain scores overlayed on Barnett et al.’s (2021) mask in Appendices 2 and 3 Figure**—**1, Appendix 4**—**Figure 1. The regions associated with a linear increase and showing increased activity for repeated/unique events than general/autobiographical facts predominantly fall within Barnett et al.’s (2021) networks. Altogether our data are consistent primarily with a quantitative rather than qualitative difference between memory conditions, that is, with mostly the same regions being engaged but to a different extent. Nevertheless, our data do not exclude the contribution of other regions. Regions associated with less decreased activity for general/autobiographical facts than repeated/unique events (i.e., negative saliences in Table 4) fall largely outside of the extended core network shown in Appendix 4**—**figure 1 and Barnett et al.’s network: the right frontal pole, the right inferior frontal gyrus, the right lateral occipital cortex, the right supramarginal gyrus, and the left occipital cortex. We nuanced the conclusions accordingly.

We added the following footnote in the discussion: “We obtained comparable non-rotated PLS results while including only voxels within the default mode and medial temporal networks from Barnett et al. (2021): 49.56% crossblock covariance (*p* <.001) for the linear contrast, and 50.44% crossblock covariance (*p* <.001). Within the selected networks, the same regions contributed to dissociate the memory conditions, and temporal brain scores peaked at lag 7. This supplementary analysis reinforces the importance of regions within the core memory network to determine the relation between memory conditions (see Appendices 2 and 3, figure 1), even though we found additional contributors at the whole brain level.”

11. I also think it would be great to focus the discussion on specific theories that fall within a component process framework to suggest why the resulting patterns emerged, such as theories that result reflect accessing context dependent vs independent information (Grilli et al.,) or different levels of consciousness (Tulving). In fact, I was curious if the authors thought that the distinctions are because different content is being accessed or the same content at different levels of representation. Both have relevance to theories on accessing information from memory that are noted. For example, the PMAT model discussing distinctions in content versus the TTT (Gilboa and Moscovitch) would note distinctions in representation, at least that is my understanding.

The emphasis on the component process view in the conclusion is compatible with numerous perspectives and theories as listed here: “*Some* of the characteristics that would influence differences in hippocampal activity and other regions of the core memory network include: the number of details (Thakral et al., 2020), their association (Duff et al., 2020; Solomon and Schapiro, 2020), their integration within a scene (Nadel, 1991; Robin, 2018; Robin et al., 2016; Rubin and Umanath, 2015) or within a situational model (Reagh and Ranganath, 2018; Summerfield et al., 2010), their coarseness and precision (Craik, 2020; Ekstrom and Yonelinas, 2020), their type and modality (e.g., perceptual, spatial, temporal, social; Binder and Desai, 2011; Grilli and Verfaellie, 2016; Sheldon et al., 2019), their stability (Auger and Maguire, 2018), as well as their projection into a temporally-distant time (Andrews-Hanna et al., 2010), the open-endedness of the representation (Sheldon and Moscovitch, 2012), the demands on pattern separation to construct unique representations or identify distinguishing features (Rolls and Kesner, 2006; Schapiro et al., 2017), and the likelihood of eliciting a specific event (Renoult et al., 2015; Westmacott et al., 2004; Westmacott and Moscovitch, 2003).” Future research will be necessary to better delineate which account(s) best explains relations between memory conditions.

Nevertheless, we clarify our perspective by spelling out the assumption of a “neural-psychological representation correspondence” (Moscovitch and Gilboa, 2022) that integrates elements of consciousness as additional “components” (e.g., the feeling of the self travelling in time; Tulving, 2002).

The question of the distinction between "different content” and "different levels of representation of the same content" is a difficult one and prone to different answers depending on how one would operationalise the notion of “content”. In our experiment, even though the sentence clauses referred to the same content across conditions (e.g., watering plants), the representational content of the elicited memories would not have been identical due to the use of the different cues, encouraging participants to process the stimuli with different levels of temporal specificity and personal relevance (e.g., Last week-end, I watered a plant versus Most people water plants).

In the discussion: “What underlies this overlap in the neural substrates of semantic and episodic memory? A parsimonious explanation is that semantic and episodic memory rely on similar elementary component processes (Cabeza and Moscovitch, 2013; Larsen, 1992; Moscovitch, 1992; Renoult et al., 2012; Rubin, 2021). All types of memories would depend on a similar network of brain regions but with different weighting of certain nodes in the network. The identification of the relative contribution of different component processes is a critical next step. *Some* of the characteristics that would influence differences in hippocampal activity and other regions of the core memory network include: the number of details (Thakral et al., 2020), their association (Duff et al., 2020; Solomon and Schapiro, 2020), their integration within a scene (Nadel, 1991; Robin, 2018; Robin et al., 2016; Rubin and Umanath, 2015) or within a situational model (Reagh and Ranganath, 2018; Summerfield et al., 2010), their coarseness and precision (Craik, 2020; Ekstrom and Yonelinas, 2020), their type and modality (e.g., perceptual, spatial, temporal, social; Binder and Desai, 2011; Grilli and Verfaellie, 2016; Sheldon et al., 2019), their stability (Auger and Maguire, 2018), as well as their projection into a temporally-distant time (Andrews-Hanna et al., 2010), the open-endedness of the representation (Sheldon and Moscovitch, 2012), the demands on pattern separation to construct unique representations or identify distinguishing features (Rolls and Kesner, 2006; Schapiro et al., 2017), and the likelihood of eliciting a specific event (Renoult et al., 2015; Westmacott et al., 2004; Westmacott and Moscovitch, 2003). For instance, episodic memory would typically rely to a greater degree than semantic memory on rich sensory-perceptual imagery, complex situational models or scenes, spatial and temporal features, and self-reflection. Accordingly, instead of activating different networks of brain regions, semantic and episodic processes may activate a similar network but with different degrees of magnitude, or recruit these brain regions in a complementary manner (Sherman et al., 2023). How each component is involved would also depend on the task at hand (e.g., Gurguryan and Sheldon, 2019); each component could be more or less engaged regardless of the memory type (e.g., semantic details can be thought of in rich details, in relation to the self, or in relation to a spatial context). Therefore, there would be a “neural-psychological representation correspondence” (Moscovitch and Gilboa, 2022) that includes elements of consciousness (e.g., feeling of being transported in time; Tulving, 2002) and that transcends categories of memory.”

Other sections of the manuscript also discuss our findings in relation to theoretical perspectives from the literature. In the introduction: “Critically, however, our additional focus on component processes in the behavioural study (i.e., amount of visual details, ability to evoke a scene, self-relevance) relies on the theoretical perspective that relations between memory types can be explained through the information accessed and cognitive processes engaged. Thus, it is implicit to our approach that the relations across memory types are not rigid and could be altered depending on task or personal goal (e.g., Grilli and Verfaellie, 2016). This study seeks to develop a framework suitable to bridge the divide in research about semantic and episodic aspects in declarative memory, and offers a complementary approach to explore the multiplicity of factors that coalescence to define the mnesic experience.”

In the discussion: “The importance of situational or contextual elements featured strongly in the neuroimaging data as in the behavioural data. The core network, also known as the default mode network, has been subdivided into an anterior temporal network linked to “entities” (Ranganath and Ritchey, 2012; Reagh and Ranganath, 2018) or “conceptual remembering” (Sheldon et al., 2019) and a posterior medial network linked to “situational models” (Ranganath and Ritchey, 2012; Reagh and Ranganath, 2018) or “perceptual remembering” (Sheldon et al., 2019). Although these networks process different kinds of information, neither is strictly dedicated to semantic or episodic memory (Reagh and Ranganath, 2018; Sheldon et al., 2019). For instance, knowledge can facilitate the search and construction of events (Irish and Piguet, 2013) and semantic memory can integrate contextual information (e.g., Greenberg et al., 2009; Sheldon and Moscovitch, 2012). Accordingly, in our study the dissociation between general/autobiographical facts and repeated/unique events did not have a clear posterior medial to anterior temporal demarcation (i.e., posterior medial activity for events and anterior temporal activity for facts). Instead, regions primarily within the posterior medial network (i.e., angular gyrus, posterior cingulate gyrus, precuneus, and parahippocampal gyrus; Ritchey et al., 2015) dissociated memory conditions, whereas those within the anterior temporal network (i.e., frontal orbital cortex, inferior anterior temporal gyrus, temporal pole, bilateral amygdala, and perirhinal cortex; Ritchey et al., 2015) did not. Indeed, events contained greater contextual information, as suggested by the increased proportion of scenes they evoked, as compared to factual memories. Further, cues were more temporally specific for repeated/unique events than general/autobiographical facts. Consistent with this, many regions of the posterior medial network were associated with greater activity for repeated/unique events than general/autobiographical facts (similar to Ford et al., 2011; Levine et al., 2004; Maguire and Mummery, 1999), and showed a linear increase from the most general type of memory (i.e., general facts) to the most specific memory (i.e., unique events). Therefore, activity in regions associated with visuospatial processing (e.g., precuneus) and scenes (e.g., medial temporal regions) coheres with behavioural data to support the prominence of contextual specificity in determining the relation across memory types. Activity in medial frontal regions is in harmony with ratings of visual details and scene perception, likewise increasing along a continuum of contextual specificity (see Figure 5d). However, this anterior portion of the medial frontal cortex may correspond best to self-processing rather than “situational” processing or mental time travel (Lieberman et al., 2019). If activity in the medial frontal cortex in our study reflected exclusively self-processing, one would expect a greater proximity between autobiographical facts and repeated events on the basis of subjective ratings of self-relevance. The additional concordance of medial frontal cortex with a continuum of contextual specificity could be a corollary of the strong links between aspects of self-relevance and episodic simulation (Tulving et al., 2002; King et al., 2022; Grysman et al., 2013; Verfaellie et al., 2019), in addition to this region’s role in modulating recollection (McCormick et al., 2020), for example through engaging schema-related information (Gilboa and Marlatte, 2017).”